SciPost Physics

Submission

# Ultrafast Control of Magnetic Correlations in a Heisenberg Spin Ladder

T. Ren[*], R. M. Konik[†]

Condensed Matter and Materials Physics Division, Brookhaven National Laboratory,
Upton, NY 11973-5000, USA
*tren@bnl.gov †rmk@bnl.gov

August 31, 2023

## Abstract

We study the time-dependent response of a Heisenberg spin ladder subjected to a time-dependent square form variation of its rung spin exchange coupling. To do so, we employ a field theoretic representation of the Heisenberg spin ladder consisting of a singlet and a triplet of Majorana fermions. Because this underlying description is free fermionic, we are able to develop closed form analytic expressions for dynamical quantities, both one-body measures of local spin correlations as well as two-time correlation functions. These expressions involve both the gaps to triplet and singlet excitations. We analyze these expressions obtaining both the time scales for their transients and long-time athermal steady state behaviors. We show that variations in the rung coupling are directly tied to changes in the local antiferromagnetic correlations. We further discuss the application of these results to pump-probe experiments on material realizations of low-dimensional magnetic systems.

# 1   Introduction

Resonant inelastic X-ray scattering (RIXS) is in general a complicated spectroscopy involving a two-step process [1–6]. In the first step a core level electron is excited into the material's valence band. In the second, the core hole is filled in one of two ways. In indirect RIXS the excited photoelectron decays filling the core hole but not before the core hole potential has created particle-hole excitations near the Fermi surface. In direct RIXS, the core hole is filled by a valence electron other than the photoelectron, and so again leaves the system with a low-energy particle-hole excitation. In certain cases, however, this two-step process maps onto to a simpler, purely magnetic, interaction. In iridate materials such as $Sr_2IrO_4$ where spin-orbit coupling is present and where the low lying excitation manifold consists of effective spin-1/2 excitations, the response function for direct RIXS at the Ir L-edge is equivalent to the dynamical spin structure factor [7]. In Cu L-edge direct RIXS on cuprate materials, the response function is again equivalent to the dynamical spin structure factor and both magnonic and bimagnonic excitations can be observed [8–14]. In cuprate Cu K-edge indirect RIXS, the response function is that of the bilinear spin-exchange operator. In cuprate chain materials the expected K-edge

response is however dominated by two-spinon instead of four-spinon excitations [15].

RIXS as a technique to measure magnetic excitations is now well established. More recently, RIXS measurements have been made in a pump-probe framework at X-ray free electron laser facilities to measure transient dynamics in a magnetic material after femtosecond pumping by a laser [16–20]. In Ref. [16], it was shown that a laser pump applied to $Sr_2IrO_4$ disrupted both two-dimensional (2D) and long range 3D order. The 2D order was seen to recover in a few picoseconds while the 3D order took hundreds of picoseconds to return to it pre-pump value, a result confirmed in [21]. In Ref. [18], an ultrafast pump was shown to make the static density wave order present in the 1/8-th doped cuprate $La_{1.875}Ba_{0.125}CuO_4$ mobile.

Like all time-dependent responses in a correlated system, computing the time-dependent RIXS response is theoretically challenging. Modeling it fully requires the computation of a four-time correlation function [22]. In the case when the RIXS response maps onto the spin response, one is left with a two-time correlation function. To evaluate such a correlation function in a correlated system typically requires the use of numerics, either exact diagonalization [22–26] or non-equilibrium dynamical mean field theory [27–30]. This makes the interpretation and quantification of the time-dependent dynamics more challenging. Thus a case, even a simplified one, where an analytical description of the time-dependence response is available is valuable. It is the purpose of this paper to offer such an example in the form of Heisenberg spin ladders.

Heisenberg spin ladders with extended sets of interactions (such as ring exchange) are interesting correlated systems. By adjusting the sign of the Heisenberg rung coupling $J_\perp$ one can move from a Haldane phase where the ladder is an effective spin-1 chain and supports symmetry protected topological edge states to an antiferromagnetic rung singlet phase. And by allowing a ring exchange term $J_\times$ the ladder sees two types of valence bond phases where dimer-like correlations exist along the legs of the ladders. Heisenberg spin ladder systems where the spin (and not charge) degrees of freedom are dominant are realized in such cuprates as $SrCu_2O_3$ [31].

Remarkably this many-body Hamiltonian has a simple low-energy description. By converting the bosonic spin degrees of freedom to fermions via a Jordan-Wigner transformation, followed by a bosonization and then subsequent refermionization, the spin ladder admits a description in terms of four Majorana fermions whose only interaction is of a marginal current-current form [32–34]. At low energies, this marginal interaction can be ignored and the ladder reduces to four non-interacting fermions. The four fermions are organized under $SU(2)$ symmetry as a triplet and a singlet and correspond to the elementary excitations of the ladder.

Because the Heisenberg spin ladder admits a low energy free fermionic description in terms of four Ising models, it becomes possible to write down closed form expressions for the system's non-equilibrium response functions. This however does not mean all the correlations are trivial. While the slowly varying component of the correlations along the legs of the ladder are described by fermion bilinears, the staggered components are given in terms of the Ising spin and disorder operators of the four Ising models. These have non-trivial correlation functions. The non-equilibrium behaviour of the Ising spin/disorder operators in individual Ising models have been studied previously [35–37], work that we will exploit here.

In previous experimental time-resolved RIXS (tr-RIXS) studies such as [16, 17] on iridate based materials, the pump served to disrupt the magnetic order by creating doubly occupied sites. Here we consider a different sort of disruption of the magnetic order, one induced by changing the magnetic exchange, $J_\perp$, between the rungs of the ladders during the pump window. The possibility of adjusting this coupling in an experimental setting has

been discussed in Ref. [38]. In Ref. [38], the authors proposed to displace atoms from the equilibrium positions through using ionic Raman scattering. In the Heisenberg limit, one expects that the interleg coupling, $J_\perp$, to behave as $t_\perp^2/U$ and that $t_\perp$ will depend on the interatomic spacing sensitively. In order to retain analytical control over the computation, we will imagine that the adjustment in $J_\perp$ takes place in a double step fashion. We do not believe this simplification will change the long time behavior of the system.

The paper is organized as follows. In Section 2 we introduce the reader to the Majorana description of the Heisenberg spin ladders. We outline there how the spin ladders can be mapped to four Ising models and how the relevant observables can be understood in terms of the operators in the four Ising models. In Section 3 we describe the pump-probe protocol and how the ground state evolves under the pump. We can write the evolved ground state in the form of an exponential of fermion bilinears. In Section 4, we then consider how one body operators (energy density, local antiferromagnetic correlations) evolve in time. Interestingly, we find that the spin-spin correlator across the rung tracks the change in $J_\perp$ closely. This suggests a simple means to control local magnetic correlations. In Section 5, we move to computing the more involved two-time non-equilibrium correlation functions. In Section 6, we wrap the paper up. This work has several appendices in which we place more technical manipulations associated with our calculations.

## 2 Overview of Heisenberg Spin Ladders and their Majorana Description

### 2.1 Heisenberg Spin Ladder

We consider the periodic, $SU(2)$-invariant antiferromagnetic Heisenberg ladder governed by the Hamiltonian

$$H = J \sum_{\ell=1,2} \sum_{r=0}^{N-2} \boldsymbol{S}_{\ell,r} \cdot \boldsymbol{S}_{\ell,r+1} + J_\perp \sum_{r=0}^{N-1} \boldsymbol{S}_{1,r} \cdot \boldsymbol{S}_{2,r} + J_\times \sum_{r=0}^{N-2} (\boldsymbol{S}_{1,r} \cdot \boldsymbol{S}_{1,r+1})(\boldsymbol{S}_{2,r} \cdot \boldsymbol{S}_{2,r+1}), \quad (2.1)$$

where $\boldsymbol{S}_{\ell,r}$ is the spin-1/2 operator located on leg $\ell$ and rung $r$ of the ladder. The couplings $J = 1, J_\perp, J_\times$ characterize leg, rung, and plaquette exchange interactions, respectively. For uncoupled Heisenberg chains, the total spin of each leg would be conserved and $S_\ell^z = \sum_r S_{\ell,r}^z, \ell = 1, 2$ are good quantum numbers. The coupling $J_\perp$ exchanges spin in integer units, $S_1^z \to S_1^z \pm 1, S_2^z \to S_2^z \mp 1$, violating the conservation of individual $S_\ell^z, \ell = 1, 2$, but constraining the even and odd combinations, $S_\pm^z = S_1^z \pm S_2^z$, such that $S_+^z$ is conserved and $S_\pm^z$ have the same parity:

$$S_+^z \equiv S_-^z \ (\mathrm{mod}\, 2). \quad (2.2)$$

This implies that the model in (2.1) has an underlying $U(1)$ symmetry, reflecting the conservation of $S_+^z$, together with a $\mathbb{Z}_2$ condition, implementing the constraint (2.2). The Heisenberg ladder described by Eq. (2.1) can be experimentally realized in ladder materials where the fluctuations of spin-1/2 degrees of freedom are dominant, such as $SrCu_2O_3$ [31, 39], or in cold atom systems [40, 41]. Ladder systems have much of the same physics as their fully 2D counterparts [39]. We thus expect our study of them here will provide useful insights into ultrafast studies of Mott insulating strontium iridium oxides [16, 17].

### 2.2 Derivation of Majorana Representation

All the structures and phenomena alluded to above afford a simple and surprisingly quantitative description in the language of Majorana fermions. The first step in deriving the

Majorana representation is abelian bosonization [32, 33, 42]. Here, we adopt the convention used in [34]. We begin by considering the spin operator $\boldsymbol{S}_{\ell,r}$ at the point $r = x/a_0$ of the $\ell$-th leg, where $a_0$ is the lattice constant. The abelian bosonized description involves splitting the spin operator into a smooth and staggered component, with both components expressed in terms of a bosonic field $\Phi_\ell$ and its dual $\Theta_\ell$:

$$
\begin{aligned}
\frac{S_{\ell,r}^z}{a_0} &= -\frac{1}{2\sqrt{2}\pi}\partial_x\Phi_\ell(x) + \frac{\lambda(-1)^r}{2\pi a_0}\sin\left(\frac{\Phi_\ell(x)}{\sqrt{2}}\right), \\
\frac{S_{\ell,r}^\pm}{a_0} &= \frac{\lambda e^{\pm i\Theta_\ell(x)/\sqrt{2}}}{2\pi a_0}\left[\cos\left(\frac{\Phi_\ell(x)}{\sqrt{2}}\right) + (-1)^r\right],
\end{aligned}
\tag{2.3}
$$

where $\lambda$ is a non-universal constant related to the frozen degrees of freedom of a parent Hubbard ladder [32, 33, 42]. Substituting Eq. (2.3) into the Hamiltonian (2.1), we arrive at the bosonized description of the Heisenberg spin ladder:

$$
H = \int \mathrm{d}x\left[\frac{v}{8\pi}\sum_{\alpha=\pm}\left[(\partial_x\Theta_\alpha)^2 + (\partial_x\Phi_\alpha)^2\right] + \sum_{\alpha=\pm}g_\alpha\cos\left(\Phi_\alpha\right) + g'\cos\left(\Theta_-\right)\right],
\tag{2.4}
$$

where we have ignored the marginal interactions and their renormalization effect on the velocity. The couplings of the nonlinear terms are related to the microscopic parameters through $g_\pm \propto (9J_\times/\pi^2 \mp J_\perp)$ and $g' \propto 2J_\perp$, and we use symmetric/antisymmetric combinations of the bosonic fields $\Phi_\pm = (\Phi_1 \pm \Phi_2)/\sqrt{2}, \Theta_\pm = (\Theta_1 \pm \Theta_2)/\sqrt{2}$. The symmetric sector of Eq. (2.4) is described by an integrable sine-Gordon model [43, 44], while the antisymmetric sector is a sine-Gordon model perturbed by the cosine of the dual field.

The next step in the derivation is to refermionize the theory by introducing the spinless chiral fermion fields:

$$
\psi_{\pm,L} = \frac{\kappa_\pm}{\sqrt{2\pi a_0}}e^{\frac{i}{2}(\Phi_\pm + \Theta_\pm)}, \quad \psi_{\pm,R} = \frac{\bar{\kappa}_\pm}{\sqrt{2\pi a_0}}e^{-\frac{i}{2}(\Phi_\pm - \Theta_\pm)},
\tag{2.5}
$$

where $\kappa_\pm, \kappa_\pm$ are Klein factors that ensure the anti-commutation relations of fermions of different species, $\{\kappa_a, \kappa_b\} = \delta_{ab}, \{\kappa_a, \bar{\kappa}_b\} = 0, \{\bar{\kappa}_a, \bar{\kappa}_b\} = \delta_{ab}$. We subsequently express the fermion fields in terms of their real and imaginary components, namely the Majorana fermions (below $p$ takes the value $L$ or $R$):

$$
\psi_{+,p} = \left(\xi_p^2 + i\xi_p^1\right)/\sqrt{2}, \quad \psi_{-,p} = \left(\xi_p^3 + i\xi_p^0\right)/\sqrt{2},
\tag{2.6}
$$

where the doublets $(\xi^2, \xi^1)$ and $(\xi^3, \xi^0)$ represent the symmetric and antisymmetric sectors, respectively. Finally, we arrive at the low energy field theory of our model defined in Eq. (2.1) in terms of free Majorana fermions:

$$
H = \int \mathrm{d}x\left[-\frac{iv}{2}\left(\xi_R^0\partial_x\xi_R^0 - \xi_L^0\partial_x\xi_L^0\right) - im_s\xi_R^0\xi_L^0 - \frac{iv}{2}\left(\boldsymbol{\xi}_R\partial_x\boldsymbol{\xi}_R - \boldsymbol{\xi}_L\partial_x\boldsymbol{\xi}_L\right) - im_t\boldsymbol{\xi}_R\cdot\boldsymbol{\xi}_L\right].
\tag{2.7}
$$

Here the Majorana fermions are rearranged into a singlet, $\xi^0$, and a triplet, $\boldsymbol{\xi} = (\xi^1, \xi^2, \xi^3)$, with the corresponding masses

$$
m_s \propto 9J_\times/\pi^2 + 3J_\perp, \quad m_t \propto 9J_\times/\pi^2 - J_\perp.
\tag{2.8}
$$

The phase boundaries of the system are then determined by the vanishing of the Majorana masses. In the Majorana representation, the $U(1) \simeq O(2)$ symmetry of the $S_+^z$ sector is realized as a continuous rotational symmetry between the mass-degenerate fields $(\xi^2, \xi^1)$, and the $\mathbb{Z}_2$ symmetry of the $S_-^z$ sector is realized via the sign inversion of fields $(\xi^3, \xi^0)$.

Importantly, these Majorana fields are in fact not independent but correlated via the $\mathbb{Z}_2$ condition (2.2). In the Majorana representation, the global $S_\pm^z$ quantum numbers assume the form $S_+^z = i \sum_a \xi_a^2 \xi_a^1 / 2$ and $S_-^z = i \sum_a \xi_a^3 \xi_a^0 / 2$, where $\sum_a$ is a formal sum over all eigenmodes of the system. The $\mathcal{Z}_2$ condition (2.2) then translate to

$$\exp\left(\pi \sum_a \xi_a^1 \xi_a^2 / 2\right) = \exp\left(\pi \sum_b \xi_b^3 \xi_b^0 / 2\right), \tag{2.9}$$

introducing entanglement between the four Majorana fermions and constraint on the ground state degeneracy of each phase.

Depending on the couplings $J_\perp, J_\times$, the four possible combination of signs for $m_s$ and $m_t$ can be realized. This corresponds to different phases with different dimerization patterns that can be supported by the original Hamiltonian defined in Eq (2.1) [34]. A schematic phase diagram from [34] is reproduced in Fig. 2.1. For strong positive $J_\perp$ and weak $J_\times$ such that $m_s > 0, m_t < 0$, the formation of rung singlets is favored. For strong negative $J_\perp$ such that $m_s < 0, m_t > 0$, rung triplets are formed which effectively implement the Haldane phase of the $S = 1$ spin chain [45]. For strong $J_\times$ such that $m_s m_t > 0$, there are two types of valence bond solids [46] distinguished by different types of periodically repeated intra-chain dimerization. The phase diagram as well as the topological property of each phase were determined in [34], where the constraint (2.2) plays an important role in determining the ground state degeneracy of each phase.

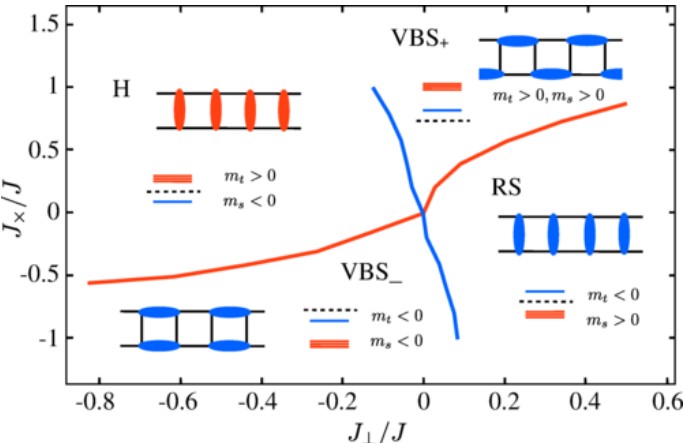

Figure 2.1: The schematic phase diagram for the antiferromagnetic Heisenberg ladder. From Phys. Rev. Lett. 122, 027201 by N. J. Robinson *et. al.*, 2019. Copyright (2019) by the American Physical Society. Reprinted with permission.

## 2.3 Description of Spin Operators in Terms of Majorana Description

We are concerned with the correlation functions of the spin operators. Each spin operator has a smooth component and a staggered component [47]:

$$S_{\pm,r}^z = M_{\pm,r} + (-)^r N_{\pm,r}. \tag{2.10}$$

The smooth component can be written locally as Majorana fermion bilinears [32]:

$$
\begin{aligned}
M_{+,r} &= \frac{i}{2}\left[\xi_R^2(x_r)\xi_R^1(x_r) + \xi_L^2(x_r)\xi_L^1(x_r)\right], \\
M_{-,r} &= \frac{i}{2}\left[\xi_R^3(x_r)\xi_R^0(x_r) + \xi_L^3(x_r)\xi_L^0(x_r)\right],
\end{aligned}
\tag{2.11}
$$

such that the calculation of their correlation functions can be carried out directly in the Majorana representation.

In contrast to the smooth components, the staggered components cannot be expressed locally in terms of the Majorana fermions. Their representation is given instead in terms of the Ising spin and disorder operators of each Majorana [32]. Each species of the four species of Majorana fermions describes a copy of the quantum Ising model, with a positive Majorana mass $m > 0$ corresponding to the ordered phase of the Ising theory and $m < 0$ corresponding to the disordered phase. Thus for each species of the Majorana fermions, we can introduce the associated Ising spin and disorder operators, $\sigma^i$ and $\mu^i$, $i = 0, 1, 2, 3$. Through the bosonization dictionary [32], the staggered component can be expressed in terms of these Ising spin and disorder operators:

$$
\begin{aligned}
N_{+,r} &= -\frac{\lambda}{\pi a_0} \sin\left(\frac{\Phi_+(x_r)}{2}\right) \cos\left(\frac{\Phi_-(x_r)}{2}\right) \\
&= \left(-\frac{\lambda}{\left|m_s m_t^3\right|^{1/8} a_0}\right) \mu^0(x_r)\sigma^1(x_r)\sigma^2(x_r)\mu^3(x_r), \\
N_{-,r} &= -\frac{\lambda}{\pi a_0} \cos\left(\frac{\Phi_+(x_r)}{2}\right) \sin\left(\frac{\Phi_-(x_r)}{2}\right) \\
&= \left(-\frac{\lambda}{\left|m_s m_t^3\right|^{1/8} a_0}\right) \sigma^0(x_r)\mu^1(x_r)\mu^2(x_r)\sigma^3(x_r),
\end{aligned}
\tag{2.12}
$$

where $N_{-,r}$ are related to $N_{+,r}$ by a duality transformation that exchanges $m \to -m, \sigma \leftrightarrow \mu$, such that their correlations are also related by such a duality transformation.

## 2.4 Quantum Ising Model

We have reduced the description of the spin ladders to that of four non-interacting Majorana fermions. Each Majorana fermion is equivalent to a massive quantum Ising model with mass $m$. The sign of the mass determines both whether the ground state of Ising Hamiltonian is ordered ($m > 0$) or disordered ($m < 0$) and the structure of the model's Hilbert space. Because of the centrality of the Ising model to our computations, we review in this subsection the details of this model, going over its space of states and the non-trivial matrix elements of the Ising spin and disorder operators. As we have seen from the previous subsection, these are needed to describe the staggered correlations of the ladder.

### 2.4.1 Massive Quantum Ising Model

The Hilbert space of the Majorana fermions is constrained by the original spin model. Since we are considering periodic boundary conditions on the spin fields, and the smooth components of the spin fields correspond to Majorana fermion bilinears, we see that we have two possibilities. The Majorana fermions can be periodic with integer moding, referred to as Ramond (R) fermions. But because we deal with fermion bilinears, the individual fermions are also allowed to be anti-periodic with half-integer modings. Such fermions are referred to as Neveu-Schwarz (NS) fermions. Below we discuss how to construct the Hilbert space of the spin ladder from the states in these two sectors as well as the matrix elements of the Ising spin operator, $\sigma$, out of which are composed the matrix elements of the staggered components of the spin operators, $N_\pm$.

### 2.4.2 Hilbert Space

To compute the matrix elements of the staggered operators, we nominally need the matrix elements of both the Ising spin, $\sigma$, and disorder, $\mu$, operators. However any correlator involving $\mu$ can be recast via the Kramers-Wannier duality as a correlator involving $\sigma$. We are thus going to focus on the part of the Hilbert space needed to construct the correlators of $\sigma$. For a periodic system of length $L$, insisting that the matrix elements of the Ising spin operator, $\sigma$, be single-valued, restricts the states in the Hilbert space coming from the NS and R sectors. In the ordered phase, $m > 0$, the Neveu-Schwarz/Ramond sectors are composed of states built from even numbers of half-integer/integer fermionic modes acting on two vacua $|0\rangle_{\text{NS}}/|0\rangle_{\text{R}}$. In the disordered phase, $m < 0$, the relevant Hilbert space consists of a single vacuum state, $|0\rangle_{\text{NS}}$, Ramond states involving odd numbers of fermions, and Neveu-Schwarz states involving even numbers of half-integer fermions. To summarize:

- $m > 0$: States in Ramond/Neveu-Schwarz sectors have even numbers of Majorana fermions with integer/half-integer modings.

- $m < 0$: States in the Ramond/Neveu-Schwarz sectors have an odd/even number of Majorana fermions with integer/half-integer modings.

At $m = 0$ the two descriptions continuously pass from one to another because the $q = 0$ mode in the Ramond sector sits at zero energy at $m = 0$. The $m \to 0$ limit of $m > 0$ Ramond states with even numbers of particles maps onto the $m \to 0$ limit of $m < 0$ Ramond states with odd numbers of particles through the addition of this zero mode. For $m > 0$, the presence of the vacuum states in both the R and NS sectors means that the $m > 0$ ground state is formed as one of the two linear combinations:

$$|0\rangle_{m>0} = \frac{1}{\sqrt{2}} \left[ |0\rangle_{\text{NS}} \pm |0\rangle_{\text{R}} \right]. \tag{2.13}$$

Such a linear combination would be selected in infinite volume as a result of spontaneous $\mathbb{Z}_2$ symmetry breaking.

### 2.4.3 Description of Form Factors of the Ising Spin, $\sigma$, Operators

Here we will record the non-trivial matrix elements of the Ising spin needed to compute the two-time correlation functions of the staggered spin components in later sections. These matrix elements are known as form factors and satisfy certain axiomatic constraints [48,49] that permits their determination. They are well understood in the literature [50]. The matrix elements of the spin operator, $\sigma$, are only non-zero for states connecting the two sectors, Ramond and Neveu-Schwarz:

$$
\begin{aligned}
{}_{\text{NS}}\langle k_1, k_2, \ldots, k_{2n} | \sigma(x) | p_1, p_2, \ldots, p_l \rangle_{\text{R}} &= e^{ix\left(\sum_{j=1}^{l} p_j - \sum_{i=1}^{2n} k_i\right)} (i)^{\left[n + \frac{l}{2}\right]} \bar{\sigma} S(L) \\
&\times \prod_{i=1}^{2n} \tilde{g}(\vartheta_{k_i}) \prod_{j=1}^{l} g(\vartheta_{p_j}) \prod_{0 < i < j \leqslant 2n} \tanh\left(\frac{\vartheta_{k_i} - \vartheta_{k_j}}{2}\right) \\
&\times \prod_{0 < s < t \leqslant l} \tanh\left(\frac{\vartheta_{p_s} - \vartheta_{p_t}}{2}\right) \prod_{\substack{0 < i \leqslant 2n \\ 0 < s \leqslant l}} \coth\left(\frac{\vartheta_{k_i} - \vartheta_{p_s}}{2}\right),
\end{aligned}
\tag{2.14}
$$

where $l$ is even for $m > 0$ and odd for $m < 0$, $\bar{\sigma} = 1.3578|m|^{1/8}$, $[\cdots]$ takes the integer part of the argument, and the rapidities $\vartheta_k, \vartheta_p$ are related to the momenta via

$$|m| \sinh \vartheta_k = vk, \quad |m| \sinh \vartheta_p = vp. \tag{2.15}$$

This expression for the $\sigma$ form factors is valid equally in the ordered and disordered phases. In the limit $L \to \infty$, we have with exponential accuracy that

$$S(L) \approx 1, \quad \tilde{g}(\vartheta) \approx g(\vartheta) \approx \sqrt{v}/\sqrt{|m|L \cosh \vartheta}. \qquad (2.16)$$

Again because we will always understand the correlation functions involving the disorder operator, $\mu$, via a Kramers-Wannier duality transformation, we do not record their matrix elements directly here.

### 2.4.4    Lehmann Expansions of Correlation Functions

According to Sec. 2.3, the staggered part of the spin operators cannot be expressed locally in terms of the Majorana fermions. On the other hand, they have representations in terms of the Ising spin and disorder operators in Eq. (2.12), whose non-trivial matrix elements are shown in the Sec. 2.4.3. Consequently, it is useful to write the correlation functions of the staggered part of the spin operators in the Lehmann expansion. Consider the following correlation function:

$$\langle \psi\left(t_2\right) | U\left(t_2, t_2'\right) N_{\pm} U\left(t_2', t_2\right) N_{\pm} | \psi\left(t_2\right) \rangle, \qquad (2.17)$$

where $t_2, t_2' > t_1$ are two post-quench times, and $U(t_2', t_2)$ is the evolution operator governed by the post-quench Hamiltonian:

$$U(t_2', t_2) = e^{-i(t_2' - t_2)H}. \qquad (2.18)$$

The Lehmann expansion of Eq. (2.17) is

$$
\begin{aligned}
&\langle \psi\left(t_2\right) | U\left(t_2, t_2'\right) N_{\pm} U\left(t_2', t_2\right) N_{\pm} | \psi\left(t_2\right) \rangle \\
&= \sum_{l,m,n} \langle \psi\left(t_2\right) | \varphi_n \rangle\, e^{iE_n\left(t_2' - t_2\right)} \langle \varphi_n | N_{\pm} | \varphi_m \rangle\, e^{-iE_m\left(t_2' - t_2\right)} \langle \varphi_m | N_{\pm} | \varphi_l \rangle \langle \varphi_l | \psi\left(t_2\right) \rangle,
\end{aligned}
\qquad (2.19)
$$

where $|\varphi_n\rangle$ are the eigenstates of $H$ with eigenenergy $E_n$. Both the eigenstates and $|\psi(t_2)\rangle$ can be expressed in the Majorana representation, such that the involved matrix elements are essentially the form factors introduced in Sec. 2.4.3. Although we cannot evaluate the sum in Eq. (2.19) in its full splendor, we can expect that a resummation of selected terms will accurately capture the most important contributions. This will be the strategy that we adopt for the calculation of the staggered correlations.

## 3    Description of Pump-Probe Protocol

### 3.1    The Pump as a Double Quench

We are interested in studying an experimentally realizable pump that targets the rung coupling $J_\perp$ which, in turn, alters the Majorana masses. Any practical pump will typically have a Gaussian profile, perhaps with higher harmonics superimposed, but we opt to replace it with a step-like profile for easy theoretical analysis (see Fig. 3.1). Correspondingly, the Majorana masses will have a similar time profile: at time $t < 0$, we have Majorana masses $m_s, m_t$; at time $0 < t < t_1$, they are changed to $M_{s,\text{pump}}, M_{t,\text{pump}}$; at $t > t_1$, they are changed back to $m_s, m_t$. We are interested in the universal features of the dynamical correlations in terms of these mass parameters. We will thus put particular focus on features that do not depend on the details of the quench protocol.

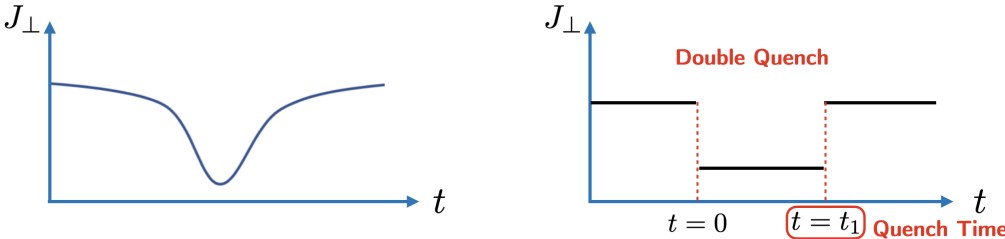

Figure 3.1: The quench protocol. (a) The time dependence induced by a Gaussian pump on $J_\perp$. (b) A toy version of (a) that is easier for theoretical investigate.

### 3.1.1   Choice of Parameters

We now present our choices for the parameters. We will take $J$ on the order of 200 meV and $t_1$ on the order of 100 fs. This implies $J \cdot t_1 \sim 30\hbar$. We will see that as long as $t_1$ is larger than a certain saturation time (a time that will be far smaller than this choice of $t_1$ for our particular choices of $J_\perp$ and $J_\times$), the dynamical correlations will present universal features independent of $t_1$. The velocity of the Majorana fermions scales as $v \sim a_0 \cdot J/\hbar$, where the lattice constant $a_0$ is on the order of several Å. In performing the calculations, we make the parameters dimensionless by setting $\hbar = 1, J = 1, v = 1$, such that we are measuring energy in units of $J$, and length in units of $a_0$, so that $t_1 = 30$. In choosing the parameters $J_\times$ and $J_\perp$ before and after the pump, we are faced with two constraints such that the field theory approach we employ is applicable. The first constraint is that the resulting Majorana masses in Eq. (2.8) should be smaller than $J$, and the second constraint is that the mode occupations populated by the pump should be significant only below the cutoff, which is on the order of $J/v$. On the other hand, we also want a considerable amount of energy pumped into the system (the calculation of which will be presented in Sec. 4.1), such that the resulting effect is appreciable. We have to compromise between these two goals, leading us to the following representative choice of parameters:

$$
\begin{aligned}
\text{set \#1: } & J_\times = 0.05J, \ J_{\perp,1} = 0.15J, \ J_{\perp,2} = 0.10J \\
& \Rightarrow \ m_s = 0.50J, \ M_{s,\text{pump}} = 0.35J; \quad m_t = -0.10J, \ M_{t,\text{pump}} = -0.05J, \\
& \Rightarrow \ \Delta E/L = 0.013J^2/v = 0.56J_\perp^2/v \\
\text{set \#2: } & J_\times = 0.05J, \ J_{\perp,1} = 0.06J, \ J_{\perp,2} = 0.04J \\
& \Rightarrow \ m_s = 0.23J, \ M_{s,\text{pump}} = 0.17J; \quad m_t = -0.014J, \ M_{t,\text{pump}} = 0.006J \\
& \Rightarrow \ \Delta E/L = 0.0028J^2/v = 0.79J_\perp^2/v
\end{aligned}
\tag{3.1}
$$

These two sets of parameters represent two different situations: one with quench within the same phase (the RS phase in Fig. 2.1), and the other with quench across the phase boundary (between the RS phase and the VBS$_+$ phase in Fig. 2.1).

### 3.2   Evolution of Ground State

Following this double quench protocol, the ground state at $t > 0$ evolves in a simple fashion that can be expressed in terms of a product of Majorana coherent states [37]. This simple evolution renders the computation of the spin correlation functions an analytically tractable task. The evolution of each of the four Majorana fermions composing our spin ladder occurs independently. To thus describe the overall evolution of the ground state,

we thus can focus on a single species of Majorana fermions (denoted by $\xi$) and its time-evolving quantum state. We write the Hamiltonian of $\xi$ at $t < 0$ and $t > t_1$ as

$$H = \int dx \left[ -i\frac{v}{2} \left( \xi_R \partial_x \xi_R - \xi_L \partial_x \xi_L \right) - im\xi_R \xi_L \right], \tag{3.2}$$

while during the quench, $0 < t < t_1$, the mass parameter is changed from $m$ to $M_{\text{pump}}$. We will approach this problem by expanding the Majorana fermion in terms of the eigenmodes, and derive the relation between the eigenmodes with mass parameters $m$ and $M_{\text{pump}}$. This approach requires careful attention to the boundary conditions of the fermions. We have two possible boundary condition to consider:

- Neveu-Schwarz sector: anti-periodic boundary condition $\xi(L + x) = -\xi(x)$;

- Ramond sector: periodic boundary condition $\xi(L + x) = \xi(x)$.

The choice of the vacuum sector for the ground state of the system at $t < 0$ depends on the sign of the mass parameter $m$, so below we discuss each possible case separately.

### 3.2.1    $m > 0$

Here the ground state of the system at $t < 0$ is two-fold degenerate:

$$|0\rangle_{m>0} = \frac{1}{\sqrt{2}} \left[ |0\rangle_{\text{NS}} \pm |0\rangle_{\text{R}} \right]. \tag{3.3}$$

Thus we have to consider both how the Neveu-Schwarz and Ramond vacua evolve under the double quench.

#### 3.2.1.1    Neveu-Schwarz Sector

The Neveu-Schwarz sector turns out to be relatively easier with which to deal. The eigenmode expansion of the Majorana fermion has the general form

$$\xi_R = \frac{1}{\sqrt{L}} \sum_{q>0} \left[ \left( \alpha_q(m)c_q(m) - \beta_q(m)c^{\dagger}_{-q}(m) \right) e^{iqx} + \text{h.c.} \right],$$

$$\xi_L = \frac{1}{\sqrt{L}} \sum_{q>0} \left[ \left( \alpha_q(m)c_{-q}(m) + \beta_q(m)c^{\dagger}_q(m) \right) e^{-iqx} + \text{h.c.} \right]. \tag{3.4}$$

where $\alpha_q(m)$ and $\beta_q(m)$ satisfy

$$|\alpha_q(m)|^2 + |\beta_q(m)|^2 = 1. \tag{3.5}$$

The fermionic operator, $c_q$, satisfies the anti-commutation relation $\{c_q, c^{\dagger}_{q'}\} = \delta_{qq'}$, the momenta are quantized as $q = (2\pi n + \pi)/L$, with $n = 0, 1, \ldots$, and $L = (N - 1)a_0$, and the band width $J$ provides an ultraviolet cutoff $q_{\max} = J/v$. The coefficients $\alpha_q(m), \beta_q(m)$ are chosen in a way such that the Hamiltonian in Eq. (3.2) is diagonalized:

$$\alpha_q(m) = \cos\left(\theta_q(m)\right), \quad \beta_q(m) = -i\sin\left(\theta_q(m)\right), \quad \theta_q(m) = \frac{1}{2}\arctan\frac{m}{vq}. \tag{3.6}$$

As a result, the diagonalized Hamiltonian takes the form

$$H = \sum_{q>0} \epsilon_q(m) \left[ c^{\dagger}_q(m)c_q(m) + c^{\dagger}_{-q}(m)c_{-q}(m) \right] - \sum_{q>0} \epsilon_q(m), \tag{3.7}$$

where $\epsilon_q(m) = \sqrt{v^2 q^2 + m^2}$. Using Eq. (3.4) and Eq. (3.6), we can relate the eigenmodes with different masses:

$$
\begin{aligned}
c_q(m) &= \cos\left(\Delta\theta_q\right) c_q(M_{\mathrm{pump}}) - i\sin\left(\Delta\theta_q\right) c_{-q}^\dagger(M_{\mathrm{pump}}), \\
c_{-q}(m) &= \cos\left(\Delta\theta_q\right) c_{-q}(M_{\mathrm{pump}}) + i\sin\left(\Delta\theta_q\right) c_q^\dagger(M_{\mathrm{pump}}), \\
c_q(M_{\mathrm{pump}}) &= \cos\left(-\Delta\theta_q\right) c_q(m) - i\sin\left(-\Delta\theta_q\right) c_{-q}^\dagger(m), \\
c_{-q}(M_{\mathrm{pump}}) &= \cos\left(-\Delta\theta_q\right) c_{-q}(m) + i\sin\left(-\Delta\theta_q\right) c_q^\dagger(m), \\
\Delta\theta_q &\equiv \theta_q(m) - \theta_q(M_{\mathrm{pump}}).
\end{aligned}
\tag{3.8}
$$

Consequently, the pre-quench ground state $|0\rangle_{\mathrm{NS}}$ at $t < 0$ can be expressed in terms of a product of coherent states with quenched mass $M_{\mathrm{pump}}$:

$$
c_q(m)\,|0\rangle_{\mathrm{NS}} = 0, \quad c_{-q}(m)\,|0\rangle_{\mathrm{NS}} = 0
$$
$$
\Rightarrow |0\rangle_{\mathrm{NS}} = \mathcal{N}^{-1/2} \exp\left[\sum_{q>0} i\tan\Delta\theta_q\; c_q^\dagger(M_{\mathrm{pump}}) c_{-q}^\dagger(M_{\mathrm{pump}})\right]|0\rangle_{\mathrm{pump}},
\tag{3.9}
$$

where $\mathcal{N} = \prod_{q>0}\left[1 + (\tan\Delta\theta_q)^2\right]$ is the normalization factor. In doing so, the evolution of the quantum state during time $0 < t < t_1$ can be trivially implemented. As for the post-quench quantum state at $t > t_1$, we can express $|0\rangle_{\mathrm{pump}}$ in terms of a product of coherent states with original mass $m$, again rendering the evolution straightforward. The complete expression for the time-evolving quantum state can be written as

$$
|\psi_{\mathrm{NS}}(t)\rangle = \begin{cases}
|0\rangle_{\mathrm{NS}} & t < 0 \\
\mathcal{N}^{-1/2} e^{\sum_{q>0} i\tan\Delta\theta_q e^{-i2\epsilon_q(M_{\mathrm{pump}})t} c_q^\dagger(M_{\mathrm{pump}}) c_{-q}^\dagger(M_{\mathrm{pump}})}\,|0\rangle_{\mathrm{pump}} & 0 < t < t_1 \;, \\
\mathcal{M}^{-1/2} e^{-\sum_{q>0} u_q c_q^\dagger(m) c_{-q}^\dagger(m)}\,|0\rangle_{\mathrm{NS}} & t > t_1
\end{cases}
$$
$$
u_q\left(m, M_{\mathrm{pump}}, t_1, t\right) = i\frac{e^{-2i\epsilon_q(m)(t-t_1)}\left(1 - e^{-2i\epsilon_q(M_{\mathrm{pump}})t_1}\right)}{\cot\Delta\theta_q + \tan\Delta\theta_q e^{-2i\epsilon_q(M_{\mathrm{pump}})t_1}}, \quad \mathcal{M} = \prod_{q>0}\left(1 + u_q^* u_q\right).
\tag{3.10}
$$

This explicit expression for the time-evolving quantum state will allow us to calculate the correlation functions within the Neveu-Schwarz sector as a function of time.

### 3.2.1.2 Ramond Sector

Now we turn to the relatively more complex situation in the Ramond sector, where the eigenmode expansion of the Majorana fermion is

$$
\begin{aligned}
\xi_R &= \frac{1}{\sqrt{L}}\sum_{q>0}\left[\left(\alpha_q(m) c_q(m) - \beta_q(m) c_{-q}^\dagger(m)\right) e^{iqx} + \mathrm{h.c.}\right] + \frac{1}{\sqrt{2L}}\left(c_0^\dagger + c_0\right), \\
\xi_L &= \frac{1}{\sqrt{L}}\sum_{q>0}\left[\left(\alpha_q(m) c_{-q}(m) + \beta_q(m) c_q^\dagger(m)\right) e^{-iqx} + \mathrm{h.c.}\right] - \frac{i}{\sqrt{2L}}\left(c_0^\dagger - c_0\right).
\end{aligned}
\tag{3.11}
$$

$\alpha_q(m)$ and $\beta_q(m)$ satisfy the same constraint as before while the momenta are quantized as $q = 2\pi n/L, n = 1, 2, \ldots,$. A complicating factor here is the presence of the $q = 0$ mode that makes the situation in the Ramond sector nontrivial [37,50]. Firstly, the diagonalized Hamiltonian takes the form

$$
H = \sum_{q>0}\epsilon_q(m)\left[c_q^\dagger(m) c_q(m) + c_{-q}^\dagger(m) c_{-q}(m)\right] - \sum_{q>0}\epsilon_q(m) + \frac{m}{2}\left(c_0^\dagger c_0 - c_0 c_0^\dagger\right).
\tag{3.12}
$$

According to Eq. (3.12), we can see that for $m > 0$ the $q = 0$ mode has positive energy, while for $m < 0$ we need to perform a particle-hole transformation $c_0 \leftrightarrow c_0^\dagger$ for the $q = 0$ mode to make its energy positive. Since the $q = 0$ mode does not mix with $q \neq 0$ modes with the current quench protocol, this only introduces a simple relation between the $q = 0$ modes with different masses:

$$c_0(m > 0) = c_0(M_{\text{pump}} > 0), \quad c_0(m > 0) = c_0^\dagger(M_{\text{pump}} < 0). \tag{3.13}$$

On the other hand, the relation between the $q \neq 0$ modes with different masses stays the same as that shown in Eq. 3.8. As a result, for the double quench from $m > 0$ to $M_{\text{pump}} > 0$ and back to $m > 0$, the evolution of the quantum state stays the same in expression as that shown in Eq. 3.10, we just need to change the subscript NS with R. In contrast, for the double quench from $m > 0$ to $M_{\text{pump}} < 0$ and back to $m > 0$, the evolution of the quantum state needs the following modification:

$$|\psi_{\text{R}}(t)\rangle = \begin{cases} |0\rangle_{\text{R}} & t < 0 \\ \mathcal{N}^{-1/2} e^{\sum_{q>0} i \tan \Delta\theta_q e^{-i2\epsilon_q(M_{\text{pump}})t} c_q^\dagger(M_{\text{pump}}) c_{-q}^\dagger(M_{\text{pump}})} \\ \qquad\qquad \times e^{-i|M_{\text{pump}}|t} c_0^\dagger(M_{\text{pump}}) |0\rangle_{\text{pump}} & 0 < t < t_1 \\ \mathcal{M}^{-1/2} e^{-\sum_{q>0} u_q c_q^\dagger(m) c_{-q}^\dagger(m)} e^{-i|M_{\text{pump}}|t_1} |0\rangle_{\text{R}} & t > t_1 \end{cases}, \tag{3.14}$$

where the normalization factor $\mathcal{N}, \mathcal{M}$ and the function $u_q(m, M_{\text{pump}}, t_1, t)$ are the same as those in Eq. (3.10).

### 3.2.2   $m < 0$

Here the ground state of the system at $t < 0$ is non-degenerate:

$$|0\rangle_{m<0} = |0\rangle_{\text{NS}}, \tag{3.15}$$

which involves only the Neveu-Schwarz sector. Consequently, the evolution of the quantum state is exactly the same as that shown in Sec. 3.2.1.1, with the explicit expression of the quantum state shown in Eq. 3.10.

## 4   Evolution of One-Body Operators

We will first study how one-body operators evolve under the current quench protocol. In particular we will study how the energy density, fermionic mode occupation, and the intersite antiferromagnetic correlations evolve under the double quench.

### 4.1   Evolution of energy density and fermionic mode occupation

As a first application, we investigate the energy pumped into the system by our quench protocol and the mode occupations populated by the pump. The energy pumped into the system is nothing but the difference in the expectation value of the Hamiltonian (3.7) before and after the quantum quench. We start with the pre-quench calculation, where we have Majorana masses $m_s, m_t$ and quantum state $|0\rangle = |0^0\rangle_{m_s} \otimes \prod_{i=1}^3 \otimes |0^i\rangle_{m_t}$:

$$E_i = \langle 0| \left( H_{m_s}^0 + \sum_{i=1}^3 H_{m_t}^i \right) |0\rangle = -\sum_{q>0} \left( \epsilon_q^0(m_s) + \sum_{i=1}^3 \epsilon_q^i(m_t) \right), \tag{4.1}$$

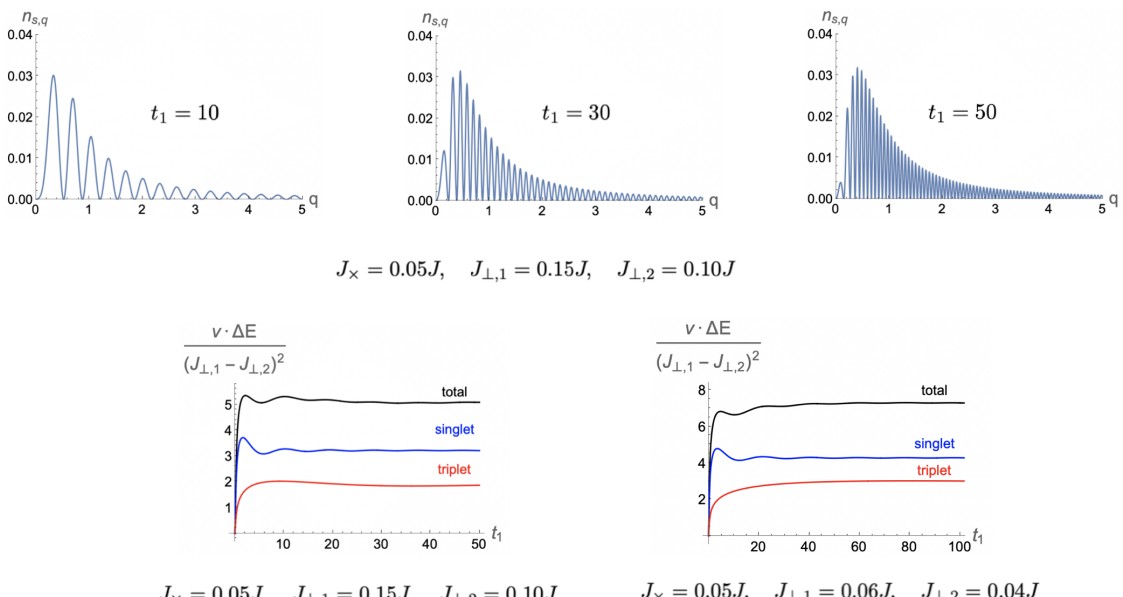

Figure 4.1: Upper panel: The mode occupations for the singlet modes with set #1 of Eq. (3.1) with different choices of $t_1$. Moving from left to right, $t_1$ increases and the density of modes occupied in the envelope increases. Lower panel: The energy pumped into the system per unit length with increasing $t_1$, where the left figure is for set #1 and right figure is for set #2 of Eq. (3.1). The contributions to $\Delta E$ from the singlet and triplet modes are shown separately such that the different time scales governing the modes can be seen clearly.

where the superscript number labels the species of the Majorana fermions. This expression for $E_i$ works for the situation where all Majoranas are sitting in the Neveu-Schwarz sector. If otherwise, we also need to include the term $q = 0$ for each species that is sitting in the Ramond sector. Next, we perform the post-quench calculation at $t > t_1$, where we still have Majorana masses $m_s, m_t$, but a different quantum state $|\psi(t)\rangle = \prod_{i=0}^{3} \otimes |\psi^i(t)\rangle$. For the Neveu-Schwarz sector, $|\psi^i(t)\rangle$ takes the form of Eq. (3.10) at $t > t_1$, while for the Ramond sector, we need to make the prescribed modifications. The final result is

$$\Delta E \equiv E_f - E_i = \langle\psi(t)|\left(H_{m_s}^0 + \sum_{i=1}^{3} H_{m_t}^i\right)|\psi(t)\rangle - E_i = \Delta E_s + \Delta E_t$$

$$\Delta E_s = \sum_{q>0} \frac{2\epsilon_q(m_s)u_q^{s*}u_q^s}{1 + u_q^{s*}u_q^s}, \quad \Delta E_t = \sum_{q>0} \frac{6\epsilon_q(m_t)u_q^{t*}u_q^t}{1 + u_q^{t*}u_q^t},$$

(4.2)

where $u_q^s \equiv u_q(m_s, M_{s,\text{pump}}, t_1, t), u_q^t \equiv u_q(m_t, M_{t,\text{pump}}, t_1, t)$. This expression for the pumped energy $\Delta E$ works for both the Neveu-Schwarz and the Ramond sector. The pumped energy $\Delta E$ is $t$-independent but $t_1$-dependent, as expected, since the $t$-dependence disappears in the combination $u_q^{s*}u_q^s$ and $u_q^{t*}u_q^t$, while the $t_1$-dependence remains. The fermionic mode occupations are also accessible as expectation values with respect to the post-quench state

$$n_q^a \equiv \langle\psi(t)|c_q^\dagger(m_a)c_q(m_a)|\psi(t)\rangle = \frac{u_q^{a*}u_q^a}{1 + u_q^{a*}u_q^a}, \quad t > t_1,$$

(4.3)

where the mode can be either the singlet ($a = s$) or triplet ($a = t$). Also we have $n_{-q}^a = n_q^a$ by inversion symmetry. The fermionic mode occupation is time-independent after the

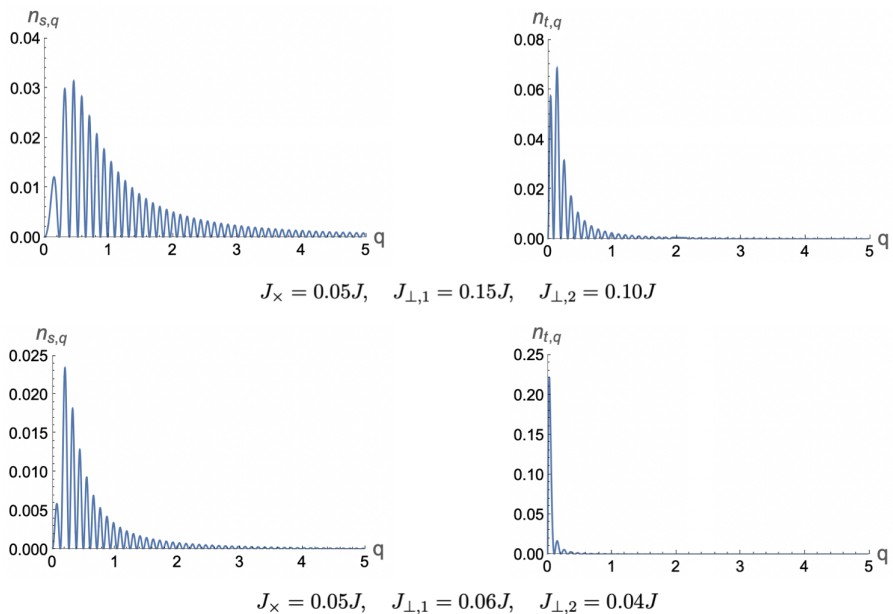

$$J_\times = 0.05J, \quad J_{\perp,1} = 0.15J, \quad J_{\perp,2} = 0.10J$$

$$J_\times = 0.05J, \quad J_{\perp,1} = 0.06J, \quad J_{\perp,2} = 0.04J$$

Figure 4.2: The mode occupations for the two sets of choice of parameters in Eq. (3.1). The upper panel is for set #1, and the lower panel is for set #2. The quench time is taken as $t_1 = 30$. The envelop has a width scaling as $\left|\frac{2mM_{\text{pump}}}{v(M_{\text{pump}}-m)}\right|$, and the oscillation below the envelop has a period scaling as $\frac{\pi}{vt_1}$. As a result, increasing $t_1$ will fill up the area below the envelop and there will be a certain saturation time such that further increasing of $t_1$ has negligible effect, see Fig. 4.1.

pump, as it should be, since we have non-interacting Majorana fermions.

The mode occupations corresponding to the two sets of choice of parameters in Eq. (3.1) are shown in Fig. 4.2, where we have three main features: (1) only the modes below a certain cutoff are significantly populated (correspondingly, in the following numerical calculations we will choose the practical cutoff of the momentum as $q_{\text{max}} = 5J/v$); (2) the mode occupations present oscillations in $q$; (3) the singlet modes have a larger width compared with the triplet modes. These features can be easily extracted from an approximation of Eq. (4.3) applicable to the parameters chosen here ($a = s, t$):

$$n_q^a \sim \frac{2 - 2\cos\left[2\epsilon_q(M_{a,\text{pump}})t_1\right]}{\cot^2\Delta\theta_q^a + \tan^2\Delta\theta_q^a}, \quad \Delta\theta_q^a = \frac{1}{2}\left(\arctan\frac{m_a}{vq} - \arctan\frac{M_{a,\text{pump}}}{vq}\right). \quad (4.4)$$

It is easy to see that the numerator determines the oscillating behavior, while the denominator determines the envelope. As a result, the oscillations in the mode occupation have a period scaling as $\frac{\pi}{vt_1}$, the envelope has a width scaling as $\left|\frac{2mM_{\text{pump}}}{v(M_{\text{pump}}-m)}\right|$ (which can be seen from the large $q$ behavior of the denominator), and at large $q$ the mode occupation falls off as $1/q^2$. Given the masses as determined in Eq. (2.8), it is easy to see that the width of the singlet mode occupations is always larger than that of the triplet mode occupations.

The energy pumped into the system corresponding to the two sets of choice of parameters in Eq. (3.1) is shown in Fig. 4.1, where we vary the pump time $t_1$. Since the envelop of the mode occupations is fixed by the denominator of Eq. (4.4), increasing the pumping time $t_1$ only fills in the area below the envelope more densely. As a result, the energy pumped into the system first experiences an increase with increasing $t_1$ and then sees a decaying weak oscillation, as shown in Fig. 4.1. According to the analysis in Appendix A,

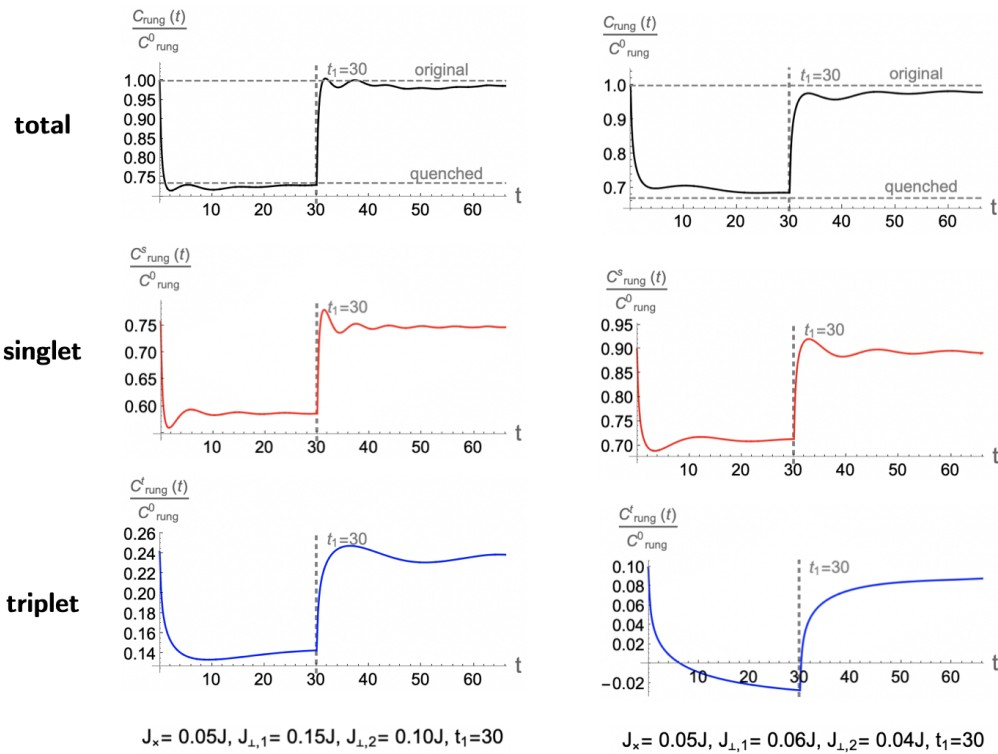

Figure 4.3: The evolution of the local antiferromagnetic correlations across the rung $C_{\text{rung}}(t)$ for the two sets of parameters chosen in Eq. (3.1), each normalized by their corresponding original, prepump value $C^0_{\text{rung}}(t)$. A dashed grey line with value 1 marks this original value in the topmost panel. In this panel we also show the equilibrium value of the rung correlations corresponding to the pumped value of the masses, $M_{\text{pump}}$ (but still normalized by that with respect to the prepump masses, $m$). The lower two panels show the contributions to the correlations from the singlet and triplet separately.

the energy pumped into the system is perturbative in the change of the parameters:

$$\Delta E_s/L \propto (m_s - M_{s,\text{pump}})^2 \propto (J_{\perp,1} - J_{\perp,2})^2,$$
$$\Delta E_t/L \propto (m_t - M_{t,\text{pump}})^2 \propto (J_{\perp,1} - J_{\perp,2})^2, \tag{4.5}$$

and the time scale for the initial growth can be estimated for the singlet and triplet sectors to be

$$t_{s,\text{saturation}} \sim \frac{1}{2\sqrt{|M_{s,\text{pump}}|(|m_s| + |M_{s,\text{pump}}|)}},$$
$$t_{t,\text{saturation}} \sim \frac{1}{2\sqrt{|M_{t,\text{pump}}|(|m_t| + |M_{t,\text{pump}}|)}}. \tag{4.6}$$

The subsequent weak oscillation decays algebraically with time constants $2\pi t_{s,\text{saturation}}$ and $2\pi t_{t,\text{saturation}}$. Consequently, as long as the quench time $t_1$ is tuned to be larger than the saturation time, the variation of $t_1$ is expected to have negligible effect on experimental observables.

## 4.2 Local Antiferromagnetic Correlations

In this subsection we investigate how the local spin-spin correlations evolve after the double quench, both across the rung, expressible as

$$C_{\text{rung}} \equiv \mathbf{S}_{1,r} \cdot \mathbf{S}_{2,r}, \tag{4.7}$$

and between two sites on the same leg, expressible as

$$C_{\text{leg}} \equiv \mathbf{S}_{1,r} \cdot \mathbf{S}_{1,r+1}. \tag{4.8}$$

Both observables should not depend on the lattice site due to the translational invariance of the system. From the bosonization procedure discussed in Sec. 2.2, we obtain that the more relevant part of the smooth component of the spin-spin interaction across the rung is given by

$$
\begin{aligned}
\mathbf{S}_{1,r} \cdot \mathbf{S}_{2,r} &\sim 2\cos\left(\Theta_-(x_r)\right) - \cos\left(\Phi_+(x_r)\right) + \cos\left(\Phi_-(x_r)\right) \\
&\sim 3i\xi_R^0\xi_L^0 - i\left(\boldsymbol{\xi}_R \cdot \boldsymbol{\xi}_L\right) \sim H_{\text{mass}},
\end{aligned} \tag{4.9}
$$

and the local spin-spin correlation along the leg is given by

$$
\begin{aligned}
\mathbf{S}_{1,r} \cdot \mathbf{S}_{1,r+1} &\sim H_{\text{kin}}(x_r) + c(-1)^r \cos\left(\frac{\Phi_+(x_r) + \Phi_-(x_r)}{2}\right), \\
H_{\text{kin}} &\sim i\left(\xi_R^0\partial_x\xi_R^0 - \xi_L^0\partial_x\xi_L^0 + \boldsymbol{\xi}_R\partial_x\boldsymbol{\xi}_R - \boldsymbol{\xi}_L\partial_x\boldsymbol{\xi}_L\right), \\
\cos\left(\frac{\Phi_+ + \Phi_-}{2}\right) &\sim \mu^0\mu^1\sigma^2\sigma^3 - \sigma^0\sigma^1\mu^2\mu^3,
\end{aligned} \tag{4.10}
$$

where $c$ is some constant. Since each term of the expression for the staggered part of $\mathbf{S}_{1,r} \cdot \mathbf{S}_{1,r+1}$ involves at the same time both the Ising spin and disorder operators of the Ising model with the triplet mass, its expectation value simply vanishes, leaving only the smooth part of $\mathbf{S}_{1,r} \cdot \mathbf{S}_{1,r+1}$. As a result, we can focus solely on how $H_{\text{mass}}$ and $H_{\text{kin}}$ evolve in order to understand the local antiferromagnetic fluctuations induced by the double quench.

Using Eq. (3.4) and (3.10), we obtain the following results for each species of the Majorana fermions sitting in the Neveu-Schwarz sector and $t > t_1$:

$$
\begin{aligned}
&\langle\psi_{\text{NS}}(t)|\, i\xi_R\xi_L\,|\psi_{\text{NS}}(t)\rangle \\
&= \frac{i}{L}\sum_{q>0}\frac{1}{1+u_q^*u_q}\left[(2\alpha_q\beta_q)\left(1-u_q^*u_q\right) + (\alpha_q^2+\beta_q^2)\left(u_q - u_q^*\right)\right], \\
&\langle\psi_{\text{NS}}(t)|\, i\left(\xi_R\partial_x\xi_R - \xi_L\partial_x\xi_L\right)|\psi_{\text{NS}}(t)\rangle \\
&= \frac{1}{L}\sum_{q>0}\frac{1}{1+u_q^*u_q}q\left[2(\alpha_q^2+\beta_q^2)\left(1-u_q^*u_q\right) + 4\alpha_q\beta_q\left(u_q - u_q^*\right)\right],
\end{aligned} \tag{4.11}
$$

where the mass parameter for the functions $\alpha_q, \beta_q$ is set to the pre-pump value, $m$. Analogously, using Eq. (3.11) and (3.14), we obtain the following results for $t > t_1$ for each species of Majorana fermions sitting in the Ramond sector:

$$
\begin{aligned}
&\langle\psi_{\text{R}}(t)|\, i\xi_R\xi_L\,|\psi_{\text{R}}(t)\rangle \\
&= \frac{i}{L}\sum_{q>0}\frac{1}{1+u_q^*u_q}\left[(2\alpha_q\beta_q)\left(1-u_q^*u_q\right) + (\alpha_q^2+\beta_q^2)\left(u_q - u_q^*\right)\right] + \frac{1}{2L}, \\
&\langle\psi_{\text{R}}(t)|\, i\left(\xi_R\partial_x\xi_R - \xi_L\partial_x\xi_L\right)|\psi_{\text{R}}(t)\rangle \\
&= \frac{1}{L}\sum_{q>0}\frac{1}{1+u_q^*u_q}q\left[2(\alpha_q^2+\beta_q^2)\left(1-u_q^*u_q\right) + 4\alpha_q\beta_q\left(u_q - u_q^*\right)\right],
\end{aligned} \tag{4.12}
$$

where the mass parameter for the functions $\alpha_q, \beta_q$ is again equal to the pre-pump value $m$. In the thermodynamic limit $L \to \infty$, the difference between Eq. (4.11) and (4.12) vanishes as $1/L$. The corresponding results for the unpumped system is obtained by taking the limit $u_q \to 0$ in Eq. (4.11) and (4.12).

Using Eq. (4.11) and (4.12), both $C_{\text{rung}} \equiv \mathbf{S}_{1,r} \cdot \mathbf{S}_{2,r}$ and $C_{\text{leg}} \equiv \mathbf{S}_{1,r} \cdot \mathbf{S}_{1,r+1}$ can be calculated explicitly. We plot in Fig. 4.3 and 4.4 the evolution of $C_{\text{rung}}$ and $C_{\text{leg}}$ for the two sets of parameters given in Eq. (3.1), each normalized by their corresponding prepump value. Similar to Fig. 4.1, after the pump is turned off and on, the evolution of the local antiferromagnetic correlations first experience a rapid relaxation and then performs a decaying weak oscillation. According to the analysis in Appendix A, the time scales for the rapid relaxation are

$$
\begin{aligned}
t_{\text{during pump}} &\sim \frac{1}{2\sqrt{|M_{\text{pump}}|(|m| + |M_{\text{pump}}|)|}}, \\
t_{\text{after pump}} &\sim \frac{1}{2\sqrt{|m|(|m| + |M_{\text{pump}}|)|}},
\end{aligned}
\tag{4.13}
$$

and the subsequent oscillations decay algebraically with time constant $2\pi t_{\text{during pump}}$ and $2\pi t_{\text{after pump}}$. On the other hand, as shown in Appendix A, the periods of these oscillations are determined solely by the unquenched mass:

$$
t_{\text{oscillation}} \sim \pi/|m|. \tag{4.14}
$$

We can see that $t_{\text{during pump}}$ is the same as the saturation time in Eq. (4.6), which is expected, and $t_{\text{after pump}}$ does not depend on the quench time $t_1$. According to Eq. (3.1), the singlet mass is bigger than the triplet mass, therefore the decaying oscillations due to the singlet have smaller periods as compared with those due to the triplet. This feature can be clearly seen in Fig. 4.3 and 4.4.

As can be seen from Fig. 4.3 and 4.4, both the rung and leg correlations are weakened after the pump is turned off. This is due to the fact that the pump creates excitations that disturb the ground state correlations. The extent to which the ground state correlations are disturbed can be roughly estimated by the change in mass scales. In fact, the correlation across the rung has a dimensionality of mass, while the correlation along the leg has a dimensionality of mass squared. As a result, the former is more disturbed, as can be seen from comparing Fig. 4.3 with 4.4.

For comparison, we shrank the pump time from $t_1 = 30 > t_{\text{during pump}}$ to $t_1 = 0.5 < t_{\text{during pump}}$ and plotted the evolution of the local antiferromagnetic correlations in Fig. 4.5. We can see that if the quench is short enough such that $t_1 < t_{\text{during pump}}$, the post-pump evolution returns closer to its prepump value and presents weaker subsequent oscillations.

## 5   Two-Time Non-equilibrium Correlations

We are interested in this work to connect to response as measured in a pump-probe time-resolved resonant inelastic x-ray scattering (tr-RIXS) experiment. The general tr-RIXS response is derived in [51]:

$$
\begin{aligned}
I(\omega_i, \omega_f, q, t) = &\int_{-\infty}^{\infty} \mathrm{d}t_2 \int_{-\infty}^{t_2} \mathrm{d}t_1 \int_{-\infty}^{\infty} \mathrm{d}t_2' \int_{-\infty}^{t_2'} \mathrm{d}t_1' e^{i\omega_i(t_1'-t_1)} e^{-i\omega_f(t_2'-t_2)} l(t_1, t_2) l(t_1', t_2') \\
&\times g(t_1, t) g(t_2, t) g(t_1', t) g(t_2', t) \sum_{r,r'} e^{iq(x_r - x_{r'})} G_{\boldsymbol{\epsilon}_i, \boldsymbol{\epsilon}_f}^{rr'}(t_1, t_2, t_2', t_1'),
\end{aligned}
\tag{5.1}
$$

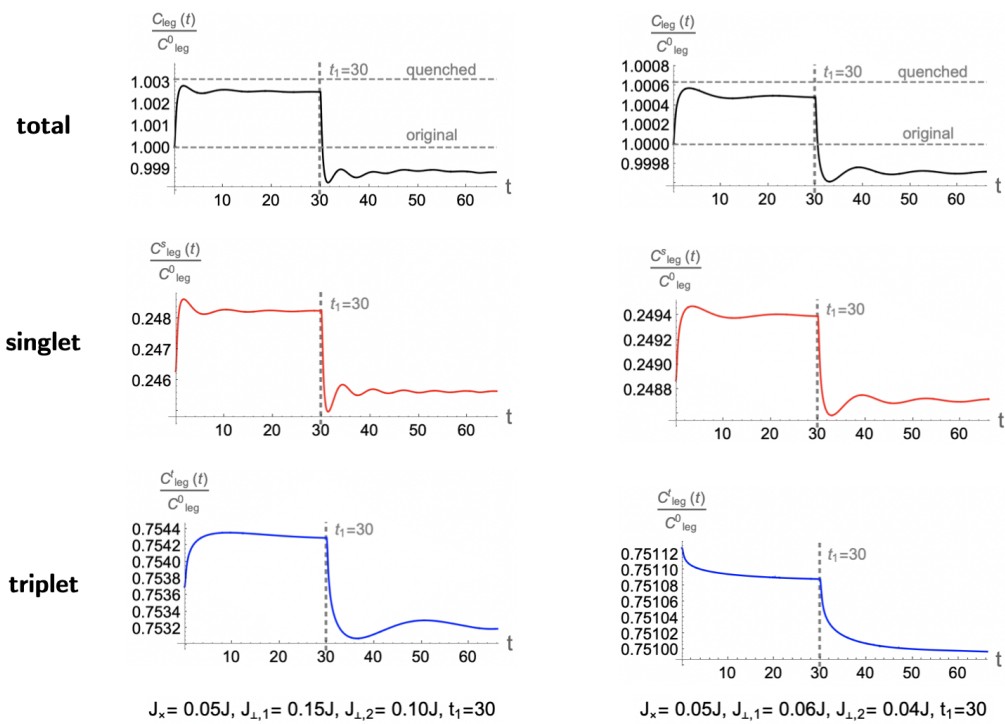

Figure 4.4: The evolution of the local antiferromagnetic correlations along the leg $C_{\text{leg}}(t)$ for the two sets of parameters given in Eq. (3.1), each normalized by their corresponding prepump value, $C^0_{\text{leg}}$. In the topmost panel, the equilibrium value of $C_{\text{leg}}(M_{\text{pump}})$ (but normalized by that with respect to $m$) is also shown as dashed grey lines at values slightly greater than 1. We can see that the quenched value of $C_{\text{leg}}(M_{\text{pump}})$ exceeds its prepump value unlike that of the rung correlations. The lower two panels show the contributions to the leg correlations from the singlet and triplet separately.

where $\omega_i$ and $\omega_f$ are the energies of the incoming and outgoing photons and $\boldsymbol{\epsilon}_i$ and $\boldsymbol{\epsilon}_f$ are their polarizations. The correlator $G^{rr'}_{\boldsymbol{\epsilon}_i,\boldsymbol{\epsilon}_f}$ is given by

$$
\begin{aligned}
G^{rr'}_{\boldsymbol{\epsilon}_i,\boldsymbol{\epsilon}_f}\left(t_1,t_2,t_2',t_1'\right) = \Big\langle & U\left(-\infty,t_1'\right)D^\dagger_{r'\boldsymbol{\epsilon}_i}\left(t_1'\right)U\left(t_1',t_2'\right)D_{r'\boldsymbol{\epsilon}_f}\left(t_2'\right) \\
&\times U\left(t_2',t_2\right)D^\dagger_{r\boldsymbol{\epsilon}_f}\left(t_2\right)U\left(t_2,t_1\right)D_{r\boldsymbol{\epsilon}_i}\left(t_1\right)U\left(t_1,-\infty\right)\Big\rangle.
\end{aligned}
\tag{5.2}
$$

Here $U\left(t_1,t_2\right) = \mathcal{T}\exp\left(-i\int_{t_2}^{t_1}H(t)\right)$ is the time-ordered evolution operator, $l(t_1,t_2) = \exp\left(-|t_2-t_1|/\tau_{ch}\right)$ is a phenomenological factor that takes into account the finite lifetime $\tau_{ch}$ of the core hole, and $g\left(t_1,t\right)$ is a Gaussian in $t_1$ centered at the measurement time $t$:

$$
g(t_1,t) = \frac{1}{\sqrt{2\pi}\tau^{1/4}}e^{-\frac{1}{2}\left(\frac{t_1-t}{\tau}\right)^2},
\tag{5.3}
$$

with $\tau$ being the time scale over which the signal is experimentally acquired, and $g(t_1,t)$ is defined in a way such that in the limit of short core hole lifetime, the resulting response is proportional to $\tau$, apart from another exponential dependence in the form of $\exp\left(-\tau^2\tilde{\omega}^2/2\right)$, where $\tilde{\omega}$ has the unit of energy. Under such a choice, we can take the limit $\tau \to \infty$ and the two factors combine to yield a $\delta-$function:

$$
\lim_{\tau\to\infty}\tau e^{-\frac{\tau^2\tilde{\omega}^2}{2}} = \sqrt{2\pi}\delta(\tilde{\omega})
\tag{5.4}
$$

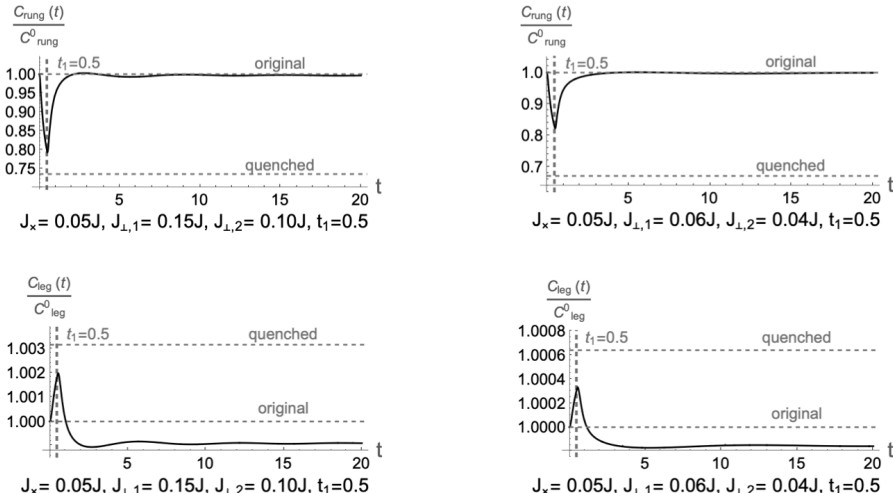

Figure 4.5: The evolution of local antiferromagnetic correlations across the rung (upper panel) and along the leg (lower panel) for the two sets of parameters chosen in Eq. (3.1) but with a shorter pump time, $t_1 = 0.5$.

This will be utilized to derive our results in the limit $\tau \to \infty$.

The correlation function, $G^{rr'}_{\epsilon_i \epsilon_f}$, involves, $D_{r\epsilon}$, a dipole operator that creates a core hole at site $r$ and a valence electron through absorption of light with polarization $\epsilon$. Under certain circumstances this correlation function reduces to one that probes non-equilibrium spin dynamics, as say in certain iridate materials [7]. In such cases, assuming an ultra-short core hole lifetime limit, we can write the dipole operators in terms of an effective pseudo-spin operator $\boldsymbol{S}$:

$$l\left(t'_1, t'_2\right) D^{\dagger}_{r'\epsilon_i}\left(t'_1\right) U\left(t'_1, t'_2\right) D_{r'\epsilon_f}\left(t'_2\right) = \tau_{ch}\delta\left(t'_1 - t'_2\right)\left(\epsilon_f \cdot \epsilon_i + i\left(\epsilon_f \times \epsilon_i\right) \cdot \boldsymbol{S}_{r'}\right). \quad (5.5)$$

If we focus on an incoming and outgoing polarization such that $\epsilon_f \times \epsilon_i \propto \hat{\boldsymbol{z}}$, we obtain for the tr-RIXS response for the smooth components, $M^{\pm}$ of the spin operator, omitting the trivial factor $\tau_{ch}^2$:

$$I^{M_\pm}\left(\omega, q, t\right) = \iint_{-\infty}^{\infty} dt_2 dt'_2\, e^{i\omega\left(t'_2 - t_2\right)} g\left(t_2, t\right)^2 g\left(t'_2, t\right)^2 \sum_{r,r'} e^{iq\left(x_r - x_{r'}\right)} G^{rr'}_{M_\pm}\left(t_2, t'_2\right)$$

$$G^{rr'}_{M_\pm}(t_2, t'_2) = \left\langle \psi\left(t_2\right)\left| U\left(t_2, t'_2\right) M_{\pm,r'} U\left(t'_2, t_2\right) M_{\pm,r}\right| \psi\left(t_2\right)\right\rangle \quad (5.6)$$

$\omega \equiv \omega_i - \omega_f$, $|\psi(t_2)\rangle$ is given by Eq. (3.10), or Eq. (3.14), or a linear combination of them. An analogous expression holds for the staggered responses, $I^{N_\pm}\left(\omega, q, t\right)$, with $N_\pm$ replacing $M_\pm$ in the above.

We will suppose that the time of observation $t$ is long past the time $t_1$ during which the pump acts. Doing so, we can simplify the evolution operator to

$$U(t'_2, t_2) = e^{-i(t'_2 - t_2)H(m_s, m_t)}. \quad (5.7)$$

In other words, the effect of the pumping is only included in the state $|\psi(t_2)\rangle$. Since the system is translational invariant, Eq. (5.6) can be simplified to

$$I^{M_\pm}\left(\omega, q, t\right) = \int_{-\infty}^{\infty} dt_2 \int_{-\infty}^{\infty} dt'_2\, e^{i\omega\left(t'_2 - t_2\right)} g\left(t_2, t\right)^2 g\left(t'_2, t\right)^2 G_{M_\pm, q}\left(t_2, t'_2\right),$$

$$G_{M_\pm, q}(t_2, t'_2) = \left\langle \psi\left(t_2\right)\left| U\left(t_2, t'_2\right) M_{\pm, q} U\left(t'_2, t_2\right) M_{\pm, -q}\right| \psi\left(t_2\right)\right\rangle. \quad (5.8)$$

The calculation of Eq. (5.8) for the smooth part of the spin operators is straightforward, since they can be expressed as Majorana fermion bilinears, see Eq. (2.11). On the other hand, the calculation of Eq. (5.8) for the staggered part of the spin operators is more involved as it involves the Ising spin and disorder operators which are non-local relative to the fermions. Nonetheless, we will follow the strategy outlined in Sec. 2.4.4. Each of $I^{M\pm}$ and $I^{N\pm}$ consists of two parts, the steady state part, $I_1^{M\pm}, I_1^{N\pm}$, and the transient part, $I_2^{M\pm}, I_2^{N\pm}$, which we define as

$$
\begin{aligned}
I_1^{M\pm}(\omega, q) &\equiv \lim_{t\to\infty} I_1^{M\pm}(\omega, q, t), \quad I_2^{M\pm}(\omega, q, t) \equiv I^{M\pm}(\omega, q, t) - I_1^{M\pm}(\omega, q); \\
I_1^{N\pm}(\omega, q) &\equiv \lim_{t\to\infty} I_1^{N\pm}(\omega, q, t), \quad I_2^{N\pm}(\omega, q, t) \equiv I^{N\pm}(\omega, q, t) - I_1^{N\pm}(\omega, q).
\end{aligned}
\tag{5.9}
$$

In the following, we will first summarize the main results, and then discuss the details of the underpinning calculations in Secs. 5.3 and 5.2.

## 5.1 Summary of the Main Results

In this section, we present a summary of the main results for the tr-RIXS responses $I^{N\pm}(\omega, q, t)$ (staggered correlations) and $I^{M\pm}(\omega, q, t)$ (smooth correlations), and postpone the calculational details to later sections. We will discuss the steady state and transients part for each tr-RIXS response separately, since they present different interesting properties. We will factor out trivial factors that depend on the choice model/pump specified parameters, and focus on universal features. When it is possible, we will take certain limits such that these universal features are most salient and unambiguous, or have analytically tractable expressions. In this way, we show the extent to which we have analytical control over the observables under a plethora of tuning knobs, thus providing guidelines to understand and interpret the time-resolved response.

The steady state part of the tr-RIXS response is defined in the limit $t \to \infty$. To obtain expressions here we will further take the infinite measurement time limit $\tau \to \infty$ to make the salient features sharply defined in the frequency domain. On the other hand, the transient part of the tr-RIXS response requires a balance between frequency resolution and time resolution and so the experimentally specified value of $\tau$ will be kept finite in our computations. Here we will separate out a trivial factor depending on $\tau$ in order to isolation the remaining non-trivial functional dependence on this parameter. For numerical computations of the steady state and transient parts, we will use the two sets of parameters specified in Eq. (3.1), and choose $t_1 = 30$. Here $t_1$ is chosen larger than the saturation time and so we will not see effects due to the system having insufficient time to respond to the pump.

Before presenting the main results, we would like to discuss the relative importance of different tr-RIXS responses defined in Eq. (5.6). For this purpose, we compare their steady state pieces. The full analytical expressions for these are complicated and the frequencies at which they are nonzero differ (see Sec. 5.2 and 5.3). But as can be seen later from Fig. 5.1 and 5.4, all these steady state parts have significant spectral weight near certain thresholds. As a rough comparison of their relative importance, it is then useful to look at their asymptotic behaviors near these threshold frequencies:

$$
I_1^{M+}(\omega \sim 2|m_t|, q \to 0) \sim I^{M+} \frac{\Theta(\omega - 2|m_t|)}{\sqrt{[\omega/(2|m_t|)]^2 - 1}},
$$

$$
I^{M+} = \frac{1}{128\pi^2} \frac{vq^2}{|m_t|^2};
$$

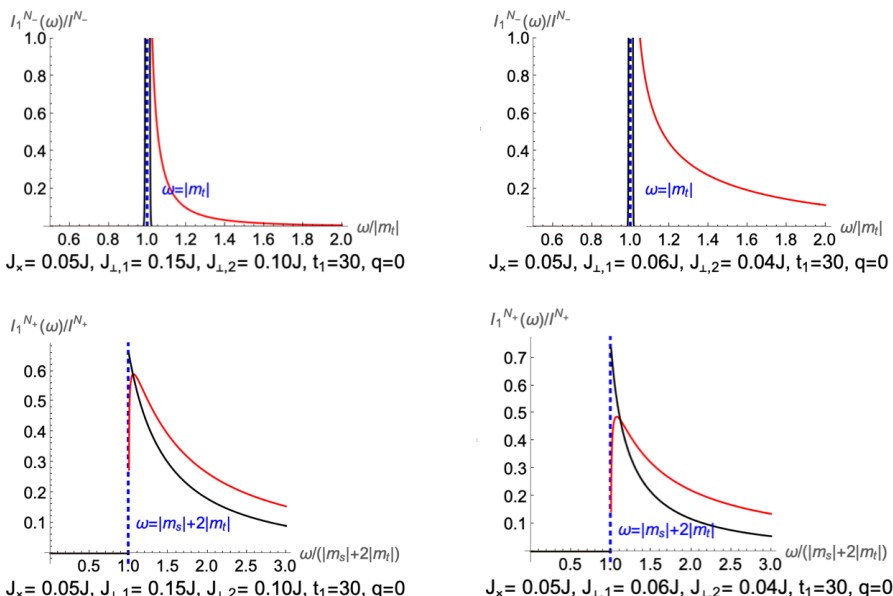

Figure 5.1: The steady state part of the staggered correlations in the limit $\tau \to \infty$, where the signals are normalized by the amplitudes shown in Eq. (5.10). The equilibrium (no pump) signal is shown in black, while the non-equilibrium response to the pump is shown in red. The upper panel shows the response for $N_-$ mode, where the equilibrium signal is a delta-function peak and the pumped response is a singular single-particle threshold. The lower panel gives the response for $N_+$, where the unpumped signal is a non-singular three-particle threshold and the pumped response sees a suppression near the threshold and enhancement along the tail.

$$I_1^{M_-}(\omega \sim |m_s| + |m_t|, q = 0) \sim I^{M_-} \frac{\Theta(\omega - |m_s| - |m_t|)}{\sqrt{[\omega/(|m_s| + |m_t|)]^2 - 1}},$$

$$I^{M_-} = \frac{1}{16\pi^2} \frac{\sqrt{|m_s m_t|}}{v(|m_s| + |m_t|)};$$

$$I_1^{N_+}(\omega \sim |m_s| + 2|m_t|, q = 0) \sim I^{N_+} f(\omega)\Theta(\omega - |m_s| - 2|m_t|),$$

$$I^{N_+} = \frac{1.3578^8 \lambda^2}{a_0^2} \frac{2}{(2\pi)^3} \frac{v}{|m_s m_t|};$$

$$f(\omega) = \sqrt{\frac{|m_s|}{|m_s| + 2|m_t|}} \left(1 - \frac{\Delta}{\pi\sqrt{|m_t|\delta}} + \frac{1}{2\pi} \sqrt{\frac{|m_s| + 2|m_t|}{|m_s|}} \frac{|m_s| + |m_t|}{|m_s| + 2|m_t|} \frac{\Delta}{\sqrt{|m_s|\delta}}\right),$$

$$\Delta \ll \delta = \omega - |m_s| - 2|m_t|;$$

$$I_1^{N_-}(\omega \sim |m_t|, q = 0) \sim I^{N_-} \frac{\Theta(\omega - |m_t|)}{\sqrt{(\omega/|m_t|)^2 - 1}},$$

$$I^{N_-} = \frac{1.3578^8 \lambda^2}{a_0^2} \frac{1}{4\pi} \frac{v}{|m_t|\Delta}, \tag{5.10}$$

where $\lambda$ is a constant on the order of one, $a_0$ is the lattice constant and $\Delta$ is proportional to

$$\Delta \propto (m_s - M_{s,\text{pump}})^2 / |m_s| + 3(m_t - M_{t,\text{pump}})^2 / |m_t| \tag{5.11}$$

which is an energy parameter that characterizes the strength of the pump, and will be

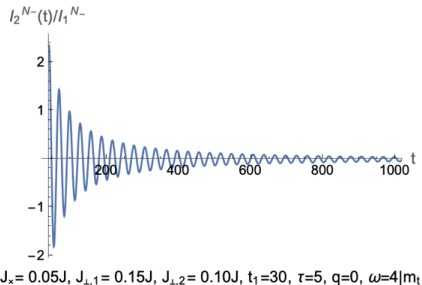
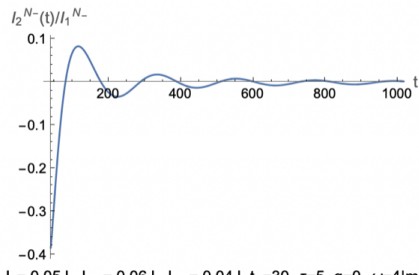

$J_x = 0.05J$, $J_{\perp,1} = 0.15J$, $J_{\perp,2} = 0.10J$, $t_1 = 30$, $\tau = 5$, $q = 0$, $\omega = 4|m_t|$      $J_x = 0.05J$, $J_{\perp,1} = 0.06J$, $J_{\perp,2} = 0.04J$, $t_1 = 30$, $\tau = 5$, $q = 0$, $\omega = 4|m_t|$

Figure 5.2: The transient part for $t > 40$ of the staggered correlations at $(\pi, \pi)$ with $\tau = 5$. We normalize the transient signal by the corresponding steady state part, $I_1^{N_-}$. We can see that the time evolution is underdamped at large $t$.

defined later in Sec. 5.2.3. $I_1^{M_+}(\omega, q), I_1^{M_-}(\omega, q), I_1^{N_-}\omega, q)$ all present divergence near the thresholds in the frequency domain, while $I_1^{N_+}(\omega, q)$ is always finite. Since $a_0 \sim v/J \ll v/|m_s|, v/|m_t|$, we conclude that $I_1^{N_-}(\omega, q)$ will be the dominant signal, and in the smooth sector, $I_1^{M_-}(\omega, q)$ will dominate over $I_1^{M_+}(\omega, q)$ for $q$ small.

### 5.1.1 Staggered Correlations

We first present the main results for the dominant staggered correlations, $I^{N_\pm}(\omega, q, t)$. For the steady state part, we take the measurement time to $\infty$, i.e., $\tau \to \infty$. Representative results are then shown in Fig. 5.1. We compare these non-equilibrium response functions against those that are obtained in equilibrium. We can see that for the $N_-$ mode, the pump changes the delta-function peak at $\omega = |m_t|$ to a square-root singularity at $\omega = |m_t|$. On the other hand, the $N_+$ mode sees a non-singular three-particle threshold at $\omega = |m_s| + 2|m_t|$. Here the pump suppresses the signal near the threshold frequency while enhancing the signal along the tail. Detailed analytical expressions for these two quantities will be derived later and can be found in Eq. (5.40) and (5.42) of Sec. 5.2.

The features in Fig. 5.1 can be understood qualitatively as follows. For response near $(\pi, \pi)$, we have $N_- \sim \sigma^0 \mu^1 \mu^2 \sigma^3$. And for the set of parameters in Eq. (3.1), we have $m_s > 0, m_t < 0$ such that

$$\langle \sigma^0 \rangle \sim \langle \mu^1 \rangle \sim \langle \mu^2 \rangle \sim \text{const.} \Rightarrow N_- \sim \sigma^3. \tag{5.12}$$

As a result, at leading order, the spectral properties near $(\pi, \pi)$ are determined by a single disordered Ising model. The pump disturbs the ground state in such a way that the coherent mode contribution to the equilibrium is broadened into a square-root singularity.

The response near $(\pi, 0)$ is governed by the $N_+ N_+$ correlator. Here we have $N_+ \sim \mu^0 \sigma^1 \sigma^2 \mu^3$. Under the condition $m_s > 0, m_t < 0$, we then have

$$\langle \mu^3 \rangle \sim \text{const.} \Rightarrow N_+ \sim \mu^0 \sigma^1 \sigma^2. \tag{5.13}$$

As a result, at leading order, the spectral properties of the $N_+$ mode is determined by convolution of three disordered Ising models. In equilibrium, this convolution mixes and smooths the coherent peak that would be present in any one of the Ising models into a non-singular threshold. The pump then further suppresses the spectral weight near threshold by disturbing the ground state.

We now turn to the transient part of the response near $(\pi, \pi)$. We do not plot the transient part at $(\pi, 0)$ as it is relatively small, something that we will demonstrate in Sec. 5.2.3. For the steady state part, we take the limit $\tau \to \infty$ and Eq. (5.4) indicates that the

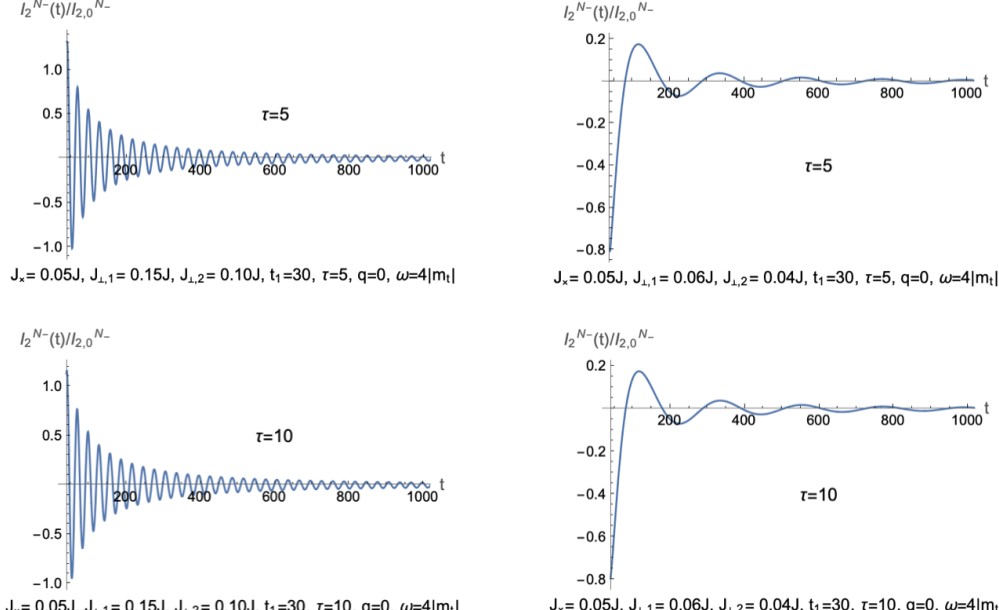

Figure 5.3: The transient part of the staggered correlations at $(\pi, \pi)$ with different choices of $\tau$, starting from $t = 40$. Here the signals are normalized by $\tau$-dependent factor $I_{2,0}^{N_-}$ derived in Eq. (5.44) of Sec. 5.2. We can see that the normalized signals are (approximately) independent of the choice of $\tau$.

final result is $\tau$-independent. For the transient part, we have to keep $\tau$ finite, and while the result is then $\tau$-dependent, we can separate out the $\tau$-dependent into a prefactor. To demonstrate this point, we first take $\tau = 5$ and show representative results in Fig. 5.2 where we normalize by the corresponding steady state part, $I_1^{N_-}$. Then we show a comparison between different choices of $\tau$ (specifically $\tau = 5$ and $\tau = 10$) in Fig. 5.3, where we instead normalize the results by $I_{2,0}^{N_-}$ a prefactor depending on $\tau$ (whose expression will be derived later in Eq. (5.44) of Sec. 5.2). After this normalization the results share certain common features independent of $\tau$. We can see that the transient signal first experiences a rapid decay on a time scale $\sim \pi/|m_t|$ and then performs underdamped oscillations afterwards. Later in Sec. 5.2.3 we will argue that the scaling of the time constants, $\tau_o$ for the oscillation and $\tau_r$ for the decay, can be estimated to be

$$\tau_o \sim \frac{\pi}{|m_t|}, \quad \tau_r \sim \frac{4\pi}{\sqrt{\Delta^2 + |m_t M_{t,\text{pump}}| + m_t^2/2}}, \tag{5.14}$$

where it is assumed that the measurement time $\tau < \tau_r$, and $\Delta \propto (m_s - M_{s,\text{pump}})^2/|m_s| + 3(m_t - M_{t,\text{pump}})^2/|m_t|$ is an energy parameter that characterizes the strength of the pump that is defined later in Sec. 5.2.3.

### 5.1.2   Smooth Correlations

In this section, we present the main results for the smooth correlations, $I^{M_\pm}(\omega, q, t)$. For the steady state part, we again take the infinite measurement time limit, $\tau \to \infty$. Representative results are shown in Fig. 5.4. We can see that the pump transfers spectral weight from the singular two-particle threshold to a region around $\omega = 0$.

The features in Fig. 5.4 can be understood qualitatively as follows. The weight near $(\pi, 0)$ is described by the $M_-$ correlator. $M_-$ mixes species in the singlet and triplet

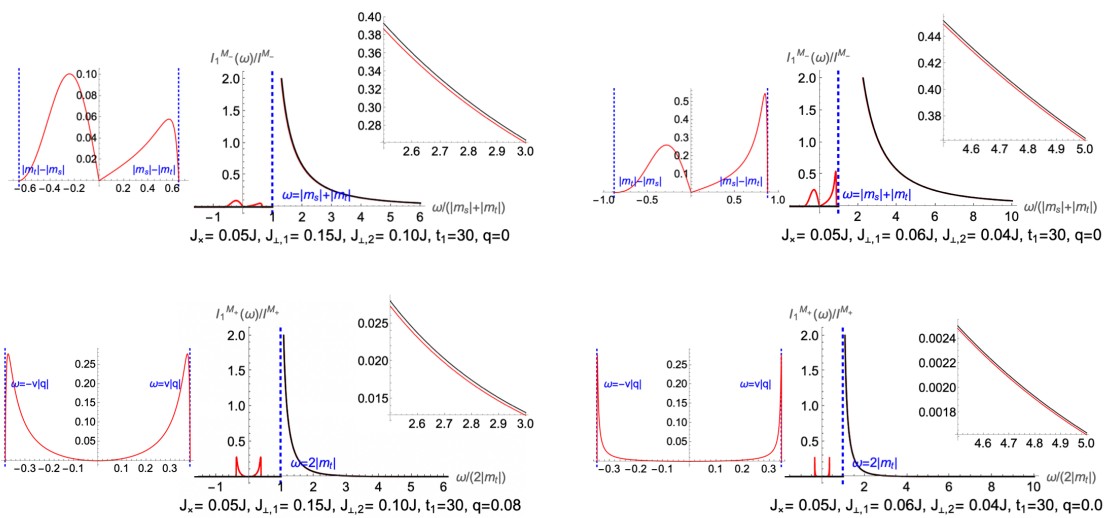

Figure 5.4: The steady state part of the smooth correlations in the limit $\tau \to \infty$, where the signals are normalized by the amplitudes shown in Eq. (5.10). The equilibrium signal is shown in black, while the pumped signal is shown in red. Upper panel shows spectral weight at $(\pi, 0)$. Lower panel is for weight near $(0, 0)$. The regions around $\omega = 0$ and above the thresholds are enlarged for better view.

sectors:

$$M_-(q = 0) \sim \sum_p \left[ c_p^3 c_p^{0\dagger} + c_p^{3\dagger} c_p^0 + c_{-p}^3 c_p^0 + c_{-p}^{3\dagger} c_p^{0\dagger} \right]. \tag{5.15}$$

As a result, acting on the ground state, we have only particle-particle processes corresponding to the threshold at $\omega = |m_s| + |m_t|$ while acting on the coherent state created by the pump, we have additional particle-hole processes corresponding to the region $\omega \in (-|m_s| + |m_t|, |m_s| - |m_t|)$. Similarly, for weight near $(0, 0)$, the $M_+$ correlator is determinative. Here, $M_+$ only involves species in the triplet sector:

$$M_+(q) \sim \sum_p \left[ c_p^2 c_{p-q}^{1\dagger} + c_p^{2\dagger} c_{p+q}^1 + c_{-p}^2 c_{p+q}^1 + c_{-p}^{2\dagger} c_{p-q}^{1\dagger} \right]. \tag{5.16}$$

As a result, acting on the ground state, we have only particle-particle processes corresponding to a threshold at $\omega = 2|m_t|$. Instead acting on the coherent state created by the pump, we have also particle-hole processes corresponding to the region $\omega \in (-v|q|, v|q|)$.

For the transient part, we will first take $\tau = 5$ and show representative results in Fig. 5.5, normalized by the corresponding steady state part. We follow this with a comparison between different choices of $\tau$ (specifically $\tau = 5$ and $\tau = 10$) in Fig. 5.6, where we normalize the results by a factor $I_{2,0}^{M_+}$ depending on $\tau$ (whose expression is presented in Eq. (5.70) of Sec. 5.3) such that the normalized results are effectively $\tau$-independent. In Fig. 5.5 and 5.6, we only show the transient part for the $M_+$ correlator. We will argue in Sec. 5.3 that the transient part for the $M_-$ correlator is negligible. We can see that the transient part performs oscillations under a certain decaying envelope, resembling the phenomenon of beats. In fact, as will be shown later in Sec. 5.3, the decay of the amplitude is algebraic, and the phenomenon of beats is a result from two periods of almost the same value $\tau_o \sim \pi/|vq|$.

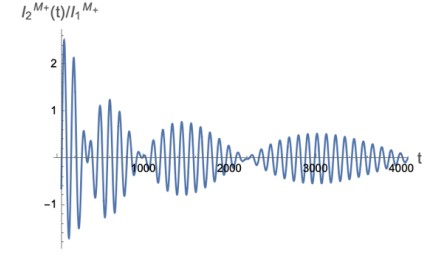
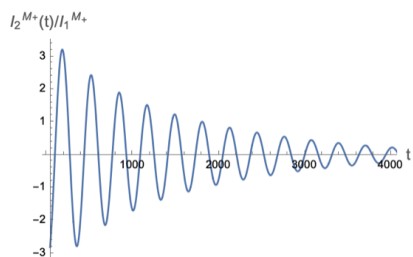

$J_x = 0.05J$, $J_{\perp,1} = 0.15J$, $J_{\perp,2} = 0.10J$, $t_1 = 30$, $\tau = 5$, $q = 0.03$, $\omega = 3|m_t|$     $J_x = 0.05J$, $J_{\perp,1} = 0.06J$, $J_{\perp,2} = 0.04J$, $t_1 = 30$, $\tau = 5$, $q = 0.01$, $\omega = 3|m_t|$

Figure 5.5: The transient part of the smooth correlations for the $M_+$ mode with $\tau = 5$, starting from $t = 40$, and the signals are normalized by the corresponding steady state part. We can see that the time evolution presents oscillations under a decaying envelope, resembling the phenomenon of beats (for the second set of parameters, beats will be seen at even longer times).

## 5.2    Calculation of the Staggered Correlations

In this section, we carry out the approximate evaluation of the staggered correlations. We will resum the leading divergences in the Lehmann expansion, similar to what has been done in [37]. Let us recall the representation for $N_\pm$ in terms of the Ising spin and disorder operators of the Ising model from Eq. (2.12):

$$N_{+,r} \sim \mu^0(x_r)\sigma^1(x_r)\sigma^2(x_r)\mu^3(x_r), \quad N_{-,r} \sim \sigma^0(x_r)\mu^1(x_r)\mu^2(x_r)\sigma^3(x_r). \tag{5.17}$$

and proceed by taking the example of $N_{+,r}$. According to Eq. (5.6), the tr-RIXS response can be expressed as

$$I^{N_+}(\omega, q, t) = \frac{S\tau^3}{\sqrt{2\pi}} \iint_{-\infty}^{\infty} dt_2 dt_2' \int_{-\infty}^{\infty} dx \, e^{i\omega(t_2' - t_2) - iqx} g(t_2, t)^2 \, g(t_2', t)^2 \, \mathcal{I}_{0,\mu} \mathcal{I}_{1,\sigma} \mathcal{I}_{2,\sigma} \mathcal{I}_{3,\mu},$$

$$\mathcal{I}_{i,\mu} \equiv \langle \psi_i(t_2)| \, U(t_2, t_2')\mu^i(x)U(t_2', t_2)\mu^i(0) \, |\psi_i(t_2)\rangle \,,$$

$$\mathcal{I}_{i,\sigma} \equiv \langle \psi_i(t_2)| \, U(t_2, t_2')\sigma^i(x)U(t_2', t_2)\sigma^i(0) \, |\psi_i(t_2)\rangle \,, \quad S = \left( \frac{\lambda}{|m_s m_t^3|^{1/8} a_0} \right)^2,$$

$$\tag{5.18}$$

where the factor $S$ comes from Eq. (2.12), $i$ labels the four Majorana species, and the quantum state $|\psi_i(t_2)\rangle$ is determined in Eq. (3.10). Ignoring the Majorana species label $i$, the four factors actually fall into two types $\mathcal{I}_\sigma$ and $\mathcal{I}_\mu$. Let's proceed with the calculation of $\mathcal{I}_\sigma$, and we will perform a form factor expansion, following the strategy of [37]. Firstly, the quantum state $|\psi(t_2)\rangle$ at $t_2 > t_1$ depends on the sign of the mass parameter:

$$|\psi(t_2)\rangle = \begin{cases} \left( |\psi(t_2)\rangle_{\text{NS}} \pm |\psi(t_2)\rangle_{\text{R}} \right)/\sqrt{2} & m > 0 \\ |\psi(t_2)\rangle_{\text{NS}} & m < 0 \end{cases}, \tag{5.19}$$

and we can expand $|\psi(t_2)\rangle_{\text{NS/R}}$ in the Fock space basis as

$$|\psi(t_2)\rangle_{\text{NS}} = \frac{1}{\mathcal{M}^{1/2}} \sum_{n=0}^{\infty} \frac{1}{n!} \left( \prod_{j=1}^{n} (-u_{k_j}(t_2)) \right) |k_1, -k_1; \cdots ; k_n, -k_n\rangle_{\text{NS}} \,,$$

$$|\psi(t_2)\rangle_{\text{R}} = \frac{e^{-i\Theta(-M_{\text{pump}})|M_{\text{pump}}|t_1}}{\mathcal{M}^{1/2}} \sum_{n=0}^{\infty} \frac{1}{n!} \left( \prod_{j=1}^{n} (-u_{k_j}(t_2)) \right) |k_1, -k_1; \cdots ; k_n, -k_n\rangle_{\text{R}} \,,$$

$$\tag{5.20}$$

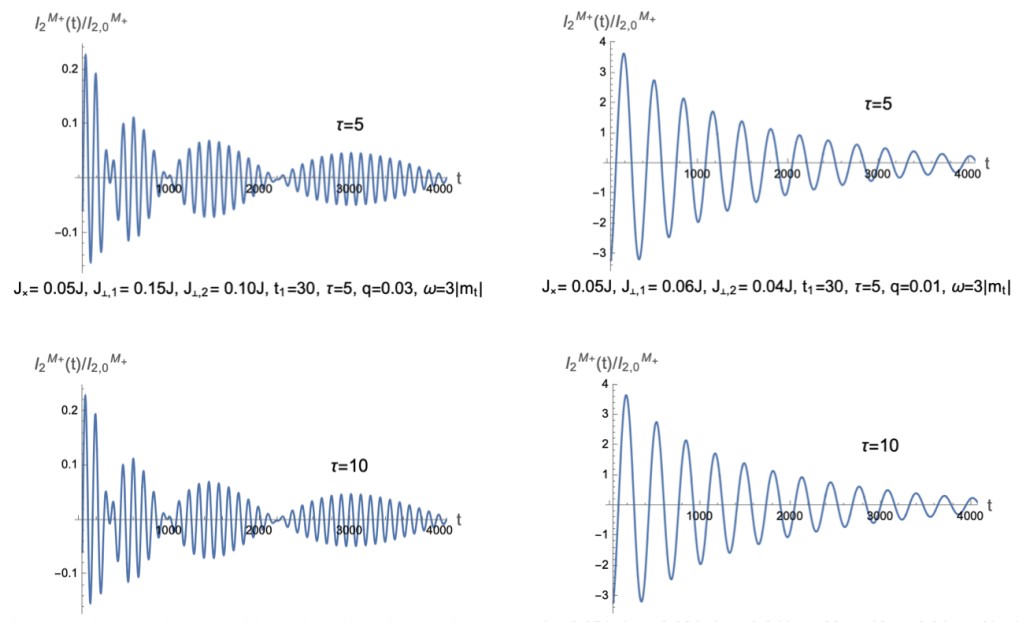

Figure 5.6: The transient part of the smooth correlations for the $M_+$ mode with different choices of $\tau$, starting from $t = 40$, and the signals are normalized by a trivial factor depending on $\tau$ (whose expression will be derived in Eq. (5.70) of Sec. 5.3. We can see that the normalized signals present universal features independent of the choice of $\tau$.

such that $\mathcal{I}_\sigma$ can be expanded in terms of the form factors presented in Eq. (2.14). For example, the contribution to $\mathcal{I}_\sigma$ from the Neveu-Schwarz sector is

$$\mathcal{I}_{\sigma,\mathrm{NS}} \equiv {}_{\mathrm{NS}} \langle \psi(t_2) | \, U(t_2, t_2') \sigma(x) U(t_2', t_2) \sigma(0) \, |\psi(t_2)\rangle_{\mathrm{NS}}$$

$$= \sum_{n,r,l=0}^{\infty} \frac{1}{n! r! l!} \sum_{\substack{\{k>0\in\mathrm{NS}\} \\ \{q>0\in\mathrm{NS}\}}} \sum_{\{p\in\mathrm{R}\}} \left( \prod_{i=1}^{n} \left(-u_{k_i}^*(t_2)\right) \right) \left( \prod_{j=1}^{l} \left(-u_{q_j}(t_2)\right) \right)$$

$$\times e^{i\tilde{t}\left(2\sum_{i=1}^{n} \epsilon_{k_i} - \sum_{j=1}^{r} \epsilon_{p_j}\right)} e^{ix\left(\sum_{j=1}^{r} p_j\right)} \mathcal{F}\mathcal{F}', \tag{5.21}$$

$$\mathcal{F} = \mathcal{M}^{-1/2} {}_{\mathrm{NS}} \langle k_1, -k_1; \cdots; k_n, -k_n | \, \sigma \, |p_1, p_2, \cdots, p_r\rangle_{\mathrm{R}},$$

$$\mathcal{F}' = \mathcal{M}^{-1/2} {}_{\mathrm{R}} \langle p_1, p_2, \cdots, p_r | \, \sigma \, |q_1, -q_1; \cdots; q_l, -q_l\rangle_{\mathrm{NS}},$$

$$u_q(t_2) \equiv u_q\left(m, M_{\mathrm{pump}}, t_1, t_2\right) = i \frac{e^{-2i\epsilon_q(m)(t_2-t_1)} \left(1 - e^{-2i\epsilon_q(M_{\mathrm{pump}})t_1}\right)}{\cot \Delta\theta_q + \tan \Delta\theta_q e^{-2i\epsilon_q(M_{\mathrm{pump}})t_1}},$$

where $\tilde{t} = t_2' - t_2$, and the form factors inside the functions $\mathcal{F}, \mathcal{F}'$ can be read off from Eq. (2.14) by setting $x = 0$. We also copied from Eq. (3.10) the expression for $u_q(t_2)$ to show its dependence on $t_2$. The contribution to $\mathcal{I}_\sigma$ from the Ramond sector ${}_{\mathrm{R}} \langle \psi(t_2)| \cdots |\psi(t_2)\rangle_{\mathrm{R}}$ can be written down similarly, while the contribution to $\mathcal{I}_\sigma$ from the cross terms such as ${}_{\mathrm{NS}} \langle \psi(t_2)| \cdots |\psi(t_2)\rangle_{\mathrm{R}}$ are simply zero. As a result, we have

$$\mathcal{I}_\sigma = \begin{cases} (\mathcal{I}_{\sigma,\mathrm{NS}} + \mathcal{I}_{\sigma,\mathrm{R}})/2 & m > 0 \\ \mathcal{I}_{\sigma,\mathrm{NS}} & m < 0 \end{cases}. \tag{5.22}$$

In the second part of the analysis, the form factor expansion of $\mathcal{I}_\sigma$ can be organized in orders of the small parameter $u_k(t_2)$. Apart from the explicit $u_k(t_2)$ dependence shown in

Eq. (5.20), there is also $u_k(t_2)$ dependence hidden inside the normalization factor $\mathcal{M}$, as shown in Eq. (3.10). Consequently, we first express $\mathcal{M}$ in orders of $u_k(t_2)$:

$$\mathcal{M} = \prod_{k>0\in\mathrm{NS}} [1 + u_k^*(t_2)u_k(t_2)] = 1 + \sum_{n=1}^{\infty} \Upsilon_{2n}, \quad \Upsilon_{2n} = \frac{1}{n!} \underset{\{k>0\in\mathrm{NS}\}}{\sum}' \prod_{i=1}^{n} \left[ u_{k_i}^*(t_2)u_{k_i}(t_2) \right],$$
(5.23)

where the primed sum means that we are summing over non-coinciding values of momenta. Then we express the combination $\mathcal{M}\mathcal{I}_\sigma$ in orders of $u_k(t_2)$:

$$\mathcal{M}\mathcal{I}_\sigma = \sum_{n,r,l=0}^{\infty} C_{(2n|r|2l)},$$
(5.24)

where $(n, r, l)$ labels the number of Majorana fermions as that in Eq. (5.21). The evaluation of Eq. (5.24) differs for $m > 0$ and $m < 0$, so we will discuss them separately in the Sec. 5.2.1 and 5.2.2. We will pick out the most divergent terms at each order in terms of the small parameter $u_k$ and resum them, and then apply the obtained result to our problem, namely the tr-RIXS response in Eq. (5.18). The quality of such an approximation will be discussed later in Sec. 5.2.4.

Before moving on to Sec. 5.2.1 and Sec. 5.2.2 for the calculation of $\mathcal{I}_\sigma$, we remind the reader of the Kramers-Wannier duality by which the correlation functions involving the disorder operator can be obtained. By this duality, we have simply:

$$\mathcal{I}_\mu(m) = \mathcal{I}_\sigma(-m).$$
(5.25)

Consequently, we can focus only on the calculation of $\mathcal{I}_\sigma$, and obtain $\mathcal{I}_\mu$ automatically from Eq. (5.25). By a similar spirit, the correlation function $I^{N-}$ can be obtained from $I^{N+}$ straightforwardly by utilizing Eq. (5.25).

### 5.2.1 The Ordered Phase with $m > 0$

Firstly, we consider the expansion of $\mathcal{M}\mathcal{I}_\sigma$ in the case $m > 0$, which corresponds to the ordered phase of the associated quantum Ising model. It can be shown that $\mathcal{M}\mathcal{I}_{\sigma,\mathrm{NS}} = \mathcal{M}\mathcal{I}_{\sigma,\mathrm{R}}$ for $m > 0$, since the Ramond sector only introduces an extra pure phase in $|\psi(t_2)\rangle_\mathrm{R}$ for $t_2 > t_1$ when compared with the Neveu-Schwarz sector, see Eq. (3.10) and (3.14). As a result, we do not need to specify the sector in this section. We are also primarily concerned with the small $q$ behavior of $I^{N+}(\omega, q, t)$, or equivalently, the large $x$ behavior of $\mathcal{M}\mathcal{I}_\sigma$. It can be shown that the leading singularities are those at order $u^{2n}, n = 0, 1, 2, \ldots$, and the most singular contribution at order $u^{2n}$ is $C_{(2n|2n|2n)}$ [37]. A simplified argument of this statement is given in Appendix B. At order $u^0$, the leading contribution is simply

$$C_{(0|0|0)} = \bar{\sigma}^2.$$
(5.26)

At order $u^2$, the leading contribution is

$$C_{(2|2|2)} = \frac{\bar{\sigma}^2}{2} \sum_{k>0\in\mathrm{NS}} \sum_{q>0\in\mathrm{NS}} \sum_{p_1\in\mathrm{R}} \sum_{p_2\in\mathrm{R}} \frac{v^4 u_k^*(t_2)u_q(t_2)e^{i\bar{t}(2\epsilon_k-\epsilon_{p_1}-\epsilon_{p_2})}e^{ix(p_1+p_2)}}{|m|^4 L^4 \cosh\vartheta_k \cosh\vartheta_q \cosh\vartheta_{p_1} \cosh\vartheta_{p_2}}$$

$$\times \tanh\vartheta_k \tanh\vartheta_q \tanh\left(\frac{\vartheta_{p_1}-\vartheta_{p_2}}{2}\right) \tanh\left(\frac{\vartheta_{p_1}-\vartheta_{p_2}}{2}\right)$$

$$\times \coth\left(\frac{\vartheta_k-\vartheta_{p_1}}{2}\right) \coth\left(\frac{\vartheta_k+\vartheta_{p_1}}{2}\right) \coth\left(\frac{\vartheta_k-\vartheta_{p_2}}{2}\right) \coth\left(\frac{\vartheta_k+\vartheta_{p_2}}{2}\right)$$

$$\times \coth\left(\frac{\vartheta_q-\vartheta_{p_1}}{2}\right) \coth\left(\frac{\vartheta_q+\vartheta_{p_1}}{2}\right) \coth\left(\frac{\vartheta_q-\vartheta_{p_2}}{2}\right) \coth\left(\frac{\vartheta_q+\vartheta_{p_2}}{2}\right).$$
(5.27)

The evaluation of the summations over momentum is a non-trivial task due to the presence of annihilation poles where the function coth diverges. The way to deal with these poles is described in Appendix B with the result

$$C_{(2|2|2)} = \bar{\sigma}^2 \left( \Upsilon_2 - \frac{4}{L} \sum_{k>0\in\text{NS}} u_k^*(t_2) u_k(t_2) \mathcal{G}(k, x, t_2 + t_2') \right),$$

$$\mathcal{G}(k, x, t_2 + t_2') = |x| + \Theta\left(-\epsilon_k'|t_2 + t_2'| + |x|\right)\left[\epsilon_k'|t_2 + t_2'| - |x|\right],$$
(5.28)

apart from terms that are either algebraically or exponentially small in $t_2 + t_2'$ and $|x|$.

In Eq. (5.28), $\epsilon_k'$ is the derivative of $\epsilon_k$ with respect to $k$, and the first term in $C_{(2|2|2)}$ just offsets the same term in the normalization factor $\mathcal{M}$ in Eq. (5.23). Generally, at order $u^{2n}$, evaluating $C_{(2n|2n|2n)}$ is just an exercise of combinatorics in $n$ copies of $C_{(2|2|2)}$, with the ease to anticipate the result

$$C_{(2n|2n|2n)} = \frac{\bar{\sigma}^2}{n!} \sum_{\{k>0\in\text{NS}\}}' \prod_{i=1}^n u_{k_i}^*(t_2) u_{k_i}(t_2) \left[1 - \frac{4}{L} \mathcal{G}(k_i, x, t_2 + t_2')\right].$$
(5.29)

Finally, we sum all the leading contributions $C_{(2n|2n|2n)}$ together:

$$\mathcal{M}\mathcal{I}_\sigma \approx \sum_{n=0}^\infty C_{(2n|2n|2n)} = \bar{\sigma}^2 \exp\left\{ \sum_{k>0\in\text{NS}} \ln\left[1 + u_k^*(t_2) u_k(t_2)\left(1 - \frac{4}{L}\mathcal{G}(k, x, t_2 + t_2')\right)\right]\right\}$$

$$\approx \mathcal{M}\bar{\sigma}^2 \exp\left[-\frac{4}{L} \sum_{k>0\in\text{NS}} u_k^*(t_2) u_k(t_2) \mathcal{G}(k, x, t_2 + t_2')\right]$$

$$= \mathcal{M}\bar{\sigma}^2 \exp\left[-\frac{2}{\pi} \int_0^\infty \mathrm{d}k\, u_k^*(t_2) u_k(t_2) \mathcal{G}(k, x, t_2 + t_2')\right],$$
(5.30)

where the normalization factor $\mathcal{M}$ is factored out so that a closed expression for $\mathcal{I}_\sigma$ can be easily written down. In Appendix B we discuss the Fourier transforms of $\mathcal{I}_\sigma$. These transforms can be used in a simple analysis to infer the thresholds shown in Fig. 5.1.

### 5.2.2 The Disordered Phase with $m < 0$

Secondly, we consider the expansion of $\mathcal{M}\mathcal{I}_\sigma$ in the case $m < 0$, which corresponds to the disordered phase of the associated quantum Ising model. Here $|\psi(t)\rangle = |\psi(t)\rangle_{\text{NS}}$ and thus we work solely in the Neveu-Schwarz sector. It can be shown that the most singular terms are $C_{(2n|2n+1|2n)}$ at $\mathcal{O}(u^{2n})$ and $C_{(2n+2|2n+1|2n)} + C_{(2n|2n+1|2n+2)}$ at $\mathcal{O}(u^{2n+1})$. Let us look at the even orders first. At order $u^0$, we have

$$C_{(0|1|0)} = \bar{\sigma}^2 \sum_{p\in\text{R}} \frac{v e^{-i\tilde{t}\epsilon_p} e^{ixp}}{|m|L\cosh\vartheta_p} = \bar{\sigma}^2 v \int_{-\infty}^\infty \frac{\mathrm{d}p}{2\pi} \frac{e^{-i\tilde{t}\epsilon_p + ipx}}{\epsilon_p} \equiv \Sigma_1.$$
(5.31)

At order $u^2$, we have

$$C_{(2|3|2)} = \frac{\bar{\sigma}^2}{6} \sum_{k>0\in\text{NS}} \sum_{q>0\in\text{NS}} \sum_{p_1,p_2,p_3\in\text{R}} \frac{u_k^*(t_2) u_q(t_2) e^{i\tilde{t}(2\epsilon_k - \epsilon_{p_1} - \epsilon_{p_2} - \epsilon_{p_3})} e^{ix(p_1+p_2+p_3)} v^5}{|m|^5 L^5 \cosh\vartheta_k \cosh\vartheta_q \cosh\vartheta_{p_1} \cosh\vartheta_{p_2} \cosh\vartheta_{p_3}}$$

$$\times \tanh\vartheta_k \tanh\vartheta_q \tanh^2\left(\frac{\vartheta_{p_1} - \vartheta_{p_2}}{2}\right) \tanh^2\left(\frac{\vartheta_{p_1} - \vartheta_{p_3}}{2}\right) \tanh^2\left(\frac{\vartheta_{p_2} - \vartheta_{p_3}}{2}\right)$$

$$\times \prod_{i=1}^3 \coth\left(\frac{\vartheta_k - \vartheta_{p_i}}{2}\right) \coth\left(\frac{\vartheta_k + \vartheta_{p_i}}{2}\right) \coth\left(\frac{\vartheta_q - \vartheta_{p_i}}{2}\right) \coth\left(\frac{\vartheta_q + \vartheta_{p_i}}{2}\right).$$
(5.32)

The treatment of the summations over momentum is the same as that for the ordered phase with $m > 0$, and the result is

$$C_{(2|3|2)} = \Sigma_1 \sum_{k>0\in\mathrm{NS}} u_k^*(t_2)u_k(t_2)\left[1 - \frac{4}{L}|x|\right]. \tag{5.33}$$

Generally, at $\mathcal{O}(u^{2n})$, we have a combinatorial exercise involving $n$ copies of $C_{(2|3|2)}$, with the result

$$C_{(2n|2n+1|2n)} = \frac{\Sigma_1}{n!}\left[1 - \frac{4}{L}|x|\right]^n \sideset{}{'}\sum_{\{k>0\in\mathrm{NS}\}} \prod_{i=1}^n u_{k_i}^*(t_2)u_{k_i}(t_2). \tag{5.34}$$

Now let's look at the odd orders. At order $u^1$, we have

$$C_{(2|1|0)} + C_{(0|1|2)} = \bar{\sigma}^2 \sum_{k>0\in\mathrm{NS}}\sum_{p\in\mathrm{R}} \frac{iv^2 e^{-i\tilde{t}\epsilon_p}e^{ixp}\tanh\vartheta_k}{|m|^2 L^2 \cosh\vartheta_k \cosh\vartheta_p}$$
$$\times \coth\left(\frac{\vartheta_k - \vartheta_p}{2}\right)\coth\left(\frac{\vartheta_k + \vartheta_p}{2}\right)\left(u_k^*(t_2)e^{2i\tilde{t}\epsilon_k} - u_k(t_2)\right). \tag{5.35}$$

Summation over $p$ gives us

$$C_{(2|1|0)} + C_{(0|1|2)} = 4\bar{\sigma}^2 v \int_0^\infty \frac{\mathrm{d}k}{2\pi}\frac{\sin(k|x|)}{\epsilon_k}\,\mathrm{Im}\left[u_k(t_2)e^{-i\epsilon_k\tilde{t}}\right] \equiv \Sigma_2. \tag{5.36}$$

At order $u^3$, or more generally $u^{2n+1}$, the evaluation uses the tricks that we have already discussed. We thus just the final expression:

$$C_{(2n+2|2n+1|2n)} + C_{(2n|2n+1|2n+2)}$$
$$= \frac{\Sigma_2}{n!}\sideset{}{'}\sum_{\{k>0\in\mathrm{NS}\}} \prod_{i=1}^n\left[u_{k_i}^*(t_2)u_{k_i}(t_2)\left(1 - \frac{4}{L}\mathcal{G}(k_i, x, t_2 + t_2')\right)\right]. \tag{5.37}$$

Finally, we sum all orders up and obtain

$$\mathcal{MI}_{\sigma,\mathrm{NS}} \approx \sum_{n=0}^\infty C_{(2n|2n+1|2n)} + \sum_{n=0}^\infty\left[C_{(2n+2|2n+1|2n)} + C_{(2n|2n+1|2n+2)}\right]$$
$$\approx \mathcal{M}\left\{\Sigma_1\exp\left[-\frac{2}{\pi}\int_0^\infty \mathrm{d}k\, u_k^*(t_2)u_k(t_2)|x|\right]\right.$$
$$\left. + \Sigma_2\exp\left[-\frac{2}{\pi}\int_0^\infty \mathrm{d}k\, u_k^*(t_2)u_k(t_2)\mathcal{G}(k, x, t_2 + t_2')\right]\right\}. \tag{5.38}$$

where $\mathcal{G}(k, x, t_2 + t_2')$ is defined in Eq. 5.28. In fact, the transient behaviors shown in Fig. 5.2 come from the second term of Eq. (5.38) that is proportional to $\Sigma_2$. As with the correlator for the ordered phase, in Appendix B, we discuss the Fourier transform of $\mathcal{I}_\sigma$ in order to infer the thresholds shown in Fig. 5.1.

### 5.2.3 Stitching the Results Together

In the previous subsections, we have evaluated $\mathcal{I}_\sigma(m)$ explicitly up to exponential accuracy while $\mathcal{I}_\mu(m)$ can be obtained via the duality transformation in Eq. (5.25). With the explicit expressions for both $\mathcal{I}_\sigma(m)$ and $\mathcal{I}_\mu(m)$ at our disposal, we can now proceed with the evaluation of $I^{N_\pm}(\omega, q, t)$. We also know from Sec. 5.2.1 that the Ramond sector makes the same contribution as the Neveu-Schwarz sector for $m > 0$. Consequently, by

Eq. (5.22) and (5.25), the calculation of $I^{N\pm}(\omega, q, t)$ boils down to the evaluation of $\mathcal{I}_\sigma(m)$ for both signs of the mass. For reference, we have listed the expressions for $I^{N\pm}(\omega, q, t)$ in terms of $\mathcal{I}_\sigma(m)$ in Appendix C. There, we can see that the expressions for $I^{N\pm}(\omega, q, t)$ are a convolution of the product of four $\mathcal{I}_\sigma(m)$'s, which generate a proliferation of terms. However if we work in the limit of large $t$ and small $q$, some terms are exponentially suppressed. The series of manipulations are shown in Appendix D, and the detailed results for finite measurement time $\tau$ are shown in E. Again there are two parts in the tr-RIXS response, the steady state part, $I_1^{N\pm}(\omega, q)$, and the transient part, $I_2^{N\pm}(\omega, q, t)$. We discuss both in this subsection.

A sanity check of our results is letting $M_{s,\text{pump}}, M_{t,\text{pump}} \to m_s, m_t$ where we expect a smooth crossover from the pumped ladder to the unpumped ladder. This is indeed the case, with the corresponding results for the unquenched ladder shown in Appendix E and a leading order correction scaling as $(M_{s,\text{pump}} - m_s)^2$ or $(M_{t,\text{pump}} - m_t)^2$. The final result for $I^{N\pm}(\omega, q, t)$ consists of a steady state part and a transient part, expressed as

$$I^{N\pm}(\omega, q, t) = I_1^{N\pm}(\omega, q) + I_2^{N\pm}(\omega, q, t), \tag{5.39}$$

For the steady state part, we are interested in its behavior in the limit $\tau \to \infty$. Here we present the approximated results at $q = 0$:

$$
\begin{aligned}
I_1^{N+}(\omega, q = 0) \approx & \frac{1.3578^8 \lambda^2}{a_0^2} \frac{4}{(2\pi)^4} \int_{\epsilon_1}^{\epsilon_2} d\epsilon \frac{\Theta(\omega - |m_s| - 2|m_t|)}{\sqrt{(\epsilon^2 - v^2 p_0^2 - 4m_t^2)(\epsilon^2 - v^2 p_0^2)}} \frac{1}{p_0} \\
& \times \left[ \arctan\left( \frac{v(p_0 + p^*)}{\Delta} \right) - \arctan\left( \frac{v(p_0 - p^*)}{\Delta} \right) \right] \\
& + \frac{1.3578^8 \lambda^2}{a_0^2} \frac{1}{(2\pi)^4} \int_{2|m_t|}^{\epsilon_1} d\epsilon \frac{2}{\sqrt{2v^2 m_t^2}} \frac{1}{p_0} \Theta(\omega - |m_s| - 2|m_t|) \\
& \times \left[ \frac{2v\Delta}{\Delta^2 + v^2(p_0 + p^*)^2} + \frac{2v\Delta}{\Delta^2 + v^2(p_0 - p^*)^2} \right]
\end{aligned} \tag{5.40}
$$

$$
I_1^{N-}(\omega, q = 0) \approx \frac{1.3578^8 \lambda^2}{a_0^2} \frac{1}{(2\pi)^2} \frac{v\Delta}{\Delta^2 + \omega^2 - m_t^2} \frac{1}{\sqrt{\omega^2 - m_t^2}} \Theta(\omega - |m_t|),
$$

where $\lambda$ is some constant on the order of one, $a_0$ is the lattice constant, $\epsilon_1, \epsilon_2, p_0, p^*$ and the small parameter $\Delta$ is defined as

$$
\begin{aligned}
\epsilon_1 &= \frac{\omega^2 - m_s^2 + 4m_t^2}{2\omega}, \quad \epsilon_2 = \omega - |m_s|, \\
p_0 &= \sqrt{\frac{(\omega - \epsilon)^2 - m_s^2}{v^2}}, \quad p^* = \sqrt{\frac{\epsilon^2 - 4m_t^2}{v^2}}, \\
\Delta &= v \frac{2}{\pi} \int_0^\infty dp \left( u_p^{s*} u_p^s + 3 u_p^{t*} u_p^t \right) \propto \frac{(m_s - M_{s,\text{pump}})^2}{|m_s|} + 3 \frac{(m_t - M_{t,\text{pump}})^2}{|m_t|}.
\end{aligned} \tag{5.41}
$$

For comparison, the corresponding results for the unpumped ladder are:

$$
\begin{aligned}
I_1^{N+}(\omega, q = 0) &\approx \frac{1.3578^8 \lambda^2}{a_0^2} \frac{2}{(2\pi)^3} \int_{\epsilon_1}^{\epsilon_2} d\epsilon \frac{\Theta(\omega - |m_s| - 2|m_t|)}{\sqrt{(\epsilon^2 - v^2 p_0^2 - 4m_t^2)(\epsilon^2 - v^2 p_0^2)}} \frac{1}{p_0}, \\
I_1^{N-}(\omega, q = 0) &\approx \frac{1.3578^8 \lambda^2}{a_0^2} \frac{1}{4\pi} \frac{v}{|m_t|} \delta(\omega - |m_t|)
\end{aligned} \tag{5.42}
$$

As shown in Fig. 5.1, for the $N_-$ mode, the pump changes the delta-function peak at $\omega = |m_t|$ to a square-root singularity at $\omega = |m_t|$. On the other hand, for the $N_+$ mode,

there is no singularity at the threshold $\omega = |m_s| + 2|m_t|$, and the pump suppresses the signal near the threshold while enhancing the signal along the tail.

For the transient part, we consider $q = 0$ with finite $\tau$ and find the following expressions:

$$
I_2^{N+}(\omega, t) \approx -\frac{1.3578^8 \lambda^2}{a_0^2} \iiint_{-\infty}^{\infty} \frac{\mathrm{d}p_1 \mathrm{d}p_2 \mathrm{d}p_3}{(2\pi)^3} \frac{v^3}{\epsilon_{p_1}^s \epsilon_{p_2}^t \epsilon_{p_3}^t} \frac{\tau}{4\pi\sqrt{2\pi}} \frac{2iv^2(p_1 + p_2 + p_3)}{\Delta^2 + v^2(p_1 + p_2 + p_3)^2}
$$

$$
\times \left[ u_{p_1}^s e^{-\frac{\tau^2}{2}\left[(-\omega + \epsilon_{p_2}^t + \epsilon_{p_3}^t)^2 + (\epsilon_{p_1}^s)^2\right]} + [(p_1, s) \leftrightarrow (p_2, t)] + [(p_1, s) \leftrightarrow (p_3, t)] - h.c. \right],
$$

$$
I_2^{N-}(\omega, t) \approx -\frac{1.3578^8 \lambda^2}{a_0^2} \int_{-\infty}^{\infty} \frac{\mathrm{d}p}{2\pi} \frac{v}{\epsilon_p^t} \frac{\tau}{4\pi\sqrt{2\pi}} \frac{2iv^2 p}{\Delta^2 + v^2 p^2} \left[ u_p^t e^{-\frac{\tau^2}{2}\left[\omega^2 + (\epsilon_p^t)^2\right]} - h.c. \right].
$$
(5.43)

Let's look at the $N_-$ mode first, which can be further approximated as

$$
I_2^{N-}(\omega, t) \approx I_{2,0}^{N-} \int_{-\infty}^{\infty} \mathrm{d}p F_{\mathrm{en}}(p) F_{\mathrm{os}}(p, t),
$$

$$
I_{2,0}^{N-} = \frac{1.3578^8 \lambda^2}{a_0^2} \frac{v\tau}{8\pi^2\sqrt{2\pi}|m_t|} e^{-\frac{\tau^2}{2}\left(\omega^2 + m_t^2\right)},
$$

$$
F_{\mathrm{en}}(p) = \frac{[2v|m_t|(m_t - M_{t,\mathrm{pump}})] v^2 p^2}{(\Delta^2 + v^2 p^2)(v^2 p^2 + |m_t M_{t,\mathrm{pump}}|)\sqrt{v^2 p^2 + m_t^2}} e^{-\frac{\tau^2}{2}v^2 p^2},
$$

$$
F_{\mathrm{os}}(p, t) = \mathrm{Re}\left[ e^{-2i\epsilon_p(m_t)(t - t_1)}\left(1 - e^{-2i\epsilon_p(M_{t,\mathrm{pump}})t_1}\right)\right],
$$
(5.44)

where $I_{2,0}^{N-}$ is a trivial factor depending on $\tau$. We will only present results with this factor divided out in order to exhibit universal behavior across different choices of $\tau$. For a fixed $t$, the integrand is an oscillating function $F_{\mathrm{os}}(p, t)$ with respect to $p$ multiplied by an envelope function $F_{\mathrm{en}}(p)$. The shape of the envelope is determined by a combination of mass parameters $\Delta$, $\sqrt{|m_t M_{t,\mathrm{pump}}|}$, and $|m_t|$, while the oscillation with respect to $p$ becomes faster with increasing $t$, thus reducing the overall amplitude. Rough estimates for the time constants for $I_2^{N-}(\omega, t)$, $\tau_o$ for the oscillations and $\tau_r$ for the decay in the envelope function are

$$
\tau_o \sim \frac{\pi}{|m_t|}, \quad \tau_r \sim \frac{4\pi}{\sqrt{\Delta^2 + |m_t M_{t,\mathrm{pump}}| + m_t^2/2}},
$$
(5.45)

provided that the measurement time $\tau$ is smaller than $\tau_r$.

Next we look at the $N_+$ correlator. In comparison with its $N_-$ counterpart, which has $pu_p^t$ inside its integrand, here we have $(p_1 + p_2 + p_3)u_{p_1}^s$, $(p_1 + p_2 + p_3)u_{p_2}^t$ and $(p_1 + p_2 + p_3)u_{p_3}^t$. Since $u_p^{s,t}$ is an odd function, this brings near-cancellation between the positive and the negative integration intervals. An implementation of numerical evaluation of $I_2^{N+}(\omega, t)$ confirms this intuition, which shows that the transient part is several orders of magnitude smaller than the steady state part and thus is negligible.

### 5.2.4   Validity of the Approximation

The resummation adopted in Sec. 5.2.1 and 5.2.2 only picks out the most divergent terms at each order in terms of the small parameter $u_k$. Here we discuss in what sense such an approximation is valid. Let us recall the complete expression for $\mathcal{I}_\sigma$ in Eq. (5.24):

$$
\mathcal{M}^2 \mathcal{I}_\sigma = \sum_{n,r,l=0}^{\infty} C_{(2n|r|2l)},
$$
(5.46)

where the expansion terms are

$$
\begin{aligned}
C_{(2n|r|2l)} = \frac{1}{n!r!l!} \sum_{\substack{\{k>0\in\text{NS}\}\\\{q>0\in\text{NS}\}}} \sum_{\{p\in\text{R}\}} &\left(\prod_{i=1}^{n}(-u_{k_i}^*(t_2))\right)\left(\prod_{j=1}^{l}(-u_{q_j}(t_2))\right) \\
&\times e^{i\tilde{t}\left(2\sum_{i=1}^{n}\epsilon_{k_i}-\sum_{j=1}^{r}\epsilon_{p_j}\right)}e^{ix\left(\sum_{j=1}^{r}p_j\right)}\mathcal{F}\mathcal{F}',
\end{aligned}
$$

$$
\mathcal{F} = {}_{\text{NS}}\langle k_1,-k_1;\cdots;k_n,-k_n|\,\sigma\,|p_1,p_2,\cdots,p_r\rangle_{\text{R}}
$$

$$
\mathcal{F}' = {}_{\text{R}}\langle p_1,p_2,\cdots,p_r|\,\sigma\,|q_1,-q_1;\cdots;q_l,-q_l\rangle_{\text{NS}},
$$

(5.47)

where $\tilde{t} = t_2' - t_2$. Their time dependence goes as:

$$
C_{(2n|r|2l)} \sim e^{i\omega\tilde{t}}e^{i\tilde{t}\left(2\sum_{i=1}^{n}\epsilon_{k_i}-\sum_{j=1}^{r}\epsilon_{p_j}\right)}e^{2it_2\left(-\sum_{i=1}^{l}\epsilon_{q_i}+\sum_{i=1}^{n}\epsilon_{k_i}\right)}. \tag{5.48}
$$

Having the expansion for $\mathcal{I}_\sigma$, the expression of the tr-RIXS signal can be written as a convolution of products of four such factors, each corresponding to one species of the Majoranas:

$$
\frac{\tau^3}{\sqrt{2\pi}}\int \mathrm{d}t_2 \int \mathrm{d}t_2'\; e^{i\omega\tilde{t}}g(t_2,t)^2 g(t_2',t)^2\mathcal{I}_0\mathcal{I}_1\mathcal{I}_2\mathcal{I}_3. \tag{5.49}
$$

Because the response involves a product of $\mathcal{I}_i$, $i = 0,1,2,3$, the threshold frequency at which the steady state response is non-zero is additive in terms of the thresholds of the individual $\mathcal{I}_i$.

Let us take for a simple example the case of $N_-$ with the $m_s > 0, m_t < 0$, where what we have for $\mathcal{I}_0\mathcal{I}_1\mathcal{I}_2\mathcal{I}_3$ is

$$
\begin{aligned}
&\mathcal{I}_\sigma(m_s > 0)\mathcal{I}_\mu(m_t < 0)\mathcal{I}_\mu(m_t < 0)\mathcal{I}_\sigma(m_t < 0) \\
=\; &\mathcal{I}_\sigma(m_s > 0)\mathcal{I}_\sigma(-m_t > 0)\mathcal{I}_\sigma(-m_t > 0)\mathcal{I}_\sigma(m_t < 0).
\end{aligned} \tag{5.50}
$$

Let's first look at the terms that are included in our approximate result. For the first three factors, we have included

$$
\sum_{n=0}^{\infty} C_{(2n|2n|2n)}, \tag{5.51}
$$

while for the last factor, we have included

$$
\sum_{n=0}^{\infty} C_{(2n|2n+1|2n)} + \sum_{n=0}^{\infty}\left(C_{(2n+2|2n+1|2n)} + C_{(2n|2n+1|2n+2)}\right), \tag{5.52}
$$

where the first term of $\mathcal{I}_3$ gives the steady state part while the second term gives the transient part. Here we restrict ourselves to considering only the steady state part.

Referring to Eq. (5.48) and (5.49), it is easy to see that our approximate result for the steady state part is only non-zero for omega satisfying

$$
\omega \geq 0 + 0 + 0 + |m_t| = |m_t|, \tag{5.53}
$$

where $\mathcal{I}_{0,1,2}$ each permits weight for $\omega$ 0 while $\mathcal{I}_3$ only permits weight at $\omega > |m_t|$. A simple argument for such a threshold is presented in Appendix B.

To demonstrate that the spectral weights around other frequencies are smaller, let us consider the weight about $3|m_t|$ arising from the next set of leading terms:

$$
\begin{aligned}
\omega &= 0 + 2|m_t| + 0 + |m_t| = 3|m_t|, \\
\text{or}\quad \omega &= 0 + 0 + 2|m_t| + |m_t| = 3|m_t|, \\
\text{or}\quad \omega &= 0 + 0 + 0 + 3|m_t| = 3|m_t|.
\end{aligned} \tag{5.54}
$$

Here we assume that $|m_s| > |m_t|$ and so $3|m_t|$ is the first threshold beyond $|m_t|$ at which weight occurs. The new expansion terms corresponding to the three cases listed above are

$$
\begin{aligned}
\text{from } \mathcal{I}_1 : \quad & \sum_{n=0}^{\infty} C_{(2n|2n+2|2n)}, \\
\text{from } \mathcal{I}_2 : \quad & \sum_{n=0}^{\infty} C_{(2n|2n+2|2n)}, \\
\text{from } \mathcal{I}_3 : \quad & \sum_{n=0}^{\infty} C_{(2n|2n+3|2n)}.
\end{aligned}
\tag{5.55}
$$

They can be resumed in a similar fashion to those terms corresponding to our approximate result. For the purpose of comparison, we shall take the unquenched limit $\Delta \to 0$ and look at the spectral weight at $q = 0$:

$$
\begin{aligned}
I^{N_-}_{3|m_t|}(0, \omega) \approx & \frac{1.3578^8 \lambda^2}{a_0^2} \iiint_{-\infty}^{\infty} \frac{\mathrm{d}p_1 \mathrm{d}p_2 \mathrm{d}p_3}{(2\pi)^3} \frac{v^3}{\epsilon_{p_1}^t \epsilon_{p_2}^t \epsilon_{p_3}^t} \frac{\tau}{4\pi\sqrt{2\pi}} \tanh^2\left(\frac{\vartheta_{p_1}^t - \vartheta_{p_2}^t}{2}\right) \\
& \times \left[1 + \frac{1}{3!} \tanh^2\left(\frac{\vartheta_{p_1}^t - \vartheta_{p_3}^t}{2}\right) \tanh^2\left(\frac{\vartheta_{p_2}^t - \vartheta_{p_3}^t}{2}\right)\right] \\
& \times (2\pi)\delta(p_1 + p_2 + p_3) e^{-\frac{\tau^2}{2}\left(-\omega + \epsilon_{p_1,t} + \epsilon_{p_2,t} + \epsilon_{p_3,t}\right)^2},
\end{aligned}
\tag{5.56}
$$
$$
I^{N_-}_{1|m_t|}(0, \omega) \approx \frac{1.3578^8 \lambda^2}{a_0^2} \int_{-\infty}^{\infty} \frac{\mathrm{d}p}{2\pi} \frac{v}{\epsilon_p^t} \frac{\tau}{4\pi\sqrt{2\pi}} (2\pi)\delta(p) e^{-\frac{\tau^2}{2}\left(-\omega + \epsilon_p^t\right)^2},
$$

where the angle $\vartheta_p^t$ is defined in Eq. (2.15) with the triplet mass $m_t$. We can check their relative importance by comparing their integrated spectral weights:

$$
r \equiv \frac{\int \mathrm{d}\omega I^{N_-}_{3|m_t|}(0, \omega)}{\int \mathrm{d}\omega I^{N_-}_{1|m_t|}(0, \omega)}
\tag{5.57}
$$

We can numerically evaluate this ratio $r$ to be $\approx 0.278$, independent of $|m_t|$. While not negligible, this contribution is significantly smaller than the leading contribution. We expect this to continue to be true for the pumped system. We also expect that at higher thresholds the weight will drop off precipitously as has been observed in a single copy of an Ising model [52].

## 5.3   Calculation of the Smooth Correlations

In this section, we carry out the exact evaluation of the smooth correlations. Depending on the sign of the mass parameter, for each Majorana species, the quantum state at $t_2 > t_1$ is

$$
|\psi(t_2)\rangle = \begin{cases} \left(|\psi(t_2)\rangle_{\mathrm{NS}} \pm |\psi(t_2)\rangle_{\mathrm{R}}\right)/\sqrt{2} & m > 0 \\ |\psi(t_2)\rangle_{\mathrm{NS}} & m < 0 \end{cases},
\tag{5.58}
$$

where $|\psi(t_2)\rangle_{\mathrm{NS}}$ and $|\psi(t_2)\rangle_{\mathrm{R}}$ are determined in Eq. (3.10) and (3.10). As a result, we need to consider both the Neveu-Schwarz sector and the Ramond sector when $m > 0$. For all species of the Majorana fermions sitting in the Neveu-Schwarz sector, the smooth component of the spin operator can be mode expanded using the massless Majorana modes

as

$$
\begin{aligned}
M_{+,q}^{\mathrm{NS}} =\frac{i}{2}&\left[\sum_{q_1>0}\left(c_{-q_1}^2 c_{-q_1-|q|}^{1\dagger}+c_{q_1}^{2\dagger}c_{q_1+|q|}^1\right)+\sum_{q_1>|q|}\left(c_{-q_1}^{2\dagger}c_{-q_1+|q|}^1+c_{q_1}^2 c_{q_1-|q|}^{1\dagger}\right)\right.\\
&\left.+\sum_{0<q_1<|q|}\left(c_{-q_1}^{2\dagger}c_{q_1-|q|}^{1\dagger}+c_{q_1}^2 c_{-q_1+|q|}^1\right)\right],\\
M_{-,q}^{\mathrm{NS}} =\frac{i}{2}&\left[\sum_{q_1>0}\left(c_{-q_1}^3 c_{-q_1-|q|}^{0\dagger}+c_{q_1}^{3\dagger}c_{q_1+|q|}^0\right)+\sum_{q_1>|q|}\left(c_{-q_1}^{3\dagger}c_{-q_1+|q|}^0+c_{q_1}^3 c_{q_1-|q|}^{0\dagger}\right)\right.\\
&\left.+\sum_{0<q_1<|q|}\left(c_{-q_1}^{3\dagger}c_{q_1-|q|}^{0\dagger}+c_{q_1}^3 c_{-q_1+|q|}^0\right)\right],
\end{aligned}
\tag{5.59}
$$

where the superscripts $(2,1)$ and $(3,0)$ label the species of the Majorana fermions, and Eq. (5.59) is valid for all $q$. The mode expansions for $M_{\pm,q}$ have exactly the same structure, only the species pair $(2,1)$ is replaced by $(3,0)$. But this will lead to important differences between them if there is mismatch between the singlet and triplet masses. In order to be applied directly on the state $|\psi_{\mathrm{NS}}(t_2)\rangle$, we need to convert the massless Majorana modes to those with $m_s, m_t$:

$$
\begin{aligned}
c_k^0 &= \alpha_k(m_s)c_k^0(m_s) - \beta_k(m_s)c_{-k}^{0\dagger}(m_s),\\
c_k^i &= \alpha_k(m_t)c_k^i(m_t) - \beta_k(m_t)c_{-k}^{i\dagger}(m_t),\quad i=1,2,3.
\end{aligned}
\tag{5.60}
$$

The evaluation of $I^{M_\pm}(\omega,q,t)$ in the Neveu-Schwarz sector is thus straightforward using Eq. (3.10), (5.8), (5.59) and (5.60), as everything can be expressed in terms of the massive modes with masses, $m_s, m_t$. We are primarily concerned with the bulk response and the long time behavior, so we will take the infinite size limit $L\to\infty$ and look at large $t$ behavior, such that the summations over momentum can be converted into integrals and terms exponentially small in $t$ can be neglected. Then the integration over $t_2, t_2'$ of Eq. (5.8) are two standard Gaussian integrals which can be done exactly. The relevant calculations and the final expression for $I^{M_\pm}(\omega,q,t)=I_1^{M_\pm}(\omega,q)+I_2^{M_\pm}(\omega,q,t)$ with finite $\tau$ is lengthy and not particularly reader friendly. Thus we relegate the expressions to Appendix F, and discuss later in this section relevant approximations.

If any of the four species of the Majorana fermions has a contribution from the Ramond sector instead, we will have addition terms due to the presence of the $q=0$ mode. Let us take the example of $M_{+,q}$ with $q>0$, where we have

$$
\begin{aligned}
M_{\pm,q}^{\mathrm{R}} = M_{\pm,q}^{\mathrm{NS}} + i\frac{\Theta(m_t)}{2\sqrt{2}}&\left[\left(ic_0^2 - ic_0^{2\dagger}\right)\left(\alpha_q c_{-q}^{1\dagger}-\beta_q c_q^1\right)+\left(\alpha_q c_{-q}^{2\dagger}-\beta_q c_q^2\right)\left(ic_0^1 - ic_0^{1\dagger}\right)\right.\\
&\left.+\left(c_0^2 + c_0^{2\dagger}\right)\left(\alpha_q c_q^1 - \beta_q c_{-q}^{1\dagger}\right)+\left[\alpha_q c_q^2 - \beta_q c_{-q}^{2\dagger}\right]\left(c_0^1 + c_0^{1\dagger}\right)\right],
\end{aligned}
\tag{5.61}
$$

where $\Theta(x)$ is the Heaviside-step function, $\alpha_q, \beta_q$ are associated with mass $m_t$, and the superscripts label the species of the Majorana fermions. The extra terms do not involve summation over internal momentum, which subsequently leads to a Ramond sector correction on the order of $(1/L)$. Since we are concerned with the bulk response in the limit $L\to\infty$, such Ramond sector corrections can be safely ignored, and we can consider the results in the Neveu-Schwarz sector only.

Before presenting some useful approximations, we discuss several general features of the exact results. Firstly, a sanity check of our result is obtained by letting $M_{s,\text{pump}}, M_{t,\text{pump}} \to m_s, m_t$ where we expect to find a smooth crossover of our result for the pumped ladder to its equilibrium counterpart. In fact, as discussed in Appendix F, for small $\Delta m_{s/t} = M_{s/t,\text{pump}} - m_{s/t}$, we can expand $I^{M\pm}(\omega, q, t)$ in powers of $\Delta m_{s/t}$, and it turns out that the leading order correction to the unquenched result is $(\Delta m_{s/t})^2$. Secondly, $I^{M+}(\omega, q = 0, t)$ vanishes as $q^2$ as is appropriate for the small momentum limit of a (magnetization) current. On the other hand, $I^{M-}(\omega, q = 0, t)$ is generally nonzero for $\omega > |m_s^2 - m_t^2|^{1/2}$. Thirdly, the dependence of $I^{M\pm}(\omega, q, t)$ on the Majorana masses has the following relations:

$$I^{M+}(m_t, M_{t,\text{pump}}) = I^{M+}(-m_t, -M_{t,\text{pump}}),$$
$$I^{M-}(m_s, M_{s,\text{pump}}; m_t, M_{t,\text{pump}}) = I^{M-}(-m_s, -M_{s,\text{pump}}; -m_t, -M_{t,\text{pump}}), \qquad (5.62)$$
$$I^{M-}(m_s, M_{s,\text{pump}}; m_t, M_{t,\text{pump}}) = I^{M-}(m_t, M_{t,\text{pump}}; m_s, M_{s,\text{pump}}),$$

which can be seen directly from their respective formulas. Finally, the reflection symmetry of the ladder geometry requires the relation that $I^{M\pm}(\omega, q, t) = I^{M\pm}(\omega, -q, t)$. These relations should be kept in mind when we present our representative examples of these correlators for specific choices of parameters.

The non-equilibrium response consists of both a steady state part and a transient part:

$$I^{M\pm}(\omega, q, t) = I_1^{M\pm}(\omega, q) + I_2^{M\pm}(\omega, q, t). \qquad (5.63)$$

We discuss each separately below. For the steady state part, we are interested in its behavior in the limit $\tau \to \infty$. Under the condition that $|q| \ll |m_t| < |m_s|$ and $|u_q| \ll 1$, we obtain

$$I_1^{M+}(\omega, q) = f_+(\omega, q)\mathbf{1}_{\omega \in (-v|q|, v|q|)} + g_+(\omega, q)\mathbf{1}_{\omega \in (2|m_t|, \infty)}$$
$$I_1^{M-}(\omega, 0) = f_-^s(\omega)\mathbf{1}_{\omega \in (|m_t| - |m_s|, 0)} + f_-^t(\omega)\mathbf{1}_{\omega \in (0, |m_s| - |m_t|)} + g_-(\omega)\mathbf{1}_{\omega \in (|m_t| + |m_s|, \infty)} \qquad (5.64)$$

where $\mathbf{1}_{\omega \in A}$ is the indicator function that equal 1 when $\omega \in A$ and 0 otherwise, and the expressions for the coefficient functions are

$$f_+(\omega, q) \approx \frac{1}{16\pi^2} \frac{M^t(q_f^+)}{\left(1 + M^t(q_f^+)\right)^2} \frac{|m_t|}{v^2|q|},$$

$$g_+(\omega, q) \approx \frac{1}{128\pi^2} \frac{1}{\left(1 + M^t(q_g^+)\right)^2} \frac{|m_t|^2 q^2}{(\omega^2 - 3m_t^2)^{3/2} q_g^+},$$

$$f_-^s(\omega) \approx \frac{1}{64\pi^2} \frac{1}{1 + M^t(q_f^-)} \frac{M^s(q_f^-)}{1 + M^s(q_f^-)} \frac{(|m_s| + |m_t|)^2 q_f^-}{|m_s m_t (|m_s| - |m_t|)|}, \qquad (5.65)$$

$$f_-^t(\omega) \approx \frac{1}{64\pi^2} \frac{1}{1 + M^s(q_f^-)} \frac{M^t(q_f^-)}{1 + M^t(q_f^-)} \frac{(|m_s| + |m_t|)^2 q_f^-}{|m_s m_t (|m_s| - |m_t|)|},$$

$$g_-(\omega) \approx \frac{1}{16\pi^2} \frac{1}{1 + M^s(q_g^-)} \frac{1}{1 + M^t(q_g^-)} \frac{(|m_s| + |m_t|)^2 \left[\omega^4 - (m_s^2 - m_t^2)^2\right]}{4v^2\omega^5 q_g^-},$$

where the function $M(q)$ is equal to $u_q^* u_q$ but with its oscillatory part gone because of the limit $t_2 \to \infty$:

$$M^s(q) = \frac{2}{\cot^2 \Delta\theta_q^s + \tan^2 \Delta\theta_q^s}, \qquad M^t(q) = \frac{2}{\cot^2 \Delta\theta_q^t + \tan^2 \Delta\theta_q^t}, \qquad (5.66)$$

and the characteristic momenta are defined as

$$q_f^+ = \frac{|m_t||\omega|}{v\sqrt{v^2 q^2 - \omega^2}}, \quad q_g^+ = \frac{\sqrt{(\omega^2 - 4m_t^2)}}{2v}$$

$$q_f^- = q_g^- = \frac{\sqrt{[\omega^2 - (|m_s| + |m_t|)^2][\omega^2 - (|m_s| - |m_t|)^2]}}{2v|\omega|}. \tag{5.67}$$

The corresponding expressions for the unquenched ladder are obtained by setting $u_q = 0$:

$$I_1^{M_+}(\omega, q) = \frac{v|m_t|^2 q^2}{64\pi^2\sqrt{(\omega^2 - 3m_t^2)^3(\omega^2 - 4m_t^2)}} \mathbf{1}_{\omega \in (2|m_t|, \infty)},$$

$$I_1^{M_-}(\omega, 0) = \frac{(|m_s| + |m_t|)^2 \left[\omega^4 - (m_s^2 - m_t^2)^2\right]}{32\pi^2 v\omega^4 \sqrt{[\omega^2 - (|m_s| + |m_t|)^2][\omega^2 - (|m_s| - |m_t|)^2]}} \mathbf{1}_{\omega \in (|m_t| + |m_s|, \infty)}. \tag{5.68}$$

We can see that the pump transfers spectral weight from the two-particle threshold (which is of the square-root singularity form) to a region $\omega \in (-v|q|, v|q|)$ in case of $M_+$ mode and a region $\omega \in (|m_t| - |m_s|, |m_s| - |m_t|)$ in case of $M_-$ mode.

For the transient part, we look at $q \to 0$ and finite $\tau$ and find the following expressions:

$$I_2^{M_+}(\omega, q, t)$$

$$= \frac{\tau}{8\pi\sqrt{2\pi}} \int_0^{|q|} \frac{dq_1}{2\pi} \sin^2\left(\theta_{q_1}^t + \theta_{|q|-q_1}^t\right) \frac{\text{Re}\left[u_{|q|-q_1}^{t*} u_{q_1}^t\right] e^{-\frac{\tau^2}{4}\left[\left(\epsilon_{|q|-q_1}^t - \epsilon_{q_1}^t\right)^2 + \omega^2\right]}}{\left(1 + u_{q_1}^{t*} u_{q_1}^t\right)\left(1 + u_{|q|-q_1}^{t*} u_{|q|-q_1}^t\right)}$$

$$- \frac{\tau}{4\pi\sqrt{2\pi}} \int_0^\infty \frac{dq_1}{2\pi} \cos^2\left(\theta_{q_1}^t - \theta_{q_1+|q|}^t\right) \frac{\text{Re}\left[u_{q_1+|q|}^{t*} u_{q_1}^t\right] e^{-\frac{\tau^2}{4}\left[\left(\epsilon_{q_1+|q|}^t - \epsilon_{q_1}^t\right)^2 + \omega^2\right]}}{\left(1 + u_{q_1}^{t*} u_{q_1}^t\right)\left(1 + u_{q_1+|q|}^{t*} u_{q_1+|q|}^t\right)}, \tag{5.69}$$

$$I_2^{M_-}(\omega, 0, t)$$

$$= -\frac{\tau}{4\pi\sqrt{2\pi}} \int_0^\infty \frac{dq_1}{2\pi} \cos^2(\theta_{q_1}^s - \theta_{q_1}^t) \frac{\text{Re}\left[u_{q_1}^{s*} u_{q_1}^t\right] e^{-\frac{\tau^2}{4}\left[\left(\epsilon_{q_1}^s - \epsilon_{q_1}^t\right)^2 + \omega^2\right]}}{\left(1 + u_{q_1}^{s*} u_{q_1}^s\right)\left(1 + u_{q_1}^{t*} u_{q_1}^t\right)}.$$

Let's consider the $M_+$ mode first. Under the condition $q \to 0, |u_q| \ll 1$, its qualitative behavior can be seen from the following approximation:

$$I_2^{M_+}(\omega, q, t) \sim -I_{2,0}^{M_+} \frac{(m_t - M_{t,\text{pump}})^2}{8m_t^2 |M_{t,\text{pump}}|} \text{Re}\left[\int_0^\infty dq_1 \, v f(q_1, q, t)\right],$$

$$I_{2,0}^{M_+} = \frac{4\tau|m_t|}{8\pi^2 v\sqrt{2\pi}} e^{-\frac{\tau^2}{2}[v^2 q^2 + \omega^2]}, \tag{5.70}$$

$$f(q_1, q, t) = \frac{[1 - \cos(2\epsilon_{q_1}(M_{t,\text{pump}})t_1)]}{\frac{v^2 q_1^2}{|m_t M_{t,\text{pump}}|} + \frac{|m_t M_{t,\text{pump}}|}{v^2 q_1^2}} e^{-2i(\epsilon_{q_1+|q|}(m_t) - \epsilon_{q_1}(m_t))(t - t_1)},$$

where $I_{2,0}^{M_+}$ is a trivial factor depending on $\tau$. We use it to normalize the transient part such that its universal behavior for different choices of $\tau$ is more readily apparent. The function $f(q_1, q, t)$ has different behaviors at small $q_1$ and large $q_1$, so it is useful to split the integral into two parts:

$$J_1(q, t) = \int_0^{|m_t|/v} dq_1 \, v f(q_1, q, t), \quad J_2(q, t) = \int_{|m_t|/v}^\infty dq_1 \, v f(q_1, q, t). \tag{5.71}$$

Their asymptotic behavior at large $t$ can be estimated as:

$$J_1(q,t) \sim \frac{i|m_t|^3}{2|vqm_t M_{t,\mathrm{pump}}|} \frac{e^{-2\pi i(t-t_1)/\tau_{o,1}}}{t-t_1}, \quad \tau_{o,1} = \frac{\pi}{v|q| - v^2 q^2/|m_t|},$$

$$J_2(q,t) \sim -\frac{|m_t M_{t,\mathrm{pump}}|}{2} \sqrt{\frac{i\pi}{m_t^2 v|q|}} \frac{e^{-2\pi i(t-t_1)/\tau_{o,2}}}{\sqrt{t-t_1}}, \quad \tau_{o,2} = \frac{\pi}{v|q|}$$

(5.72)

We can see that both $J_1(q,t)$ and $J_2(q,t)$ decays algebraically, and they are oscillating with almost the same frequencies, producing the phenomenon of beats seen in Fig. 5.5:

$$\frac{|\tau_{o,1} - \tau_{o,2}|}{\tau_{o,1} + \tau_{o,2}} \ll 1.$$

(5.73)

Next let's consider the $M_-$ mode. A crude estimation shows that

$$\frac{I_2^{M_-}(\omega,0,t)}{I^{M_-}} < \frac{\tau\sqrt{|m_t m_s|}}{\sqrt{2\pi}} e^{-\frac{\tau^2}{4}\left[(|m_s|-|m_t|)^2+\omega^2\right]},$$

(5.74)

where $I^{M_-}$ is the amplitude of the steady state part shown in Eq. (5.10). For $\tau = 5$ and $\omega$ above the threshold, the right-hand side of Eq. (5.74) is orders of magnitude smaller than one: for example, $I_2^{M_-}/I^{M_-} < 0.02$ for $\omega = 2(|m_s|+|m_t|)$. Actually numerical evaluations of $I_2^{M_-}(\omega,0,t)$ above the threshold show that the left-hand side of Eq. (5.74) is further several orders of magnitude smaller.

# 6  Discussion and Conclusion

In this paper we have derived analytical expressions for the local antiferromagnetic correlations and the dynamical correlations that can in principle be measured in a time-resolved RIXS experiment on a Heisenberg-spin like ladder. This has been made possible by the fact that the low energy description of the Heisenberg spin ladder admits a representation in terms of non-interacting Majorana fermions.

We have found that the local antiferromagnetic correlations see fast relaxation accompanied by weak, damped oscillations both during and after the pump. The same quench protocol has been considered for several different models previously [53, 54], where fast relaxation is also observed. We have been able to express the associated timescales in terms of the singlet and triplet gaps which in turn are determined by the interleg, $J_\perp$, and the ring, $J_\times$, spin exchanges. For the dynamical correlations after the pump, we have computed both the smooth correlations, available exactly because of their associated fermion bilinear representations, and the staggered correlations, available approximately using the form factor approach as applied to the Majorana Ising spin and disorder operators. Both consist of a steady state and transient parts and present universal features controlled only by the Majorana mass parameters. They are independent of the pump and measurement times provided the pump time is longer than the saturation time. The frequency support of the spectral weight in the steady state response is changed qualitatively by the pump with equilibrium features both broadening and spectral weight appearing at energies forbidden in equilibrium. The transient responses show underdamped oscillations long after the pump. Both types of responses provide unambiguous characteristics of the pump.

Previous work on the analysis of time-resolved pump-probe experiments have employed time-dependent density functional theory [55], time-dependent dynamical mean field theory [27–30], time-dependent density renormalization group theory [56–58], and

time-dependent exact diagonalization [22–26]. However they have been applied to cases involving Hubbard models where the pump photo-dopes the system. With such photo-doping, it is necessary to take into account both the charge and spin degrees of freedom. In contrast, we consider a pump that adjusts the interleg spin exchange in the ladder, the possibility of which was envisioned in [38], and thus we focus on the spin degrees of freedom only. When charge degrees of freedom are present, the magnetic order is melted by transferring spectral weight to doublon-holon pairs and midgap single-particle excitations [22, 23, 28, 58], while when it is absent, the melting instead goes through the channel of transient spinons and magnons [16, 17]. In comparison with these findings for the Hubbard model, our results indicate that the spin channel features a much faster magnetic relaxation. This is consistent with the observation in [16], where the magnetic excitation dominated intra-plane magnetic relaxation is much faster than the non-magnetic excitation dominated inter-plane magnetic relaxation.

By pumping only the spin degrees of the freedom, we are unable to observe certain spin-charge interplay phenomena. As one obvious example, we are not able to investigate the local superexchange mediated pairing between holes [59]. A more subtle example here concerns the effect on the magnetic excitations. All magnetic excitations in our model are dispersive, in contrast to the observation of magnon localization in [22, 60]. DMRG computations in Ref. [60] suggest that the flattening of the magnon dispersion can be realized by charge doping outside the ladder plane. On the other hand, the exact diagonalization calculations in [22] suggest that a similar effect can be produced by allowing a sufficiently long core hole lifetime. The absence of magnon localization in our model is thus consistent with these observations, since our pump does not photo-dope and we are working in the ultra-short core hole lifetime limit.

Our analytical expressions and numerical demonstrations showed that the magnetic correlations are melted not by an incoherent heating process and so cannot be characterized by an effective temperature. Clear and direct evidence of this is provided by the mode occupation shown in Fig. 4.2 and 4.1, where the mode occupation presents a peak at a relatively higher energy and a highly nonthermal distribution. This is in agreement with the previously findings [16, 17, 23], indicating that the nonthermality is a common feature for dynamics after a pump.

A key conclusion from our results is that the transient dynamics can be described by a few effective parameters characterizing the pump. Such universality is in line with other theoretical approaches [23, 30, 61]. In [61], the pump is smoothly centered around the frequency $\Omega$ such that the transient dynamics can be described in terms of the difference between the probing frequency $\omega$ and the central pumping frequency $\Omega$, and this universality persists in the presence of electron-electron interaction. In [23, 30], a similar pump protocol is used, and the transient dynamics can be explained by the first few non-equilibrium Floquet states. In the case of our pump, a steplike double quench, the transient dynamics can be described by the singlet and triplet gaps, or equivalently, the Majorana mass parameters.

The ladder geometry adopted in our model presents several unique features compared with both the chain geometry and plane geometry. If we quench the ladder from the rung singlet phase (see Fig. 2.1), then the correlations in the $N_-$ sector (namely, around the $(\pi, \pi)$ point of the Brillouin zone) are characterized by a single particle singularity. This is shown in the upper left panel of Fig. 5.1, where the single particle pole in the equilibrium system is transformed into a single-particle threshold by the quench. This is in sharp contrast to the chain geometry, where spinons are always created in pairs and we only see a two-particle continuum in the low lying spectrum [62]. On the other hand, the ladder geometry limits us from seeing genuine two-dimensional phenomena. One such

example is the antiphase oscillation between two orthogonal momentum directions parallel and perpendicular to the electric field of the pump pulse observed in [25,57]. We do not see such oscillations because the ladder geometry lacks $\pi/4$ rotation symmetry, and thus the $(0,\pi)$ and $(\pi,0)$ points of the Brillouin zone correspond to different operators, namely $M_-$ and $N_+$ respectively, and their transient oscillations have different periods.

One advantage that our theoretical approach offers over competing techniques is that we are not limited by either the frequency/momentum resolution or the system size. The price that we pay is we can focus only on the spin degrees of freedom in our ladder model. Nevertheless, we are able to provide a full analytical analysis over the post-pump steady state behaviour and are able to identify universal features in the response that depend only on a few effective parameters. This is particularly useful as pump-probe experiments have a plethora of experimentally specified tuning knobs. Our results thus provide an analytical benchmark for both numerical simulations and potential future experiments performed on ladder materials, both strontium iridium oxides [31,39] and artificially designed cold atom systems [40,41]. Our results are expected to complement and to be compared against the understanding of the time resolved pump-probe signals coming from other fluctuation channels such as charge, orbital and lattice [16,17,21,63–65]. We plan to perform such comparative work in the immediate future.

## Acknowledgements

We would like to acknowledge helpful conversations with Mark Dean, Andrew James, and Elie Merhej. T.R. and R.M.K. was supported by the U.S. Department of Energy, Office of Basic Energy Sciences, under Contract No. DE-SC0012704.

## A    Analysis of Several Integrals from Section 4.1

In Sec. 4.1, we encountered the following integral from Eq. (4.2):

$$I(t_1) \equiv \int_0^\infty \frac{\mathrm{d}q}{2\pi} \frac{2\epsilon_q(m)u_q^* u_q}{1 + u_q^* u_q} R(q), \tag{A.1}$$

where $R(q)$ is some regulator to make the integral convergent, the function $u_q$ is defined in Eq. (3.10) and the combination $u_q^* u_q$ is the following:

$$
\begin{aligned}
u_q^* u_q &= \frac{2 - 2\cos\left(2\epsilon_q(M_{\text{pump}})t_1\right)}{\left(\cot \Delta\theta_q\right)^2 + \left(\tan \Delta\theta_q\right)^2 + 2\cos\left(2\epsilon_q(M_{\text{pump}}t_1)\right)}, \\
\Delta\theta_q &= \frac{1}{2}\left(\arctan \frac{m}{vq} - \arctan \frac{M_{\text{pump}}}{vq}\right).
\end{aligned}
\tag{A.2}
$$

We are interested in the qualitative behaviors of this integral, which is controlled by the small and large $q$ behaviors of the integrand. For small $q$, we have

$$
\begin{aligned}
\Delta\theta_q &\approx \frac{1}{2}\left(\frac{\pi}{2}\operatorname{sgn} m - \frac{vq}{m} - \frac{\pi}{2}\operatorname{sgn} M_{\text{pump}} + \frac{vq}{M_{\text{pump}}}\right) \\
\Rightarrow \quad u_q^* u_q &\approx \frac{(m - M_{\text{pump}})^2}{2\,(mM_{\text{pump}})^2} \frac{\epsilon_q(m)\left[1 - \cos(2\epsilon_q(M_{\text{pump}})t_1)\right]}{1/v^2 q^2}.
\end{aligned}
\tag{A.3}
$$

For large $q$, we have

$$\Delta\theta_q \approx \frac{1}{2}\left(\frac{m}{vq} - \frac{M_{\text{pump}}}{vq}\right)$$
$$\Rightarrow\ u_q^* u_q \approx \frac{(m - M_{\text{pump}})^2}{2}\frac{\epsilon_q(m)\left[1 - \cos(2\epsilon_q(M_{\text{pump}})t_1)\right]}{v^2 q^2}. \tag{A.4}$$

The small $q$ and large $q$ behaviors shown above captured by the interpolation in the following expression:

$$u_q^* u_q \approx \frac{(m - M_{\text{pump}})^2}{2|mM_{\text{pump}}|}\frac{\epsilon_q(m)\left[1 - \cos(2\epsilon_q(M_{\text{pump}})t_1)\right]}{\left(\frac{vq}{\sqrt{|mM_{\text{pump}}|}}\right)^2 + \left(\frac{\sqrt{|mM_{\text{pump}}|}}{vq}\right)^2}, \tag{A.5}$$

which leads to the following approximation for $I(t_1)$:

$$I(t_1) \approx \int_0^\infty \frac{dq}{2\pi}\frac{(m - M_{\text{pump}})^2}{|mM_{\text{pump}}|}\frac{\epsilon_q(m)\left[1 - \cos(2\epsilon_q(M_{\text{pump}})t_1)\right]}{\left(\frac{vq}{\sqrt{|mM_{\text{pump}}|}}\right)^2 + \left(\frac{\sqrt{|mM_{\text{pump}}|}}{vq}\right)^2}R(q). \tag{A.6}$$

Several conclusions can be drawn from this approximation. Firstly, we can see that the pump energy is perturbative in the change of the masses from equilibrium to pump:

$$I(t_1) \propto (m - M_{\text{pump}})^2 \propto (J_{\perp,1} - J_{\perp,2})^2. \tag{A.7}$$

Secondly, we can see that without the regulator, the integral is logarithmically divergent at large $q$, so the application of a proper regulator $R(q)$ is necessary for its evaluation. We choose the following regulator in the actual numerical evaluations:

$$R(q) = \frac{q_{\text{max}}^2}{q^2 + q_{\text{max}}^2}, \tag{A.8}$$

where $q_{\text{max}}$ is chosen according to the mode occupation shown in Fig. 4.2 such that only the modes below $q < q_{\text{max}}$ are significantly populated. Finally, we can estimate the time scale for the initial rapid growth of the pumped energy as

$$t_{\text{saturation}} \sim \frac{1}{2\sqrt{|M_{\text{pump}}|(|m| + |M_{\text{pump}}|)|}}, \tag{A.9}$$

and the subsequent decaying oscillation as follows

$$I(t_1 \gg t_{\text{saturation}}) \sim \text{const} - \frac{(m - M_{\text{pump}})^2}{2\pi v}\,\text{Ci}(t_1/t_{\text{saturation}}), \tag{A.10}$$

where $\text{Ci}(x)$ is the cosine integral whose asymptotic behavior at large argument is

$$\text{Ci}(x \gg 1) \sim -\frac{\cos x}{x^2} + \frac{\sin x}{x}. \tag{A.11}$$

As a result, the oscillation decays algebraically and has a time scale $2\pi t_{\text{saturation}}$. This behavior can be seen in Fig. 4.1 in the main text.

In Sec. 4.2, we encountered the following integrals from Eq. (4.11):

$$I_1(t', t) = \int_0^\infty \frac{dq}{2\pi}\frac{1}{1 + u_q^* u_q}\left[\frac{m}{\epsilon_q(m)}\left(1 - u_q^* u_q\right) - \frac{2vq}{\epsilon_q(m)}\,\text{Im}[u_q]\right]R_1(q),$$
$$I_2(t', t) = \int_0^\infty \frac{dq}{2\pi}\frac{1}{1 + u_q^* u_q}\left[\frac{2vq^2}{\epsilon_q(m)}\left(1 - u_q^* u_q\right) + \frac{4mq}{\epsilon_q(m)}\,\text{Im}[u_q]\right]R_2(q), \tag{A.12}$$

where $R_1(q)$ and $R_2(q)$ are some regulators to make the integrals convergent. For evaluation during the pump, we need $I_1(t,t)$ and $I_2(t,t)$ with $t < t_1$; while for evaluation after the pump, we need $I_1(t_1,t)$ and $I_2(t_1,t)$ with $t > t_1$. It is useful to repeat the expression for $u_q$ here:

$$u_q\left(m, M_{\text{pump}}, t', t\right) = i \frac{e^{-2i\epsilon_q(m)(t-t')}\left(1 - e^{-2i\epsilon_q(M_{\text{pump}})t'}\right)}{\cot \Delta\theta_q + \tan \Delta\theta_q e^{-2i\epsilon_q(M_{\text{pump}})t'}}. \tag{A.13}$$

It can be approximated as

$$u_q \approx \begin{cases} i\frac{(m-M_{\text{pump}})}{2\sqrt{|mM_{\text{pump}}|}} \frac{e^{-2i\epsilon_q(m)\left(t-t'\right)}\left(1-e^{-2i\epsilon_q(M_{\text{pump}})t'}\right)}{\left(\frac{vq}{\sqrt{|mM_{\text{pump}}|}}\right)+\left(\frac{\sqrt{|mM_{\text{pump}}|}}{vq}\right)} & \text{if } mM_{\text{pump}} > 0, \\[3mm] i\frac{(m-M_{\text{pump}})}{2\sqrt{|mM_{\text{pump}}|}} \frac{e^{-2i\epsilon_q(m)\left(t-t'\right)}\left(1-e^{-2i\epsilon_q(M_{\text{pump}})t'}\right)}{\left(\frac{vq}{\sqrt{|mM_{\text{pump}}|}}\right)+\left(\frac{\sqrt{|mM_{\text{pump}}|}}{vq}\right)e^{-2i\epsilon_q(M_{\text{pump}})t'}} & \text{if } mM_{\text{pump}} < 0. \end{cases} \tag{A.14}$$

We can see that $u_q$ is regular at small $q$, and it diverges as $1/q$ at large $q$. As a result, both integrals in Eq. (A.12) are regular at small $q$, while $I_1(t',t)$ diverges logarithmically and $I_1(t',t)$ diverges quadratically at large $q$. Correspondingly, we choose the following regulator in the actual numerical evaluations:

$$R_1(q) = \frac{q_{\text{max}}^2}{q^2 + q_{\text{max}}^2}, \quad R_2(q) = \left(\frac{q_{\text{max}}^2}{q^2 + q_{\text{max}}^2}\right)^2. \tag{A.15}$$

Alternatively we might have chosen more simply $R_{1,2} = \Theta(q_{\text{max}} - |q|)$, but this would have led to artificial ringing. Approximations of the two integrals in Eq. (A.12) can be made in a similar way to that in Eq. (A.6), and the relevant time scales can be estimated. For the evaluation during the pump, we have $t' = t < t_1$, while for the evaluation after the pump, we have $t' = t_1, t > t_1$. The time scales for the initial rapid change with respect to $t$ differ in these two situations:

$$\begin{aligned} t_{\text{during pump}} &\sim \frac{1}{2\sqrt{|M_{\text{pump}}|(|m| + |M_{\text{pump}}|)}}, \\ t_{\text{after pump}} &\sim \frac{1}{2\sqrt{|m|(|m| + |M_{\text{pump}}|)}}. \end{aligned} \tag{A.16}$$

If the pump time $t_1 \gg t_{\text{during pump}}$, we will see decaying oscillation with time constant $2\pi t_{\text{during pump}}$. For a post-pump time $(t - t_1) \gg t_{\text{after pump}}$, we will see decaying oscillations with a time constant $2\pi t_{\text{after pump}}$. As for the periods of these decaying oscillations, they are determined by factoring out the $q$-independent yet $t$-dependent phase factor in $u_q$ shown in Eq. (A.13:

$$t_{\text{oscillation}} \sim \pi/|m|. \tag{A.17}$$

This behavior can be seen clearly in Fig. 4.3 and 4.4.

# B    Details on Calculating $\mathcal{I}_\sigma$

In this Appendix, we present details of the calculation of the staggered correlations. We follow the approach used in [37]. The case of the ordered phase with $m > 0$ already presents all of the nontrivial aspects of calculations. We thus confine ourselves to it here. The expressions that we encounter here are singular due to the annihilation poles encoded

in the $\coth[(\vartheta_k - \vartheta_p)/2]$ factors that appear in the form factors. For finite systems, since $k$ and $p$ sit in the Neveu-Schwarz and Ramond sector respectively, the annihilation poles are always avoided by $1/L$ factors due to the different quantization rules for momentum in the two sectors. Thus the singularities manifest themselves as terms diverging with system size.

The most singular terms of $\mathcal{MI}_\sigma$ are those with most even pairing between the momenta from the Neveu-Schwarz sector and those from the Ramond sector. In the case of $m > 0$, the leading contributions at order $u^{2n}$ are $C_{(2n|2n|2n)}$. To get a sense of this, let us take the simplest example of order $u^0$ contributions $C_{(0|2s|0)}$. For $s = 0$, we have

$$C_{(0|0|0)} = \bar{\sigma}^2. \tag{B.1}$$

For $s = 1$, we have

$$
\begin{aligned}
C_{(0|2|0)} &= \frac{v^2 \bar{\sigma}^2}{2(m)^2} \iint_{-\infty}^{\infty} \frac{\mathrm{d}p_1 \mathrm{d}p_2}{(2\pi)^2} \frac{e^{ix(p_1+p_2)} e^{-2i\tilde{t}(\epsilon_{p_1}+\epsilon_{p_2})}}{\cosh \vartheta_{p_1} \cosh \vartheta_{p_2}} \tanh^2 \left( \frac{\vartheta_{p_1} - \vartheta_{p_2}}{2} \right) \\
&= \frac{\bar{\sigma}^2}{2} \iint_{-\infty}^{\infty} \frac{\mathrm{d}\vartheta_1 \mathrm{d}\vartheta_2}{(2\pi)^2} e^{i\frac{|m|x}{v}(\sinh \vartheta_1 + \sinh \vartheta_2)} e^{-2i|m|\tilde{t}(\cosh \vartheta_1 + \cosh \vartheta_2)} \tanh^2 \left( \frac{\vartheta_1 - \vartheta_2}{2} \right) \\
&= \bar{\sigma}^2 \iint_{-\infty}^{\infty} \frac{\mathrm{d}\vartheta_+ \mathrm{d}\vartheta_-}{(2\pi)^2} e^{2i\frac{|m|x}{v} \sinh \vartheta_+ \cosh \vartheta_-} e^{-4i|m|\tilde{t} \cosh \vartheta_+ \cosh \vartheta_-} \tanh^2 \vartheta_-,
\end{aligned}
\tag{B.2}
$$

where we have converted the summations over momentum into integrals in the limit $L \to \infty$. This can be done without problem since the integrand is regular.

We will work in the limit where $|x| \gg v|\tilde{t}|$. Then for $|mx|/v \gg 1$, $C_{(0|2|0)} \sim (v/|mx|)e^{-2|mx|/v}$ is exponentially suppressed. The same analysis can be performed for $C_{(0|2s|0)}$ with $s > 1$. We now proceed to calculate the leading contribution at $\mathcal{O}(u^2)$ coming from $C_{(2|2|2)}$ in Eq. (5.27). Generally, for a summand $f(p), p \in \mathbb{R}$ containing isolated poles and branch cuts along the imaginary axis, we have

$$
\begin{aligned}
\frac{1}{L} \sum_p f(p) &= \int_C \frac{\mathrm{d}p}{2\pi} \frac{f(p)}{e^{ipL} - 1} + \text{corrections}_1 \\
&= -\int_C \frac{\mathrm{d}p}{2\pi} \frac{f(p)}{e^{-ipL} - 1} + \text{corrections}_2,
\end{aligned}
\tag{B.3}
$$

where the contour $C$ is shown in the left panel of Fig. B.1 and the corrections take into account the residuals of the integrands at the isolated poles of $f(p)$:

$$
\begin{aligned}
\text{correction}_1 &= -\frac{2\pi i}{2\pi} \sum_{\text{poles of } f(p)} \text{Res} \left[ \frac{f(p)}{e^{ipL} - 1} \right], \\
\text{correction}_2 &= \frac{2\pi i}{2\pi} \sum_{\text{poles of } f(p)} \text{Res} \left[ \frac{f(p)}{e^{-ipL} - 1} \right].
\end{aligned}
\tag{B.4}
$$

The summations over $p_1$ and $p_2$ in Eq. (5.27) can be carried out using Eq. (B.3). For $x > 0$, we use the top equation in Eq. (B.3), while for $x < 0$, we use the bottom equation. Because we are treating the case where $|x| \gg v|\tilde{t}|$, for both $x > 0$ and $x < 0$, the lower branch of $C$ can be deformed into the lower-half plane while the upper branch of $C$ can be bent to the upper-half plane (see right panel of Fig. B.1), both going around the branch cuts and making the integrals exponentially small, either in $|x|$ or $(L - |x|)$. As a

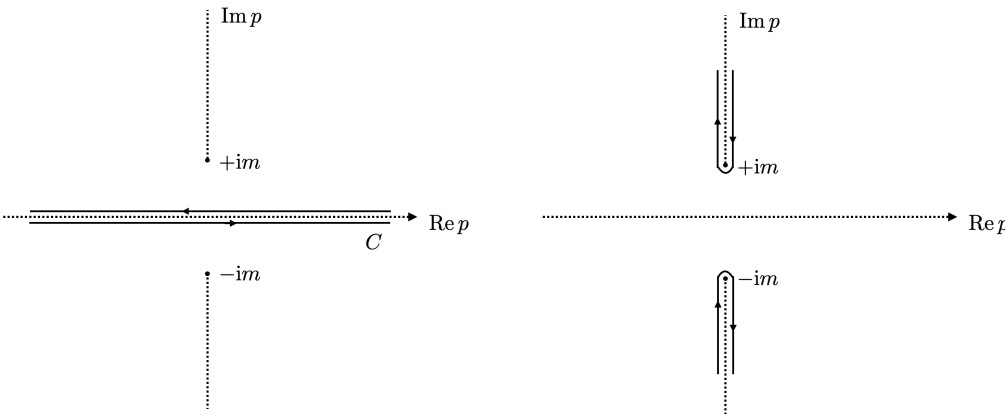

Figure B.1: The choice of contour in Eq. (B.3).

demonstration, let's take the example of $x > 0$, and write $f(p) = g(\epsilon_p)e^{ipx}$, we then have

$$\int_C \frac{\mathrm{d}p}{2\pi} \frac{f(p)}{e^{ipL} - 1} = \left( \int_{+i\infty+0^+}^{im+0^+} + \int_{im-0^-}^{i\infty-0^-} + \int_{-i\infty-0^+}^{-im-0^+} + \int_{-im+0^+}^{-i\infty+0^+} \right) \frac{\mathrm{d}p}{2\pi} \frac{g(\epsilon_p)e^{ipx}}{e^{ipL} - 1}$$

$$= \int_m^\infty i \frac{\mathrm{d}q}{2\pi} \frac{g\left(-i\sqrt{q^2 - m^2}\right) - g\left(i\sqrt{q^2 - m^2}\right)}{e^{-qL} - 1} e^{-qx} \qquad (B.5)$$

$$- \int_m^\infty i \frac{\mathrm{d}q}{2\pi} \frac{g\left(-i\sqrt{q^2 - m^2}\right) - g\left(i\sqrt{q^2 - m^2}\right)}{e^{qL} - 1} e^{qx},$$

It is easy to see that the first term is exponentially small in $|x|$, and the second term is exponentially small in $(L - |x|)$. In the end, the summations over $p_1$ and $p_2$ in Eq. (5.27) are dominated by the corresponding corrections in Eq. (B.4), with the result:

$$C_{(2|2|2)} = C_{(2|2|2)}^{(1)} + C_{(2|2|2)}^{(2)},$$

$$C_{(2|2|2)}^{(1)} = \bar{\sigma}^2 \sum_{\substack{k,q>0\in\mathrm{NS} \\ k\neq q}} \frac{v^2 u_k^*(t_2)u_q(t_2)\tanh\vartheta_k\tanh\vartheta_q}{|m|^2 L^2 \cosh\vartheta_k \cosh\vartheta_q} \coth^2\left(\frac{\vartheta_q + \vartheta_k}{2}\right) \coth^2\left(\frac{\vartheta_q - \vartheta_k}{2}\right)$$

$$+\bar{\sigma}^2 \sum_{\substack{k,q>0\in\mathrm{NS} \\ k\neq q}} \frac{v^2 u_k^*(t_2)u_q(t_2)e^{i\tilde{t}(2\epsilon_k - 2\epsilon_q)}\tanh\vartheta_q\tanh\vartheta_k}{|m|^2 L^2 \cosh\vartheta_k \cosh\vartheta_q} \coth^2\left(\frac{\vartheta_k + \vartheta_q}{2}\right) \coth^2\left(\frac{\vartheta_k - \vartheta_q}{2}\right)$$

$$+\bar{\sigma}^2 \sum_{\substack{k,q>0\in\mathrm{NS} \\ k\neq q}} \frac{v^2 u_k^*(t_2)u_q(t_2)e^{i\tilde{t}(\epsilon_k - \epsilon_q)}\cos\left(x(q+k)\right)}{|m|^2 L^2 \cosh\vartheta_k \cosh\vartheta_q} \coth^2\left(\frac{\vartheta_k + \vartheta_q}{2}\right)$$

$$-2\bar{\sigma}^2 \sum_{\substack{k,q>0\in\mathrm{NS} \\ k\neq q}} \frac{v^2 u_k^*(t_2)u_q(t_2)e^{i\tilde{t}(\epsilon_k - \epsilon_q)}\cos\left(x(q-k)\right)}{|m|^2 L^2 \cosh\vartheta_k \cosh\vartheta_q} \coth^2\left(\frac{\vartheta_k - \vartheta_q}{2}\right),$$

$$C_{(2|2|2)}^{(2)} = \bar{\sigma}^2 \sum_{k>0\in\mathrm{NS}} u_k^*(t_2)u_k(t_2) \cdot \left[1 - \frac{4}{L}|x|\right]. \qquad (B.6)$$

We now evaluate the summation over $q$ in $C_{(2|2|2)}^{(1)}$.

To do so, we convert the sum to an integral along the contour $C_0$ (see Fig B.2). For $t_2 + t_2' > 0$, we define $C_0$ by deforming into the lower half plane around the pole at

$q = k$. When all terms in $C_{(2|2|2)}^{(1)}$ are summed, the pole terms vanish but we specify this deformation so that we can treat the four terms in $C_{(2|2|2)}^{(1)}$ individually. For the first two

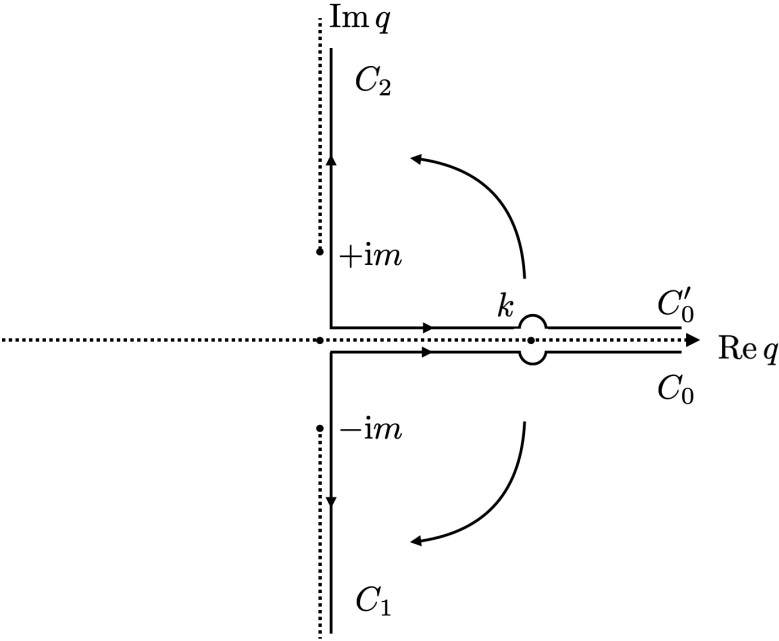

Figure B.2: The contours of the integral representing the summations in $C_{(2|2|2)}^{(1)}$.

terms in $C_{(2|2|2)}^{(1)}$, we deform the integrals into the lower-half plane (see the contour $C_1$ in Fig. B.2) as they contain the factor $e^{-i\epsilon_q(m)(t_2+t_2')}$. It can be shown that both of them are algebraically suppressed in terms of $t_2 + t_2'$. The case for $t_2 + t_2' < 0$ can be analyzed analogously, only that we should use the contour $C_0'$ instead of $C_0$ in Fig. B.2, and deform it into the upper-half plane (see the contour $C_2$ in Fig. B.2).

The last two terms of $C_{(2|2|2)}^{(1)}$ contain the factor

$$e^{-i\epsilon_q(m)(t_2+t_2')\pm iqx}. \tag{B.7}$$

We consider the case $t_2 + t_2' > 0$ first. For $\epsilon_q'(t_2 + t_2') \mp x > 0$, we can still deform the integrals into the lower-half plane (see the contour $C_1$ in Fig. B.2) and obtain sub-leading contributions. But for $\epsilon_q'(t_2 + t_2') \mp x < 0$, we have to deform the integrals into the upper-half plane (see the contour $C_2$ in Fig. B.2). We deform about the double pole at $q = k$ and taking this into account produces the dominant contribution:

$$
\begin{aligned}
C_{(2|2|2)}^{(1)} = &- \frac{4\bar{\sigma}^2}{L} \sum_{k>0\in\text{NS}} u_k^*(t_2)u_k(t_2)\Theta\left(-\epsilon_k'(t_2+t_2')+x\right)\left[\epsilon_k'(t_2+t_2')-x\right] \\
&- \frac{4\bar{\sigma}^2}{L} \sum_{k>0\in\text{NS}} u_k^*(t_2)u_k(t_2)\Theta\left(-\epsilon_k'(t_2+t_2')-x\right)\left[\epsilon_k'(t_2+t_2')+x\right].
\end{aligned}
\tag{B.8}
$$

The other case $t_2 + t_2' < 0$ can be analyzed analogously by replacing the contour $C_0$ with $C_0'$ in Fig. B.2. For $\epsilon_q'(t_2 + t_2') \mp x < 0$, we can deform it into the upper-half plane and obtain sub-leading contributions, but for $\epsilon_q'(t_2 + t_2') \mp x > 0$, we have to deform it into the lower-half plane and take into account the contribution from the double pole at $q = k$.

We then arrive at

$$
\begin{aligned}
C_{(2|2|2)}^{(1)} = & -\frac{4\bar{\sigma}^2}{L} \sum_{k>0 \in \text{NS}} u_k^*(t_2) u_k(t_2) \Theta\left(\epsilon_k'(t_2 + t_2') - x\right)\left[-\epsilon_k'(t_2 + t_2') + x\right] \\
& -\frac{4\bar{\sigma}^2}{L} \sum_{k>0 \in \text{NS}} u_k^*(t_2) u_k(t_2) \Theta\left(\epsilon_k'(t_2 + t_2') + x\right)\left[-\epsilon_k'(t_2 + t_2') - x\right].
\end{aligned}
\tag{B.9}
$$

In summary, we have

$$
\begin{aligned}
C_{(2|2|2)}^{(1)} = & -\frac{4\bar{\sigma}^2}{L} \sum_{k>0 \in \text{NS}} u_k^*(t_2) u_k(t_2) \Theta\left(-\epsilon_k'|t_2 + t_2'| + x\right)\left[\epsilon_k'|t_2 + t_2'| - x\right] \\
& -\frac{4\bar{\sigma}^2}{L} \sum_{k>0 \in \text{NS}} u_k^*(t_2) u_k(t_2) \Theta\left(-\epsilon_k'|t_2 + t_2'| - x\right)\left[\epsilon_k'|t_2 + t_2'| + x\right] \\
= & -\frac{4\bar{\sigma}^2}{L} \sum_{k>0 \in \text{NS}} u_k^*(t_2) u_k(t_2) \Theta\left(-\epsilon_k'|t_2 + t_2'| + |x|\right)\left[\epsilon_k'|t_2 + t_2'| - |x|\right].
\end{aligned}
\tag{B.10}
$$

Summing $C_{(2|2|2)}^{(1)}$ and $C_{(2|2|2)}^{(2)}$, we finally arrive at the result in Eq. (5.28), which is valid under the conditions $|x| \gg v|\tilde{t}|$, $|mx|/v \gg 1$, and $|m(t_2 + t_2')| \gg 1$.

As described in Sec. 5.2.1, for $\mathcal{I}_\sigma(m > 0)$ in the ordered phase, we can sum all the leading contributions and arrive at Eq. (5.30):

$$
\begin{aligned}
\mathcal{I}_\sigma(m > 0) &\approx \bar{\sigma}^2 \exp\left[-\frac{2}{\pi}\int_0^\infty \mathrm{d}k \; u_k^*(t_2) u_k(t_2) \mathcal{G}(k, x, t_2 + t_2')\right], \\
\mathcal{G}(k, x, t_2 + t_2') &= |x| + \Theta\left(-\epsilon_k'|t_2 + t_2'| + |x|\right)\left[\epsilon_k'|t_2 + t_2'| - |x|\right].
\end{aligned}
\tag{B.11}
$$

On the other hand, as described in Sec. 5.2.2, for $\mathcal{I}_\sigma(m < 0)$ in the disordered phase, we can sum all the leading contributions and arrive at Eq. (5.38):

$$
\begin{aligned}
\mathcal{I}_\sigma(m < 0) &\approx \Sigma_1 \exp\left[-\frac{2}{\pi}\int_0^\infty \mathrm{d}k \; u_k^*(t_2) u_k(t_2)|x|\right] \\
&\quad + \Sigma_2 \exp\left[-\frac{2}{\pi}\int_0^\infty \mathrm{d}k \; u_k^*(t_2) u_k(t_2) \mathcal{G}(k, x, t_2 + t_2')\right], \\
\Sigma_1 &= \bar{\sigma}^2 v \int_{-\infty}^\infty \frac{\mathrm{d}p}{2\pi} \frac{e^{-i\tilde{t}\epsilon_p + ipx}}{\epsilon_p}, \quad \Sigma_2 = 4\bar{\sigma}^2 v \int_0^\infty \frac{\mathrm{d}k}{2\pi} \frac{\sin(k|x|)}{\epsilon_k} \mathrm{Im}\left[u_k(t_2)e^{-i\epsilon_k \tilde{t}}\right].
\end{aligned}
\tag{B.12}
$$

It is useful to look at the Fourier transform of $\mathcal{I}_\sigma(m)$ in the limit $t_2 \to \infty$ and in the two phases respectively. Let us look at $\mathcal{I}_\sigma(m > 0)$ first:

$$
\lim_{t_2 \to \infty} \int \mathrm{d}(t_2' - t_2) \int \mathrm{d}x \; e^{i\omega(t_2 - t_2') - iqx} \mathcal{I}_\sigma(m > 0).
\tag{B.13}
$$

Since the weight in the above integral occurs near $t_2 \sim t_2'$ and $x \sim 0$, in the limit $t_2 \to \infty$ we can take $|x| \ll |t_2 + t_2'|$ such that $\mathcal{G}(k, x, t_2 + t_2') \to |x|$. Consequently, we have

$$
\begin{aligned}
&\lim_{t_2 \to \infty} \int \mathrm{d}(t_2' - t_2) \int \mathrm{d}x \; e^{i\omega(t_2 - t_2') - iqx} \mathcal{I}_\sigma(m > 0) \\
&= \lim_{t_2 \to \infty} \int \mathrm{d}x \int \mathrm{d}t_2' \; e^{i\omega(t_2 - t_2') - iqx} \bar{\sigma}^2 \exp\left[-\frac{2}{\pi}\int \mathrm{d}k \; u_k^*(t_2) u_k(t_2)|x|\right] \propto \delta(\omega),
\end{aligned}
\tag{B.14}
$$

find the response at leading order is a $\delta$-function in $\omega$.

The Fourier transform of $\mathcal{I}_\sigma(m < 0)$ can be analyzed analogously. We have

$$
\int \mathrm{d}t'_2 \int \mathrm{d}x \; e^{i\omega(t_2 - t'_2) - iqx} \mathcal{I}_\sigma(m > 0) \propto \delta(\omega),
$$
$$
\int \mathrm{d}t'_2 \int \mathrm{d}x \; e^{i\omega(t_2 - t'_2) - iqx} \mathcal{I}_\sigma(m < 0) \propto \int \mathrm{d}p \; \delta(\omega - \epsilon_p) f(p),
$$
(B.15)

where the function $f(p)$ takes into account the residual dependence on $p$. This result can be used to infer the thresholds for the RIXS spectral functions. For $N_- \sim \sigma^0 \mu^1 \mu^2 \sigma^3$, we then have

$$
\int \mathrm{d}t'_2 \int \mathrm{d}x \; e^{i\omega(t_2 - t'_2) - iqx} \mathcal{I}_\sigma(m_s > 0) \mathcal{I}_\mu(m_t < 0) \mathcal{I}_\mu(m_t < 0) \mathcal{I}_\sigma(m_t < 0)
$$
$$
= \int \mathrm{d}t'_2 \int \mathrm{d}x \; e^{i\omega(t_2 - t'_2) - iqx} \mathcal{I}_\sigma(m_s > 0) \mathcal{I}_\sigma(-m_t > 0) \mathcal{I}_\sigma(-m_t > 0) \mathcal{I}_\sigma(m_t < 0)
$$
$$
\sim \int \prod_{i=1}^{4} \mathrm{d}\omega_i \; \delta(\omega - \omega_1 - \omega_2 - \omega_3 - \omega_4) \int \mathrm{d}p \; \delta(\omega_1) \delta(\omega_2) \delta(\omega_3) \delta(\omega_4 - \epsilon_p^t) f^t(p)
$$
$$
\sim \int \mathrm{d}p \; \delta(\omega - \epsilon_p^t) f^t(p)
$$
(B.16)

The unquenched system has $f^t(p) \propto \delta(p - q)$, so the spectral function is proportional to a delta function $\delta(\omega - \epsilon_q^t)$. The quenched system broadens this delta function and so the spectral function has a threshold at $\omega = |m_t|$.

For the case $N_+ \sim \mu^0 \sigma^1 \sigma^2 \mu^3$, we have

$$
\int \mathrm{d}t'_2 \int \mathrm{d}x \; e^{i\omega(t_2 - t'_2) - iqx} \mathcal{I}_\mu(m_s > 0) \mathcal{I}_\sigma(m_t < 0) \mathcal{I}_\sigma(m_t < 0) \mathcal{I}_\mu(m_t < 0)
$$
$$
= \int \mathrm{d}t'_2 \int \mathrm{d}x \; e^{i\omega(t_2 - t'_2) - iqx} \mathcal{I}_\sigma(-m_s < 0) \mathcal{I}_\sigma(m_t < 0) \mathcal{I}_\sigma(m_t < 0) \mathcal{I}_\sigma(-m_t > 0)
$$
$$
\sim \int \mathrm{d}x \int \prod_{i=1}^{4} \mathrm{d}\omega_i \; \delta(\omega - \omega_1 - \omega_2 - \omega_3 - \omega_4) \int \prod_{j=1}^{3} \mathrm{d}p_j
$$
$$
\times \delta(\omega_1 - \epsilon_{p1}^s) \delta(\omega_2 - \epsilon_{p2}^t) \delta(\omega_3 - \epsilon_{p3}^t) \delta(\omega_4) f^s(p_1) f^t(p_2) f^t(p_3)
$$
$$
\sim \int \mathrm{d}x \int \prod_{j=1}^{3} \mathrm{d}p_j \; \delta(\omega - \epsilon_{p1}^s - \epsilon_{p2}^t - \epsilon_{p3}^t) f^s(p_1) f^t(p_2) f^t(p_3)
$$
(B.17)

The unquenched system has a delta function $\delta(p_1 + p_2 + p_3 - q)$ in the integrand, while the quenched system broadens this delta function. In both cases, the spectral function presents a threshold at $\omega = |m_s| + 2|m_t|$. The simple analysis discussed here agrees with the results shown in Fig. 5.1 of Sec. 5.1.

## C    $I^{N_\pm}(\omega, q, t)$ in terms of $\mathcal{I}_\sigma(m)$

For reference, we list in this appendix the expressions for $I^{N_\pm}(\omega, q, t)$ in terms of $\mathcal{I}_\sigma(m)$, where the expressions for the latter are obtained in Sec. 5.2.1 and 5.2.2. According to

(2.12), the tr-RIXS response $I^{N\pm}(\omega, q, t)$ can be expressed as

$$I^{N+}(\omega, q, t) = \frac{S\tau^3}{\sqrt{2\pi}} \iint_{-\infty}^{\infty} dt_2 \, dt_2' \int_{-\infty}^{\infty} dx e^{i\omega(t_2' - t_2) - iqx} g(t_2, t)^2 \, g(t_2', t)^2 \, F^{N+},$$

$$I^{N-}(\omega, q, t) = \frac{S\tau^3}{\sqrt{2\pi}} \iint_{-\infty}^{\infty} dt_2 \, dt_2' \int_{-\infty}^{\infty} dx e^{i\omega(t_2' - t_2) - iqx} g(t_2, t)^2 \, g(t_2', t)^2 \, F^{N-}, \qquad \text{(C.1)}$$

$$F^{N+} \equiv \mathcal{I}_{0,\mu}\mathcal{I}_{1,\sigma}\mathcal{I}_{2,\sigma}\mathcal{I}_{3,\mu}, \quad F^{N-} \equiv \mathcal{I}_{0,\sigma}\mathcal{I}_{1,\mu}\mathcal{I}_{2,\mu}\mathcal{I}_{3,\sigma}, \quad S = \left(\frac{\lambda}{\left|m_s m_t^3\right|^{1/8} a_0}\right)^2.$$

For the case where $m_s > 0, m_t > 0$, we then have

$$F^{N+}(\omega, q, t) = \mathcal{I}_\sigma(-m_s < 0)\mathcal{I}_\sigma(m_t > 0)\mathcal{I}_\sigma(m_t > 0)\mathcal{I}_\sigma(-m_t < 0),$$
$$F^{N-}(\omega, q, t) = \mathcal{I}_\sigma(m_s > 0)\mathcal{I}_\sigma(-m_t < 0)\mathcal{I}_\sigma(-m_t < 0)\mathcal{I}_\sigma(m_t > 0), \qquad \text{(C.2)}$$

where we have used the duality relation $\mathcal{I}_\mu(m) = \mathcal{I}_\sigma(-m)$. For the case where $m_s > 0, m_t < 0$, we have instead

$$F^{N+}(\omega, q, t) = \mathcal{I}_\sigma(-m_s < 0)\mathcal{I}_\sigma(m_t < 0)\mathcal{I}_\sigma(m_t < 0)\mathcal{I}_\sigma(-m_t > 0),$$
$$F^{N-}(\omega, q, t) = \mathcal{I}_\sigma(m_s > 0)\mathcal{I}_\sigma(-m_t > 0)\mathcal{I}_\sigma(-m_t > 0)\mathcal{I}_\sigma(m_t < 0), \qquad \text{(C.3)}$$

while for $m_s < 0, m_t > 0$, we obtain

$$F^{N+}(\omega, q, t) = \mathcal{I}_\sigma(-m_s > 0)\mathcal{I}_\sigma(m_t > 0)\mathcal{I}_\sigma(m_t > 0)\mathcal{I}_\sigma(-m_t < 0),$$
$$F^{N-}(\omega, q, t) = \mathcal{I}_\sigma(m_s < 0)\mathcal{I}_\sigma(-m_t < 0)\mathcal{I}_\sigma(-m_t < 0)\mathcal{I}_\sigma(m_t > 0), \qquad \text{(C.4)}$$

and finally, for the case where $m_s < 0, m_t < 0$,

$$F^{N+}(\omega, q, t) = \mathcal{I}_\sigma(-m_s > 0)\mathcal{I}_\sigma(m_t < 0)\mathcal{I}_\sigma(m_t < 0)\mathcal{I}_\sigma(-m_t > 0),$$
$$F^{N-}(\omega, q, t) = \mathcal{I}_\sigma(m_s < 0)\mathcal{I}_\sigma(-m_t > 0)\mathcal{I}_\sigma(-m_t > 0)\mathcal{I}_\sigma(m_t < 0). \qquad \text{(C.5)}$$

# D   Details about the Calculation of $I^{N+}(\omega, q, t)$

As discussed in Sec. 5.2.3, we only need to care about the calculation of $I^{N+}(\omega, q, t)$ in Eq. (5.18) within the Neveu-Schwarz sector, and $I^{N-}(\omega, q, t)$ within the Neveu-Schwarz sector is obtained via

$$I^{N-}(m_s, m_t) = I^{N+}(-m_s, -m_t). \qquad \text{(D.1)}$$

This formula is understood as follows: the left-hand side takes the expression of the right-hand side, which depends on the sign of the mass parameters; then the expression is calculated using the original mass parameters of the left-hand side. In this section, we present the details of the calculation of $I^{N+}(\omega, q, t)$ within the Neveu-Schwarz sector.

Let's consider the case with $m_s > 0, m_t > 0$, then the product $\mathcal{I}_0\mathcal{I}_1\mathcal{I}_2\mathcal{I}_3$ reads

$$\mathcal{I}_0\mathcal{I}_1\mathcal{I}_2\mathcal{I}_3/(\bar{\sigma}_s^2\bar{\sigma}_t^6)$$

$$= \int_{-\infty}^{\infty} \frac{v dp}{2\pi} \frac{e^{-i\tilde{t}\epsilon_p^s + ipx}}{\epsilon_p^s} \int_{-\infty}^{\infty} \frac{v dp}{2\pi} \frac{e^{-i\tilde{t}\epsilon_p^t + ipx}}{\epsilon_p^t}$$

$$\times \exp\left\{-\frac{2}{\pi} \int_0^{\infty} dk \left[f_s(k, t_2)|x| + 2f_t(k, t_2)\mathcal{G}_t(k, x, t_2 + t_2') + f_t(k, t_2)|x|\right]\right\}$$

$$+ \int_{-\infty}^{\infty} \frac{v\mathrm{d}p}{2\pi} \frac{e^{-i\tilde{t}\epsilon_p^s + ipx}}{\epsilon_p^s} \int_{-\infty}^{\infty} \frac{v\mathrm{d}k}{2\pi} \frac{2}{\epsilon_k^t} \operatorname{Re}\left[u_k^t(t_2) e^{-ik|x| - i\epsilon_k^t |t_2' + t_2|}\right]$$

$$\times \exp\left\{-\frac{2}{\pi} \int_0^{\infty} \mathrm{d}k \left[f_s(k, t_2)|x| + 3f_t(k, t_2)\mathcal{G}_{t,x,t_2+t_2'}(k)\right]\right\}$$

$$+ \int_{-\infty}^{\infty} \frac{v\mathrm{d}p}{2\pi} \frac{e^{-i\tilde{t}\epsilon_p^t + ipx}}{\epsilon_p^t} \int_{-\infty}^{\infty} \frac{v\mathrm{d}k}{2\pi} \frac{2}{\epsilon_k^s} \operatorname{Re}\left[u_k^s(t_2) e^{-ik|x| - i\epsilon_k^s |t_2' + t_2|}\right]$$

$$\times \exp\left\{-\frac{2}{\pi} \int_0^{\infty} \mathrm{d}k \left[f_s(k, t_2)\mathcal{G}_s(k, x, t_2 + t_2') + 2f_t(k, t_2)\mathcal{G}_t(k, x, t_2 + t_2') + f_t(k, t_2)|x|\right]\right\}$$

$$+ \int_{-\infty}^{\infty} \frac{v\mathrm{d}k}{2\pi} \frac{2}{\epsilon_k^s} \operatorname{Re}\left[u_k^s e^{-ik|x| - i\epsilon_k^s |t_2' + t_2|}\right] \int_{-\infty}^{\infty} \frac{v\mathrm{d}k}{2\pi} \frac{2}{\epsilon_k^t} \operatorname{Re}\left[u_k^t(t_2) e^{-ik|x| - i\epsilon_k^t |t_2' + t_2|}\right]$$

$$\times \exp\left\{-\frac{2}{\pi} \int_0^{\infty} \mathrm{d}k \left[f_s(k, t_2)\mathcal{G}_s(k, x, t_2 + t_2') + 3f_t(k, t_2)\mathcal{G}_t(k, x, t_2 + t_2')\right]\right\}, \qquad \text{(D.2)}$$

where $f(k, t_2) = u_k^*(t_2) u_k(t_2)$, and the subscripts $s, t$ label the association with the singlet and triplet masses. For an illustration of the tricks used to deal with these integrals, we take the first term of Eq. (D.2) as an example. It can be separated into two parts as $\bar{\sigma}_s^2 \bar{\sigma}_t^6 (U_1 + U_2)$:

$$U_1 = \quad \Theta(x)\Theta(x - v|t_2 + t_2'|) \iint_{-\infty}^{\infty} \frac{\mathrm{d}p_1 \mathrm{d}p_2}{(2\pi)^2} \frac{v^2}{\epsilon_{p_1}^s \epsilon_{p_2}^t}$$

$$\times \exp\left\{ip_1 x + ip_2 x - \frac{2}{\pi} \int_0^{\infty} \mathrm{d}k \left[f_s(k, t_2) + f_t(k, t_2)\right] x\right\}$$

$$\times \exp\left\{-i\tilde{t}\epsilon_{p_1}^s - i\tilde{t}\epsilon_{p_2}^t - \frac{4}{\pi} \int_0^{\infty} \mathrm{d}k \, f_t(k, t_2)\epsilon_k^{t\,\prime} |t_2 + t_2'|\right\} + (x \to -x),$$

$$U_2 = \quad \Theta(x)\Theta(-x + v|t_2 + t_2'|) \iint_{-\infty}^{\infty} \frac{\mathrm{d}p_1 \mathrm{d}p_2}{(2\pi)^2} \frac{v^2}{\epsilon_{p_1}^s \epsilon_{p_2}^t}$$

$$\times \exp\left\{ip_1 x + ip_2 x - \frac{2}{\pi} \int_0^{\infty} \mathrm{d}k \left[f_s(k, t_2) + f_t(k, t_2)\right] x - \frac{4}{\pi} \int_{k_0}^{\infty} \mathrm{d}k \, f_t(k, t_2) x\right\}$$

$$\times \exp\left\{-i\tilde{t}\epsilon_{p_1}^s - i\tilde{t}\epsilon_{p_2}^t - \frac{4}{\pi} \int_0^{k_0} \mathrm{d}k \, f_t(k, t_2)\epsilon_k^{t\,\prime} |t_2 + t_2'|\right\} + (x \to -x), \qquad \text{(D.3)}$$

where the cutoff momentum $k_0$ is determined as

$$k_0 = \sqrt{\frac{\tilde{v}^2 m_1^2}{v^4 - \tilde{v}^2 v^2}}, \quad \tilde{v} = \frac{|x|}{|t_2 + t_2'|} \approx \frac{|x|}{2t}. \qquad \text{(D.4)}$$

For the evaluation of $I^{N+}(\omega, q, t)$, we need to take the convolution of $U_1, U_2$ over $t_2, t_2'$. Under the approximation of Eq. (D.4), this can be easily done:

$$\tilde{U}_1 \equiv \quad \iint_{-\infty}^{\infty} \mathrm{d}t_2 \mathrm{d}t_2' \, e^{i\omega(t_2' - t_2)} g(t_2, t)^2 g(t_2', t)^2 U_1$$

$$\approx \quad \Theta(x)\Theta(x - 2vt) \iint_{-\infty}^{\infty} \frac{\mathrm{d}p_1 \mathrm{d}p_2}{(2\pi)^2} \frac{v^2}{\epsilon_{p_1}^s \epsilon_{p_2}^t} \frac{\tau}{4\pi\sqrt{2\pi}} e^{i[p_1 + p_2 + iA_s(\infty) + iA_t(\infty)]x}$$

$$\times e^{-4tB_t(\infty)} e^{-\frac{\tau^2}{2}[(-\omega + \epsilon_{p_1}^s + \epsilon_{p_2}^t)^2 - 4B_t(\infty)^2]} + (x \to -x),$$

$$\tilde{U}_2 \equiv \quad \iint_{-\infty}^{\infty} \mathrm{d}t_2 \mathrm{d}t_2' \, e^{i\omega(t_2' - t_2)} g(t_2, t)^2 g(t_2', t)^2 U_2$$

$$\approx \quad \Theta(x)\Theta(-x + 2vt) \iint_{-\infty}^{\infty} \frac{\mathrm{d}p_1 \mathrm{d}p_2}{(2\pi)^2} \frac{v^2}{\epsilon_{p_1}^s \epsilon_{p_2}^t} \frac{\tau}{4\pi\sqrt{2\pi}} e^{i[p_1 + p_2 + iA_s(\infty) + i3A_t(\infty) - 2iA_t(k_0)]x}$$

$$\times e^{-4tB_t(k_0)} e^{-\frac{\tau^2}{2}[(-\omega+\epsilon_{p_1}^s+\epsilon_{p_2}^t)^2-4B_t(k_0)^2]} + (x \to -x), \tag{D.5}$$

where the functions $A(k), B(k)$ are defined as

$$A(k) = \frac{2}{\pi} \int_0^k \mathrm{d}p \; u_p^*(t)u_p(t), \quad B(k) = \frac{2}{\pi} \int_0^k \mathrm{d}p \; u_p^*(t)u_p(t)\epsilon_p'. \tag{D.6}$$

We can see that $\tilde{U}_1$ is exponentially small in $t$, and $U_2$ is of considerable size only if $\tilde{v} \ll v$, such that it can be approximated as

$$\tilde{U}_2 \approx \iint_{-\infty}^{\infty} \frac{\mathrm{d}p_1 \mathrm{d}p_2}{(2\pi)^2} \frac{v^2}{\epsilon_{p_1}^s \epsilon_{p_2}^t} \frac{\tau}{4\pi\sqrt{2\pi}} e^{i[p_1+p_2+iA_s(\infty)+i3A_t(\infty)]|x|-\frac{1}{\gamma^2}\left(\frac{x}{2vt}\right)^2} e^{-\frac{\tau^2}{2}(-\omega+\epsilon_{p_1}^s+\epsilon_{p_2}^t)^2}, \tag{D.7}$$

where we have introduced a Gaussian weight with a small $\gamma > 0$ to implement the constraint that $\tilde{v} \ll v$. Finally, the contribution to $I^{N+}(\omega,q,t)$ from the first term of Eq. (D.2) is obtained by convolution of $\tilde{U}_2$ over $x$. For large $t$, this gives us

$$\begin{aligned}
&\int \mathrm{d}x \; e^{-iqx}\tilde{U}_2 \\
&\approx \iint_{-\infty}^{\infty} \frac{\mathrm{d}p_1 \mathrm{d}p_2}{(2\pi)^2} \frac{v^2}{\epsilon_{p_1}^s \epsilon_{p_2}^t} \frac{\tau}{4\pi\sqrt{2\pi}} \left[\frac{2(A_s+3A_t)}{(A_s+3A_t)^2+(p_1+p_2-q)^2}\right] e^{-\frac{\tau^2}{2}(-\omega+\epsilon_{p_1}^s+\epsilon_{p_2}^t)^2},
\end{aligned} \tag{D.8}$$

where we have shorted $A_s(\infty), A_t(\infty)$ for $A_s, A_t$ respectively. The remaining terms of Eq. (D.2) can be dealt with in the same spirit, so are the remaining cases with other choices of the Majorana masses.

# E    More Results for $I^{N+}(\omega,q,t)$

Here we present the expression for $I^{N+}(\omega,q,t)$ within the Neveu-Schwarz sector in different cases depending on the sign of the Majorana masses. For $m_s > 0, m_t > 0$, we have

$$I^{N+}(\omega,q,t) = S\bar{\sigma}_s^2\bar{\sigma}_t^6 \left[I_{1,>>}^{N+}(\omega,q) + I_{2,>>}^{N+}(\omega,q,t)\right], \quad S\bar{\sigma}_s^2\bar{\sigma}_t^6 = \frac{1.3578^8\lambda^2}{a_0^3}$$

$$I_{1,>>}^{N+} \approx \iint_{-\infty}^{\infty} \frac{\mathrm{d}p_1 \mathrm{d}p_2}{(2\pi)^2} \frac{v^2}{\epsilon_{p_1}^s \epsilon_{p_2}^t} \frac{\tau}{4\pi\sqrt{2\pi}} \frac{2(A_s+3A_t)e^{-\frac{\tau^2}{2}(-\omega+\epsilon_{p_1}^s+\epsilon_{p_2}^t)^2}}{(A_s+3A_t)^2+(p_1+p_2-q)^2},$$

$$I_{2,>>}^{N+} \approx -\iint_{-\infty}^{\infty} \frac{\mathrm{d}p_1 \mathrm{d}p_2}{(2\pi)^2} \frac{v^2}{\epsilon_{p_1}^s \epsilon_{p_2}^t} \frac{\tau}{4\pi\sqrt{2\pi}} \left[\frac{2i(p_1+p_2-q)}{(A_s+3A_t)^2+(p_1+p_2-q)^2}\right]$$

$$\times \left[u_{p_2}^t(t)e^{-\frac{\tau^2}{2}[(-\omega+\epsilon_{p_1}^s)^2+(\epsilon_{p_2}^t)^2]} + u_{p_1}^s(t)e^{-\frac{\tau^2}{2}[(-\omega+\epsilon_{p_2}^t)^2+(\epsilon_{p_1}^s)^2]} - h.c.\right]$$

$$+ \iint_{-\infty}^{\infty} \frac{\mathrm{d}p_1 \mathrm{d}p_2}{(2\pi)^2} \frac{v^2}{\epsilon_{p_1}^s \epsilon_{p_2}^t} \frac{\tau}{4\pi\sqrt{2\pi}} \left[\frac{2(A_s+3A_t)}{(A_s+3A_t)^2+(p_1+p_2-q)^2}\right]$$

$$\times \left[u_{p_1}^s(t)u_{p_2}^t(t)e^{-\frac{\tau^2}{2}[\omega^2+(\epsilon_{p_1}^s+\epsilon_{p_2}^t)^2]} - u_{p_1}^s(t)u_{p_2}^t(t)^* e^{-\frac{\tau^2}{2}[\omega^2+(\epsilon_{p_1}^s-\epsilon_{p_2}^t)^2]} + h.c.\right], \tag{E.1}$$

For $m_s > 0, m_t < 0$, we have

$$I^{N+}(\omega,q,t) = S\bar{\sigma}_s^2\bar{\sigma}_t^6 \left[I_{1,><}^{N+}(\omega,q) + I_{2,><}^{N+}(\omega,q,t)\right],$$

$$I_{1,><}^{N_+} \approx \iiint_{-\infty}^{\infty} \frac{\mathrm{d}p_1\mathrm{d}p_2\mathrm{d}p_3}{(2\pi)^3} \frac{v^3}{\epsilon_{p_1}^s \epsilon_{p_2}^t \epsilon_{p_3}^t} \frac{\tau}{4\pi\sqrt{2\pi}} \frac{2(A_s+3A_t)e^{-\frac{\tau^2}{2}(-\omega+\epsilon_{p_1}^s+\epsilon_{p_2}^t+\epsilon_{p_3}^t)^2}}{(A_s+3A_t)^2+(p_1+p_2+p_3-q)^2},$$

$$I_{2,><}^{N_+} \approx -\iiint_{-\infty}^{\infty} \frac{\mathrm{d}p_1\mathrm{d}p_2\mathrm{d}p_3}{(2\pi)^3} \frac{v^3}{\epsilon_{p_1}^s \epsilon_{p_2}^t \epsilon_{p_3}^t} \frac{\tau}{4\pi\sqrt{2\pi}} \frac{2i(p_1+p_2+p_3-q)}{(A_s+3A_t)^2+(p_1+p_2+p_3-q)^2}$$

$$\times \left[ u_{p_1,s}(t)e^{-\frac{\tau^2}{2}\left[(-\omega+\epsilon_{p_2}^t+\epsilon_{p_3}^t)^2+(\epsilon_{p_1}^s)^2\right]} + [(p_1,s)\leftrightarrow(p_2,t)] + [(p_1,s)\leftrightarrow(p_3,t)] - h.c. \right]$$

$$+ \iiint_{-\infty}^{\infty} \frac{\mathrm{d}p_1\mathrm{d}p_2\mathrm{d}p_3}{(2\pi)^3} \frac{v^3}{\epsilon_{p_1}^s \epsilon_{p_2}^t \epsilon_{p_3}^t} \frac{\tau}{4\pi\sqrt{2\pi}} \frac{2(A_s+3A_t)}{(A_s+3A_t)^2+(p_1+p_2+p_3-q)^2}$$

$$\times \left[ u_{p_1}^s(t)u_{p_2}^t(t)e^{-\frac{\tau^2}{2}\left[(-\omega+\epsilon_{p_3}^t)^2+(\epsilon_{p_1}^s+\epsilon_{p_2}^t)^2\right]} - u_{p_1}^s(t)u_{p_2}^t(t)^*e^{-\frac{\tau^2}{2}\left[(-\omega+\epsilon_{p_3}^t)^2+(\epsilon_{p_1}^s-\epsilon_{p_2}^t)^2\right]} \right.$$

$$\left. +[(p_2,t)\leftrightarrow(p_3,t)] + [(p_1,s)\leftrightarrow(p_3,t)] + h.c. \right]$$

$$- \iiint_{-\infty}^{\infty} \frac{\mathrm{d}p_1\mathrm{d}p_2\mathrm{d}p_3}{(2\pi)^3} \frac{v^3}{\epsilon_{p_1}^s \epsilon_{p_2}^t \epsilon_{p_3}^t} \frac{\tau}{4\pi\sqrt{2\pi}} \frac{2i(p_1+p_2+p_3-q)}{(A_s+3A_t)^2+(p_1+p_2+p_3-q)^2}$$

$$\times \left[ u_{p_1}^s(t)u_{p_2}^t(t)u_{p_3}^t(t)e^{-\frac{\tau^2}{2}\left[\omega^2+(\epsilon_{p_1}^s+\epsilon_{p_2}^t+\epsilon_{p_3}^t)^2\right]} \right.$$

$$-u_{p_1}^s(t)u_{p_2}^t(t)u_{p_3}^t(t)^*e^{-\frac{\tau^2}{2}\left[\omega^2+(\epsilon_{p_1}^s+\epsilon_{p_2}^t-\epsilon_{p_3}^t)^2\right]}$$

$$-u_{p_1}^s(t)u_{p_2}^t(t)^*u_{p_3}^t(t)e^{-\frac{\tau^2}{2}\left[\omega^2+(\epsilon_{p_1}^s-\epsilon_{p_2}^t+\epsilon_{p_3}^t)^2\right]}$$

$$\left. +u_{p_1}^s(t)u_{p_2}^t(t)^*u_{p_3}^t(t)^*e^{-\frac{\tau^2}{2}\left[\omega^2+(\epsilon_{p_1}^s-\epsilon_{p_2}^t-\epsilon_{p_3}^t)^2\right]} - h.c. \right]. \tag{E.2}$$

For $m_s < 0, m_t > 0$, we have

$$I^{N_+}(\omega,q,t) = S\bar{\sigma}_s^2\bar{\sigma}_t^6 \left[ I_{1,<>}^{N_+}(\omega,q) + I_{2,<>}^{N_+}(\omega,q,t) \right],$$

$$I_{1,<>}^{N_+} \approx \int_{-\infty}^{\infty} \frac{\mathrm{d}p}{2\pi} \frac{v}{\epsilon_p^t} \frac{\tau}{4\pi\sqrt{2\pi}} \frac{2(A_s+3A_t)e^{-\frac{\tau^2}{2}(-\omega+\epsilon_p^t)^2}}{(A_s+3A_t)^2+(p-q)^2},$$

$$I_{2,<>}^{N_+} \approx -\int_{-\infty}^{\infty} \frac{\mathrm{d}p}{2\pi} \frac{v}{\epsilon_p^t} \frac{\tau}{4\pi\sqrt{2\pi}} \frac{2i(p-q)}{(A_s+3A_t)^2+(p-q)^2} \left[ u_p^t(t)e^{-\frac{\tau^2}{2}\left[\omega^2+(\epsilon_p^t)^2\right]} - h.c. \right]. \tag{E.3}$$

Finally, for $m_s < 0, m_t < 0$, we have

$$I^{N_+}(\omega,q,t) = S\bar{\sigma}_s^2\bar{\sigma}_t^6 \left[ I_{1,<<}^{N_+}(\omega,q) + I_{2,<<}^{2,N_+}(\omega,q,t) \right],$$

$$I_{1,<<}^{N_+} \approx \iint_{-\infty}^{\infty} \frac{\mathrm{d}p_1\mathrm{d}p_2}{(2\pi)^2} \frac{v^2}{\epsilon_{p_1}^t \epsilon_{p_2}^t} \frac{\tau}{4\pi\sqrt{2\pi}} \frac{2(A_s+3A_t)e^{-\frac{\tau^2}{2}(-\omega+\epsilon_{p_1}^t+\epsilon_{p_2}^t)^2}}{(A_s+3A_t)^2+(p_1+p_2-q)^2},$$

$$I_{2,<<}^{N_+} \approx -\iint_{-\infty}^{\infty} \frac{\mathrm{d}p_1\mathrm{d}p_2}{(2\pi)^2} \frac{v^2}{\epsilon_{p_1}^t \epsilon_{p_2}^t} \frac{\tau}{4\pi\sqrt{2\pi}} \left[ \frac{2i(p_1+p_2-q)}{(A_s+3A_t)^2+(p_1+p_2-q)^2} \right]$$

$$\times \left[ u_{p_2}^t(t)e^{-\frac{\tau^2}{2}\left[(-\omega+\epsilon_{p_1}^t)^2+(\epsilon_{p_2}^t)^2\right]} + u_{p_1}^t(t)e^{-\frac{\tau^2}{2}\left[(-\omega+\epsilon_{p_2}^t)^2+(\epsilon_{p_1}^t)^2\right]} - h.c. \right]$$

$$+ \iint_{-\infty}^{\infty} \frac{\mathrm{d}p_1\mathrm{d}p_2}{(2\pi)^2} \frac{v^2}{\epsilon_{p_1}^t \epsilon_{p_2}^t} \frac{\tau}{4\pi\sqrt{2\pi}} \left[ \frac{2(A_s+3A_t)}{(A_s+3A_t)^2+(p_1+p_2-q)^2} \right]$$

$$\times \left[ u^t_{p_1}(t) u^t_{p_2}(t) e^{-\frac{\tau^2}{2}\left[\omega^2 + (\epsilon^t_{p_1} + \epsilon^t_{p_2})^2\right]} - u^t_{p_1}(t) u^t_{p_2}(t)^* e^{-\frac{\tau^2}{2}\left[\omega^2 + (\epsilon^t_{p_1} - \epsilon^t_{p_2})^2\right]} + h.c. \right]. \quad \text{(E.4)}$$

For comparison, we present the corresponding results for the unpumped ladder, which can be obtained simply by setting $u_q = 0$, and contains only the steady state part:

$$I^{N_+}_{>>} \approx \frac{\tau}{\sqrt{2\pi}} \iint_{-\infty}^{\infty} \frac{\mathrm{d}p_1 \mathrm{d}p_2}{(2\pi)^2} \frac{v^2 e^{-\frac{\tau^2}{2}(-\omega + \epsilon^s_{p_1} + \epsilon^t_{p_2})^2}}{2\epsilon^s_{p_1}\epsilon^t_{p_2}} \delta(p_1 + p_2 - q),$$

$$I^{N_+}_{><} \approx \frac{\tau}{\sqrt{2\pi}} \iiint_{-\infty}^{\infty} \frac{\mathrm{d}p_1 \mathrm{d}p_2 \mathrm{d}p_3}{(2\pi)^3} \frac{v^3 e^{-\frac{\tau^2}{2}(-\omega + \epsilon^s_{p_1} + \epsilon^t_{p_2} + \epsilon^t_{p_3})^2}}{2\epsilon^s_{p_1}\epsilon^t_{p_2}\epsilon^t_{p_3}} \delta(p_1 + p_2 + p_3 - q),$$

$$I^{N_+}_{<>} \approx \frac{\tau}{\sqrt{2\pi}} \int_{-\infty}^{\infty} \frac{\mathrm{d}p}{2\pi} \frac{v e^{-\frac{\tau^2}{2}(-\omega + \epsilon^t_p)^2}}{2\epsilon^t_p} \delta(p - q),$$

$$I^{N_+}_{<<} \approx \frac{\tau}{\sqrt{2\pi}} \iint_{-\infty}^{\infty} \frac{\mathrm{d}p_1 \mathrm{d}p_2}{(2\pi)^2} \frac{v^2 e^{-\frac{\tau^2}{2}(-\omega + \epsilon^t_{p_1} + \epsilon^t_{p_2})^2}}{2\epsilon^t_{p_1}\epsilon^t_{p_2}} \delta(p_1 + p_2 - q). \quad \text{(E.5)}$$

The expressions for $I^{N_-}(\omega, q, t)$ within the Neveu-Schwarz sector are obtained from Eq. (D.1).

For the steady state part, the limit $\tau \to \infty$ can be safely taken such that the salient features are sharply defined. Here we consider only the case $m_s > 0, m_t < 0$, which is relevant to the choice of parameters specified in Eq. (3.1). By using Eq. (F.11), we obtain

$$\lim_{\tau \to \infty} I^{N_+}(\omega, q) \approx S\bar{\sigma}_s^2 \bar{\sigma}_t^6 \iiint_{-\infty}^{\infty} \frac{\mathrm{d}p_1 \mathrm{d}p_2 \mathrm{d}p_3}{(2\pi)^3} \frac{v^3}{\epsilon^s_{p_1}\epsilon^t_{p_2}\epsilon^t_{p_3}} \frac{1}{4\pi} \frac{2(A_s + 3A_t)}{(A_s + 3A_t)^2 + (p_1 + p_2 + p_3 - q)^2}$$
$$\times \delta(-\omega + \epsilon^s_{p_1} + \epsilon^t_{p_2} + \epsilon^t_{p_3}),$$

$$\lim_{\tau \to \infty} I^{N_-}(\omega, q) \approx S\bar{\sigma}_s^2 \bar{\sigma}_t^6 \int_{-\infty}^{\infty} \frac{\mathrm{d}p}{2\pi} \frac{v}{\epsilon^t_p} \frac{1}{4\pi} \frac{2(A_s + 3A_t)}{(A_s + 3A_t)^2 + (p - q)^2} \delta(-\omega + \epsilon^t_p).$$
$$\text{(E.6)}$$

The corresponding result for the unquenched ladder is obtained by setting $A_s + 3A_t \to 0$:

$$\lim_{\tau \to \infty} I^{N_+}(\omega, q) \approx S\bar{\sigma}_s^2 \bar{\sigma}_t^6 \iint_{-\infty}^{\infty} \frac{\mathrm{d}p_1 \mathrm{d}p_2}{2(2\pi)^3} \frac{v^3}{\epsilon^s_{q-p_2-p_3}\epsilon^t_{p_2}\epsilon^t_{p_3}} \delta(-\omega + \epsilon^s_{q-p_2-p_3} + \epsilon^t_{p_2} + \epsilon^t_{p_3}),$$
$$\lim_{\tau \to \infty} I^{N_-}(\omega, q) \approx S\bar{\sigma}_s^2 \bar{\sigma}_t^6 \frac{1}{4\pi} \frac{v}{\epsilon^t_q} \delta(-\omega + \epsilon^t_q). \quad \text{(E.7)}$$

These expressions are approximately evaluated at $q = 0$ and presented in the main text.

# F Details about the Calculation of $I^{M_\pm}(\omega, q, t)$

In this appendix, we present the detailed calculations of the smooth correlations with all the species of the Majorana fermions sitting in the Neveu-Schwarz sector. The tr-RIXS response in Eq. (5.8) is a two-fold convolution of the following correlator:

$$G_{\pm,q}(t_2, t_2') = \langle \psi(t_2) | U(t_2, t_2') M_{\pm,q} U(t_2', t_2) M_{\pm,-q} | \psi(t_2) \rangle. \quad \text{(F.1)}$$

The evolution operator $U(t_2', t_2)$ can be moved to the left of $M_{\pm,q}$ using the Baker-Campbell-Hausdorff formula :

$$e^A B e^{-A} = B + [A, B] + \frac{1}{2}[A, [A, B]] + \cdots + \frac{1}{n!} \overbrace{[A, [A, \cdots [A}^{n}, B]] + \cdots. \quad \text{(F.2)}$$

After that, $U(t_2', t_2)$ gets cancelled by $U(t_2, t_2')$, and the remaining operators act directly upon the state $|\psi(t_2)\rangle$:

$$|\psi(t_2)\rangle = |\psi_{\text{NS}}^0(t)\rangle \otimes |\psi_{\text{NS}}^1(t)\rangle \otimes |\psi_{\text{NS}}^2(t)\rangle \otimes |\psi_{\text{NS}}^3(t)\rangle,$$

$$|\psi_{\text{NS}}^0(t)\rangle = \mathcal{M}^{-1/2} e^{-\sum_{q>0} u_q c_q^{0\dagger}(m_s) c_{-q}^{0\dagger}(m_s)} |0\rangle_{m_s}, \tag{F.3}$$

$$|\psi_{\text{NS}}^i(t)\rangle = \mathcal{M}^{-1/2} e^{-\sum_{q>0} u_q c_q^{i\dagger}(m_t) c_{-q}^{i\dagger}(m_t)} |0\rangle_{m_t}, \quad i = 1, 2, 3,$$

where $\mathcal{M} = \prod_{q>0} \left(1 + u_q^* u_q\right)$ and the $u_q$ are defined in Eq. (3.10) with Majorana masses $m_s, M_{s,\text{pump}}$ or $m_t, M_{t,\text{pump}}$. The following calculations are purely algebraic manipulations of the anti-commutation relations and the Baker-Campbell-Hausdorff formula, and the final result is quite lengthy. To simplify matters, we then define the following quantities:

$$
\begin{aligned}
F(q, m, M_{\text{pump}}, t_2, t_1) &= \alpha_q(m) + \beta_q(m) u_q(m, M_{\text{pump}}, t_1, t_2); \\
K(q, m, M_{\text{pump}}, t_2, t_1) &= \alpha_q(m) u_q(m, M_{\text{pump}}, t_1, t_2) + \beta_q(m); \\
H(q, m, M_{\text{pump}}, t_2, t_1, \gamma) &= \alpha_q(m) e^{-\gamma \epsilon_q(m)} \\
&- (\alpha_q(m) e^{-\gamma \epsilon_q(m)} u_q(m, M_{\text{pump}}, t_1, t_2) + \beta_q(m) e^{\gamma \epsilon_q(m)}) v_q(m, M_{\text{pump}}, t_1, t_2); \\
J(q, m, M_{\text{pump}}, t_2, t_1, \gamma) &= \beta_q(m) e^{-\gamma \epsilon_q(m)} \\
&- (\alpha_q(m) e^{\gamma \epsilon_q(m)} + \beta_q(m) e^{-\gamma \epsilon_q(m)} u_q(m, M_{\text{pump}}, t_1, t_2)) v_q(m, M_{\text{pump}}, t_1, t_2)
\end{aligned}
\tag{F.4}
$$

where $\alpha_q, \beta_q$ are defined in Eq. (3.6), $v_q = u_q^*/(1 + u_q^* u_q)$ with $u_q$ defined in Eq. (3.10), and $\gamma = i(t_2' - t_2)$. Using these quantities, we can express Eq.(F.1) as

$$(-)LG_{\pm,q}(t_2, t_2') =$$

$$
\sum_{q_1 > 0} [F_+(q_1) K_\pm(q_1 + q) - K_+(q_1) F_\pm(q_1 + q)] [-H_+(q_1) J_\pm(q_1 + q) + J_+(q_1) H_\pm(q_1 + q)]
$$

$$
+ \sum_{q_1 > q} [K_+(q_1) F_\pm(q_1 - q) + F_+(q_1) K_\pm(q_1 - q)] [-J_+(q_1) H_\pm(q_1 - q) + H_+(q_1) J_\pm(q_1 - q)]
$$

$$
+ \sum_{0 < q_1 < q} [K_+(q_1) K_\pm(-q_1 + q) + F_+(q_1) F_\pm(-q_1 + q)]
$$

$$
\times [J_+(q_1) J_\pm(-q_1 + q) + H_+(q_1) H_\pm(-q_1 + q)], \tag{F.5}
$$

where we have made the dependence on $m$ and $M_{\text{pump}}$ and $t_1, t_2$ implicit for conciseness. Here on the r.h.s. of Eq. (F.5), all functions with a '+' subscript are functions of $m_t$ and $M_{t,\text{pump}}$, while functions with '-' subscript are functions of $m_s$ and $M_{s,\text{pump}}$. The expression in Eq. (F.5) is written for $q > 0$ or $q = 0$, while for $q < 0$, we have $G_{\pm,q}(t_2, t_2') = G_{\pm,|q|}(t_2, t_2')$, reflecting the reflection symmetry of our ladder geometry. In the infinite size limit $L \to 0$, the summation over momentum can be converted into an integral according to

$$\frac{1}{L} \sum_{q_1} \to \int \frac{dq_1}{2\pi}. \tag{F.6}$$

Since we are primarily concerned with the long time behavior with large $t_2, t_2'$, some of the terms in Eq. (F.5) will make exponentially small contributions and can be simply ignored. For identification of these negligible terms, it is useful to note that $u_q \sim e^{-2i\epsilon_q t_2} \sim e^{-2i\epsilon_q t}$, $v_q \sim u_q^*$. So we put $u$- and $v$-factors in pairs, regardless of their momentum labels. Those with one $u$- or $v$-factor left unpaired will be the ones that we can safely ignore. Finally, we substitute the explicit expressions for $\alpha_q, \beta_q$ and $u_q$ into $G_{\pm,q}(t_2, t_2')$, and perform the integrations in Eq. (5.8) over $t_2, t_2'$ according to

$$\int d\tilde{t}\, g(\tilde{t}, t)^2 e^{iA\tilde{t}} = \frac{e^{iAt}}{2\pi\tau^2} \int d\tilde{t}\, e^{-\left(\frac{\tilde{t}-t}{\tau}\right)^2 + iA(\tilde{t}-t)} = \frac{1}{2\sqrt{\pi}\tau} e^{-\frac{1}{4}A^2\tau^2 + iAt}. \tag{F.7}$$

The final result is

$$I^{M+}(\omega, q, t) = I_1^{M+}(\omega, q) + I_2^{M+}(\omega, q, t);$$

$$4I_1^{M+}(\omega, q) = R^{1a}(q_1 \to \{m_t, M_{t,\text{pump}}\}, q_1 + |q| \to \{m_t, M_{t,\text{pump}}\})$$

$$+ R^{1b}(q_1 \to \{m_t, M_{t,\text{pump}}\}, |q| - q_1 \to \{m_t, M_{t,\text{pump}}\});$$

$$R^{1a,M+}(q_1 \to \{m_t, M_{t,\text{pump}}\}, q_1 + |q| \to \{m_t, M_{t,\text{pump}}\}) =$$

$$\frac{2\tau}{4\pi\sqrt{2\pi}} \int_0^\infty \frac{dq_1}{2\pi} \cos^2\left(\theta_{q_1} - \theta_{q_1+|q|}\right) \frac{1}{1 + u_{q_1}^* u_{q_1}} \frac{u_{q_1+|q|}^* u_{q_1+|q|}}{1 + u_{q_1+|q|}^* u_{q_1+|q|}} e^{-\frac{\tau^2}{2}\left(\omega - \epsilon_{q_1} + \epsilon_{q_1+|q|}\right)^2}$$

$$+ \frac{2\tau}{4\pi\sqrt{2\pi}} \int_0^\infty \frac{dq_1}{2\pi} \cos^2\left(\theta_{q_1} - \theta_{q_1+|q|}\right) \frac{1}{1 + u_{q_1+|q|}^* u_{q_1+|q|}} \frac{u_{q_1}^* u_{q_1}}{1 + u_{q_1}^* u_{q_1}} e^{-\frac{\tau^2}{2}\left(\omega + \epsilon_{q_1} - \epsilon_{q_1+|q|}\right)^2}$$

$$+ \frac{2\tau}{4\pi\sqrt{2\pi}} \int_0^\infty \frac{dq_1}{2\pi} \sin^2\left(\theta_{q_1} - \theta_{q_1+|q|}\right) \frac{1}{1 + u_{q_1+|q|}^* u_{q_1+|q|}} \frac{1}{1 + u_{q_1}^* u_{q_1}} e^{-\frac{\tau^2}{2}\left(\omega - \epsilon_{q_1} - \epsilon_{q_1+|q|}\right)^2}$$

$$+ \frac{2\tau}{4\pi\sqrt{2\pi}} \int_0^\infty \frac{dq_1}{2\pi} \sin^2\left(\theta_{q_1} - \theta_{q_1+|q|}\right) \frac{u_{q_1+|q|}^* u_{q_1+|q|}}{1 + u_{q_1+|q|}^* u_{q_1+|q|}} \frac{u_{q_1}^* u_{q_1}}{1 + u_{q_1}^* u_{q_1}} e^{-\frac{\tau^2}{2}\left(\omega + \epsilon_{q_1} + \epsilon_{q_1+|q|}\right)^2};$$

$$R^{1b}(q_1 \to \{m_t, M_{t,\text{pump}}\}, |q| - q_1 \to \{m_t, M_{t,\text{pump}}\}) =$$

$$\frac{\tau}{4\pi\sqrt{2\pi}} \int_0^{|q|} \frac{dq_1}{2\pi} \sin^2\left(\theta_{q_1} + \theta_{|q|-q_1}\right) \frac{1}{1 + u_{q_1}^* u_{q_1}} \frac{u_{|q|-q_1}^* u_{|q|-q_1}}{1 + u_{|q|-q_1}^* u_{|q|-q_1}} e^{-\frac{\tau^2}{2}\left(\omega - \epsilon_{q_1} + \epsilon_{|q|-q_1}\right)^2}$$

$$+ \frac{\tau}{4\pi\sqrt{2\pi}} \int_0^{|q|} \frac{dq_1}{2\pi} \sin^2\left(\theta_{q_1} + \theta_{|q|-q_1}\right) \frac{1}{1 + u_{|q|-q_1}^* u_{|q|-q_1}} \frac{u_{q_1}^* u_{q_1}}{1 + u_{q_1}^* u_{q_1}} e^{-\frac{\tau^2}{2}\left(\omega + \epsilon_{q_1} - \epsilon_{|q|-q_1}\right)^2}$$

$$+ \frac{\tau}{4\pi\sqrt{2\pi}} \int_0^{|q|} \frac{dq_1}{2\pi} \cos^2\left(\theta_{q_1} + \theta_{|q|-q_1}\right) \frac{1}{1 + u_{|q|-q_1}^* u_{|q|-q_1}} \frac{1}{1 + u_{q_1}^* u_{q_1}} e^{-\frac{\tau^2}{2}\left(\omega - \epsilon_{q_1} - \epsilon_{|q|-q_1}\right)^2}$$

$$+ \frac{\tau}{4\pi\sqrt{2\pi}} \int_0^{|q|} \frac{dq_1}{2\pi} \cos^2\left(\theta_{q_1} + \theta_{|q|-q_1}\right) \frac{u_{|q|-q_1}^* u_{|q|-q_1}}{1 + u_{|q|-q_1}^* u_{|q|-q_1}} \frac{u_{q_1}^* u_{q_1}}{1 + u_{q_1}^* u_{q_1}} e^{-\frac{\tau^2}{2}\left(\omega + \epsilon_{q_1} + \epsilon_{|q|-q_1}\right)^2};$$

$$4I_2^{M+}(\omega, q) = R^{2a}(q_1 \to \{m_t, M_{t,\text{pump}}\}, q_1 + |q| \to \{m_t, M_{t,\text{pump}}\})$$

$$+ R^{2b}(q_1 \to \{m_t, M_{t,\text{pump}}\}, |q| - q_1 \to \{m_t, M_{t,\text{pump}}\});$$

$$R^{2a}(q_1 \to \{m_t, M_{t,\text{pump}}\}, q_1 + |q| \to \{m_t, M_{t,\text{pump}}\}) =$$

$$-\frac{2\tau}{4\pi\sqrt{2\pi}} \int_0^\infty \frac{dq_1}{2\pi} \cos^2\left(\theta_{q_1} - \theta_{q_1+|q|}\right) \frac{1}{1 + u_{q_1}^* u_{q_1}} \frac{1}{1 + u_{q_1+|q|}^* u_{q_1+|q|}}$$

$$\times \left(u_{q_1+|q|}^* u_{q_1} + u_{q_1}^* u_{q_1+|q|}\right) e^{-\frac{\tau^2}{4}\left(\omega - \epsilon_{q_1} + \epsilon_{q_1+|q|}\right)^2 - \frac{\tau^2}{4}\left(\omega + \epsilon_{q_1} - \epsilon_{q_1+|q|}\right)^2};$$

$$R^{2b}(q_1 \to \{m_t, M_{t,\text{pump}}\}, |q| - q_1 \to \{m_t, M_{t,\text{pump}}\}) =$$

$$\frac{\tau}{4\pi\sqrt{2\pi}} \int_0^{|q|} \frac{dq_1}{2\pi} \sin^2\left(\theta_{q_1} + \theta_{|q|-q_1}\right) \frac{1}{1 + u_{q_1}^* u_{q_1}} \frac{1}{1 + u_{|q|-q_1}^* u_{|q|-q_1}}$$

$$\times \left(u_{|q|-q_1}^* u_{q_1} + u_{q_1}^* u_{|q|-q_1}\right) e^{-\frac{\tau^2}{4}\left(\omega - \epsilon_{q_1} + \epsilon_{|q|-q_1}\right)^2 - \frac{\tau^2}{4}\left(\omega + \epsilon_{q_1} - \epsilon_{|q|-q_1}\right)^2}, \quad \text{(F.8)}$$

where the notation, $q_1 \to \{m_t, M_{t,\text{pump}}\}$, in the arguments of the functions $R^{1/2,a/b}$ indicates that the functions with a $q_1$ dependence, i.e., $u_{q_1}, \epsilon_{q_1}$, and $\theta_{q_1}$ are defined in terms

of the masses $m_t, M_{t,\text{pump}}$. As before, there are two parts to the signal, the steady state part, $I_1^{M+}$, and the transient part, $I_2^{M+}$. The fact that $I^{M+}(\omega, q, t)$ only depends on the magnitude of $q$ reflects the reflection symmetry of our ladder geometry.

For the correlations near $(0, \pi)$ encapsulated in $I^{M-}(\omega, q, t)$, we can obtain expressions in terms of the $R^{1/2,a/b}$ functions introduced above:

$$
\begin{aligned}
I^{M-}(\omega, q, t) &= I_1^{M-}(\omega, q) + I_2^{M-}(\omega, q, t) \\
4I_{1,2}^{M-}(\omega, q) &= \frac{1}{2} R^{1,2a}(q_1 \to \{m_t, M_{t,\text{pump}}\}, q_1 + |q| \to \{m_s, M_{s,\text{pump}}\}) \\
&\quad + \frac{1}{2} R^{1,2a}(q_1 \to \{m_s, M_{s,\text{pump}}\}, q_1 + |q| \to \{m_t, M_{t,pump}\}) \\
&\quad + R^{1,2b}(q_1 \to \{m_t, M_{t,\text{pump}}\}, |q| - q_1 \to \{m_s, M_{s,\text{pump}}\}), \quad \text{(F.9)}
\end{aligned}
$$

where we see $I^{M-}$ depends on both the triplet and singlet masses.

For comparison, we calculate the same response function with respect to the unpumped ladder. This calculation is simpler, where we replace the complicated state $|\psi(t_2)\rangle$ with the simple vacuum $|0\rangle = |0^0\rangle_{m_s} \otimes \prod_{i=1}^{3} \otimes |0^i\rangle_{m_t}$. After some trivial algebraic manipulations and integrations over $t_2, t_2'$, we arrive at

$$
\begin{aligned}
4I^{M+}(\omega, q) &= \frac{2\tau}{4\pi\sqrt{2\pi}} \int_0^\infty \frac{dq_1}{2\pi} \sin^2\left(\theta_{q_1}^t - \theta_{q_1+q}^t\right) e^{-\frac{\tau^2}{2}(\omega - \epsilon_{q_1}^t - \epsilon_{q_1+q}^t)^2} \\
&\quad + \frac{\tau}{4\pi\sqrt{2\pi}} \int_0^q \frac{dq_1}{2\pi} \cos^2\left(\theta_{q_1}^t + \theta_{q-q_1}^t\right) e^{-\frac{\tau^2}{2}(\omega - \epsilon_{q_1}^t - \epsilon_{q-q_1}^t)^2}, \\
4I^{M-}(\omega, q) &= \frac{\tau}{4\pi\sqrt{2\pi}} \int_0^\infty \frac{dq_1}{2\pi} \sin^2\left(\theta_{q_1}^t - \theta_{q_1+q}^s\right) e^{-\frac{\tau^2}{2}(\omega - \epsilon_{q_1}^t - \epsilon_{q_1+q}^s)^2} \\
&\quad + \frac{\tau}{4\pi\sqrt{2\pi}} \int_0^\infty \frac{dq_1}{2\pi} \sin^2\left(\theta_{q_1}^s - \theta_{q_1+q}^t\right) e^{-\frac{\tau^2}{2}(\omega - \epsilon_{q_1}^s - \epsilon_{q_1+q}^t)^2} \\
&\quad + \frac{\tau}{4\pi\sqrt{2\pi}} \int_0^q \frac{dq_1}{2\pi} \cos^2\left(\theta_{q_1}^t + \theta_{q-q_1}^s\right) e^{-\frac{\tau^2}{2}(\omega - \epsilon_{q_1}^t - \epsilon_{q-q_1}^s)^2}, \quad \text{(F.10)}
\end{aligned}
$$

where the function $\theta_q$ is defined in Eq. (3.6), and the superscripts $t, s$ denote the association with the triplet mass and singlet mass, respectively. An easier way to arrive at Eq. (F.10) is to set $u_q = 0$ in Eq. (F.8) and (F.9). In fact, by letting $M_{s,\text{pump}}, M_{t,\text{pump}} \to m_s, m_t$ in Eq. (F.8) and (F.9) for the pumped ladder, we indeed recover the same expressions in Eqs. (F.10) with corrections on the order of $(M_{s,\text{pump}} - m_s)^2$ or $(M_{t,\text{pump}} - m_t)^2$.

For the steady state part, the limit $\tau \to \infty$ can be safely taken such that the salient features are sharply defined. This limit can be conveniently taken by using the following relation:

$$
\lim_{\tau \to \infty} \frac{\tau}{\sqrt{2\pi}} e^{-\frac{\tau^2}{2}(\omega - \bar{\omega})^2} = \delta(\omega - \bar{\omega}) \quad \text{(F.11)}
$$

Using this, we arrive at the following expression:

$$
\begin{aligned}
\lim_{\tau \to \infty} I^{M+}(\omega, q) &= f_L \Theta(v|q| + \omega)\Theta(|m_t| - \epsilon_{|q|} - \omega) + f_R \Theta(\omega + |m_t| - \epsilon_{|q|})\Theta(v|q| - \omega) \\
&\quad + f_C \Theta(\omega - |m_t| + \epsilon_{|q|})\Theta(-\omega + \epsilon_{|q|} - |m_t|) \\
&\quad + g_L \Theta(-\omega - |m_t| - \epsilon_{|q|}) + h_L \Theta(\omega + |m_t| + \epsilon_{|q|})\Theta(-\omega - 2\epsilon_{|q|/2}) \\
&\quad + h_R \Theta(\omega - 2\epsilon_{|q|/2})\Theta(-\omega + |m_t| + \epsilon_{|q|}) + g_R \Theta(\omega - |m_t| - \epsilon_{|q|}), \quad \text{(F.12)}
\end{aligned}
$$

where the coefficient functions $f_L, f_R, f_C, g_L, g_R, h_L,$ and $h_R$ are defined as

$$f_L(\omega, q) = \frac{1}{16\pi^2} \cos^2\left(\theta_{q_1} - \theta_{q_1+|q|}\right) \frac{1}{1 + u_{q_1}^* u_{q_1}} \frac{u_{q_1+|q|}^* u_{q_1+|q|}}{1 + u_{q_1+|q|}^* u_{q_1+|q|}} \frac{1}{|\epsilon_{q_1}' - \epsilon_{q_1+|q|}'|},$$

$$f_R(\omega, q) = \frac{1}{16\pi^2} \cos^2\left(\theta_{q_1} - \theta_{q_1+|q|}\right) \frac{1}{1 + u_{q_1+|q|}^* u_{q_1+|q|}} \frac{u_{q_1}^* u_{q_1}}{1 + u_{q_1}^* u_{q_1}} \frac{1}{|\epsilon_{q_1}' - \epsilon_{q_1+|q|}'|},$$

$$f_C(\omega, q) = \frac{1}{32\pi^2} \sin^2\left(\theta_{q_1} + \theta_{|q|-q_1}\right) \frac{u_{q_1}^* u_{q_1} + u_{|q|-q_1}^* u_{|q|-q_1}}{\left(1 + u_{q_1}^* u_{q_1}\right)\left(1 + u_{|q|-q_1}^* u_{|q|-q_1}\right)} \frac{1}{|\epsilon_{q_1}' + \epsilon_{|q|-q_1}'|},$$

$$g_L(\omega, q) = \frac{1}{16\pi^2} \sin^2\left(\theta_{q_1} - \theta_{q_1+|q|}\right) \frac{u_{q_1+|q|}^* u_{q_1+|q|}}{1 + u_{q_1+|q|}^* u_{q_1+|q|}} \frac{u_{q_1}^* u_{q_1}}{1 + u_{q_1}^* u_{q_1}} \frac{1}{|\epsilon_{q_1}' + \epsilon_{q_1+|q|}'|},$$

$$g_R(\omega, q) = \frac{1}{16\pi^2} \sin^2\left(\theta_{q_1} - \theta_{q_1+|q|}\right) \frac{1}{1 + u_{q_1+|q|}^* u_{q_1+|q|}} \frac{1}{1 + u_{q_1}^* u_{q_1}} \frac{1}{|\epsilon_{q_1}' + \epsilon_{q_1+|q|}'|},$$

$$h_L(\omega, q) = \frac{1}{16\pi^2} \cos^2\left(\theta_{q_1} + \theta_{|q|-q_1}\right) \frac{u_{|q|-q_1}^* u_{|q|-q_1}}{1 + u_{|q|-q_1}^* u_{|q|-q_1}} \frac{u_{q_1}^* u_{q_1}}{1 + u_{q_1}^* u_{q_1}} \frac{1}{|\epsilon_{q_1}' - \epsilon_{|q|-q_1}'|},$$

$$h_R(\omega, q) = \frac{1}{16\pi^2} \cos^2\left(\theta_{q_1} + \theta_{|q|-q_1}\right) \frac{1}{1 + u_{|q|-q_1}^* u_{|q|-q_1}} \frac{1}{1 + u_{q_1}^* u_{q_1}} \frac{1}{|\epsilon_{q_1}' - \epsilon_{|q|-q_1}'|}, \quad \text{(F.13)}$$

where the value of $q_1 = q_1(\omega, q)$ in each of the these formulas is determined by the solution of the following equations (note that $q_1 > 0$ is required):

$$f_L: \quad +\epsilon_{q_1} - \epsilon_{q_1+|q|} = \omega,$$

$$f_R: \quad -\epsilon_{q_1} + \epsilon_{q_1+|q|} = \omega,$$

$$f_C: \quad -\epsilon_{q_1} + \epsilon_{|q|-q_1} = |\omega|,$$

$$g_L: \quad -\epsilon_{q_1} - \epsilon_{q_1+|q|} = \omega,$$

$$g_R: \quad +\epsilon_{q_1} + \epsilon_{q_1+|q|} = \omega,$$

$$h_L: \quad -\epsilon_{q_1} - \epsilon_{|q|-q_1} = \omega,$$

$$h_R: \quad +\epsilon_{q_1} + \epsilon_{|q|-q_1} = \omega. \quad \text{(F.14)}$$

The existence of a solution to the above equations is guaranteed by the step functions in Eq. (F.12). When there are two solutions to any of these equations, they contribute equally and the result in Eq. (F.13) takes both contributions into account. Thus one can choose any one of the two solutions in Eq. (F.13) as the value for $q_1$.

The corresponding result for the unquenched ladder can then be obtained by setting $u_q = 0$:

$$\lim_{\tau \to \infty} I^{M+}(\omega, q) = g_R \Theta(\omega - |m_t| - \epsilon_{|q|}) + h_R \Theta(\omega - 2\epsilon_{|q|/2})\Theta(-\omega + |m_t| + \epsilon_{|q|}), \quad \text{(F.15)}$$

where the coefficient functions are

$$g_R(\omega, q, u_q = 0) = \frac{1}{16\pi^2} \sin^2\left(\theta_{q_1} - \theta_{q_1+|q|}\right) \frac{1}{|\epsilon_{q_1}' - \epsilon_{q_1+|q|}'|},$$

$$h_R(\omega, q, u_q = 0) = \frac{1}{16\pi^2} \cos^2\left(\theta_{q_1} + \theta_{|q|-q_1}\right) \frac{1}{|\epsilon_{q_1}' - \epsilon_{|q|-q_1}'|}. \quad \text{(F.16)}$$

The lengthy expressions for $I^{M+}(\omega, q)$ are greatly simplified under the condition $|q| \ll |m_t|$ and $|u_q| \ll 1$, the case presented and discussed in the main text. For the $M_-$ sector, the analysis is similar. We do however need to pay attention to the fact that the $M_-$ correlator depends on both sets of masses, $(m_t, m_{t,\text{pump}})$ and $(m_s, m_{s,\text{pump}})$.

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
