# Peer review of "Ultrafast Control of Magnetic Correlations in a Heisenberg Spin Ladder"

_SciPost Physics_

## Round 3 · Referee Report · Anonymous · 2023-10-14

Strengths

1) Expert field-theoretical treatment of spin-ladder model

2) Systematic account of contributing terms in field-theoretical treatment

Weaknesses

1) Fully unrealistic treatment of ultrafast pumped spin systems, particularly irrelevant to RIXS

2) Extremely crude approximations, both of the spin ladder and of the driving pulse, and narrow parameter ranges of investigation

3) Presentation is rather light on physics and very heavy on technical details that lack a structure allowing a reasonable overview

Report

The authors use field-theoretical methods to study the response of a Heisenberg spin ladder to a double-quench switching of the rung interaction. The manuscript presents a curious mismatch between advanced and extensive analysis of time-dependence in multicomponent free-fermion models and borderline incoherent physics logic where the authors try to connect their considerations (of meV-scale processes in a 1D Heisenberg spin sector) to ultrafast RIXS probing of 2D iridates (i.e. of keV-scale spin-charge processes in strongly spin-orbit-coupled systems). The authors should give serious consideration to the context and content of their study before recasting it into one or more coherent contributions to the literature.

Beginning with the most general comments, the manuscript evinces a shortage of physics insight at all steps. While the analytical manipulations are impressive, their connection to ladder and pump-probe physics is poorly explained and, most fundamentally, the authors never state which of the restrictions on the scope of their considerations is enforced by the formalism they use.

The discussion of pumping is completely unrealistic: the authors show no understanding of how to change a superexchange constant. What they appeal to in Ref. [38] is nonlinear phononics, which is the intense driving of IR phonons well into the anharmonic regime so that the phonon modes can mix -- this is not Raman scattering and the authors of [38] said nothing direct about magnetic interactions. What the present authors want is "magnetophononics:"

M. Fechner, A. Sukhov, L. Chotorlishvili, C. Kenel, J. Berakdar, and N. A. Spaldin, Magnetophononics: Ultrafast spin control through the lattice, Phys. Rev. Mater. 2, 064401 (2018).

and this requires THz pumping to activate the phonons. There is no THz pumping in RIXS and no concept of a THz pump in the authors' present discussion. The shape of the double-quench "pulse" is also meaningless in ultrafast experiments: anythingn ultrafast enough to be a step is a very short pulse, and it is not possible to maintain this same pump for an arbitrarily long time. Many other authors have implemented Gaussian or other pulses [37,38], and so the reader is forced to ask why this route is not followed here ? Does the free-fermion formalism collapse if m is not a constant ?

Still on the pulse time structure, coherent light leads to rapid heating effects, such that long-duration pumping is impossible (in fact this is a primary reason why these experiments in quantum materials are ultrafast, and then the probing timescale is also fast). It is not at all clear that the time regimes discussed by the authors reflect these restrictions.

A further serious issue is dissipation: it seems that the authors implement none at all, in which case it is not surprising that all of their observations are "athermal" (there is no way for system to become thermal). However, what then is the concept of the "steady state" ? It seems that the systems is quenched twice and rings at its normal modes, but because it is composed of free fermions the only thing that can happen is that these continue to ring. Clearly the system has correlations and constraints that are not clearly presented, but without dissipation why do the effects relax at all ? At what level are the authors includimg damping implicitly in the "regulator functions" they introduce to avoid divergent integrals ?

Turning to specific comments:

Title: there is no "control" in this manuscript, but "response" might be appropriate.

Abstract: the concept of a timescale is unclear for free particles; the concept of long-time behavior is problematic (above); the system is obviously athermal; it is trivially obvious that changing the rung strength changes the spin correlations; as above there is no connection of the double-quench protocol to ultrafast pump-probe experiments.

Sec. 1: it is certainly true that core-hole processes are very complex, and thus the authors should be clear about how much motivation they are borrowing from RIXS ? Is it known when the RIXS response is "just the spin response" ? It would be helpful to review these criteria. This referee is not aware of any general agreement in the community that spinons are required to explain the magnetic spectra of perovskite iridates. The introduction suffers from poor referencing, in that more individual citations are required for specific statements instead of referencing in blocks. It is very important to note that femtosecond pumping means very high energies, very short timescales and very intense pulses that generally destroy and kind of order, which then recovers on its characteristic timescales. For in-plane processes in perovskites this can be inverse THz timescale, whereas between planes the correlations are destroyed by thermal effects. This referee agrees that simplified treatments can be helpful, but oversimplified ones are between worthless and dangerous. Is it helpful to classify correlations only as "trivial" and "nontrivial," and is it possible to find some more physically revealing terminology ? The general comments made above about pumping are relevant here, and make the authors' use of "imagine" and "believe" appropriate because regrettably nothing in this discussion is plausible.

Sec. 2: it is hard to follow the physics right from the outset because of the choice J = 1, which (i) drops the key chain interaction from the expressions and (ii) means that one may no longer count dimensions. The words about ladders being similar to 2D systems are quite implausible, given how far 1D Heisenberg systems are far from 2D spin-orbit-coupled ones. Can the authors explain the relevance of the Hubbard ladder origin, and provide some words on the meaning of the marginal terms that are neglected ? How low is the "low-energy field theory" ? Most ladder treatments are capable of including a full tight-binding band, but here it is not obvious if the energies under consideration are far lower (barely above the gap). Such basic physics should be expressed much more clearly than it is. If the fermions are correlated by the Z_2 condition, why do the authors persist in referring to them as "free" and "non-interacting" ?

In Fig. 2.1, what is the quantitative accuracy of these phase boundaries ? This system has been studied intensively by a range of numerical methods, and these should be cited to benchmark the results ? Is there a reason why all the J_\perp values are rather small, even though the perturbation of this parameter is the most important process considered ?

Sec. 3: here the physics statements made above about pumping are relevant. Stated bluntly, the double quench is painfully crude for quantum materials, and perhaps the authors should focus on cold-atoms instead. Is m really the only parameter one can use to apply the free-fermion formalism ? The parameter choices seem questionable throughout the manuscript, because the range of values consisdered is so restrictive in so many instances. Why can the authors not investigate a much bigger range of J (or m) variables and many more phases in Fig. 2.1 ? Is it in fact critical to remain only in the regime of weakly coupled chains. If the range is restricted by the nature of the treatment, this should be explained. The authors talk about the "energy pumped into the system," but it is not clear what effect this has, given that there are no dissipative or temperature terms. At this point in the discussion it seems as though the free fermions are simply ringing. From here the authors embark on the detailed analysis of one-body terms, but it is not at all clear how much these issues matter when (i) interactions are ignored and (ii) the approximations are in a regime of quadratic dispersion apparently not too far above the gap (as above, the choice of q is not well explained). There seems to be a typo around Eq. (3.8), with M_pump involved before the first quench (t < 0).

Sec. 4: now the authors show results that are somewhat difficult to assess, given the extensive approximations, curious time structure of the two quenches and narrow parameter ranges studied. This referee has to apologize that the physical meaning of the densely oscillating mode occupations is completely opaque. The statement that "only modes below a certain cutoff are populated" must depend on the pump protocol, and it is simply not clear what the authors wish to claim as a conclusion from this observation. The following figures show that when the pump disrupts the rung correlations, the leg correlations become stronger, but it is difficult to find a description of this physics. Can the changes to the correlation terms, and to their singlet and triplet parts, ever be larger than the authors show ? Once again the reader is struck by the very narrow range of parameters (and phases of Fig. 2.1) the authors actually investigate. This makes it rather difficult to understand the generality of any of the results. Again the authors' discussion of different time regimes seems to be entirely independent of the possibility that long pump durations simply melt any physical system.

Sec. 5: in the discussion of two-time correlations, the leading confusion is what is meant by "steady state" (above). After any pulsed driving the system must relax to zero, but are the authors describing continuous driving or the complete absence of dissipation ? It is also difficult to understand what the authors mean by an "equilibrium response" that seems to contain an excited state. Turning to the "transient part," what is causing it to decay away ? The lack of physical descriptions here leaves the reader no reference point to deduce what the authors might mean if their language were clearer.
From Sec. 5.1.1 the text suddenly refers to (\pi,\pi) and (\pi,0) points, which seem to be notation relevant to a 2D system; again a clearer explanation would be helpful, including of how one uses coherent light to probe (\pi,\pi). Fig. 5.3 again seems to draw rather general conclusions from a very narrow range of parameters. Here is it also not clear that 2 decay times are required to produce the form of these curves: if the authors performed a deeper analysis of their data then it would be helpful to show it. The end of Sec. 5.1.1 becomes extremely difficult to follow, and requires many of the following pages of more detailed results to understand. The switch of notation in Eq. (5.15) to c, with a previously unspecified superscript, is jarring. On the physics side, which two frequencies do the authors wish to highlight as being responsible for the beating in Fig. 5.5 ?

Beyond this point, the manuscript consists of the detailed analysis that gave the results presented so far. This referee would suggest placing all of Secs. 5.2 and 5.3 in appendices with a much more systematic structure including figures and tables (above). The discussion of the validity of the approximations seems rather short compared to the variety of energies, wavevectors, masses, pump-probe time structures and phases of the model (Fig. 2.1) that should in principle be covered, and a more systematic integration into the appendix would be beneficial.

Sec. 6 repeats many of the issues highlighted above. The authors have no scheme for "controlling" the Majorana parameters, having apparently missed entirely that this requires THz phonons; long pumping times have no physical relevance; there is no damping channel; the words about observing characteristics of the pump should be reviewed -- in physics we want to understand the system, not the pump we apply to it. Plenty of authors have performed nonequilibrium studies of ordered magnetic systems, phononic systems ([38] does only this) and ordered magnetic systems coupled to phonons. What is meant by "magnetic relaxation" ? It is curious that the authors begin to use the word "magnon" only in the section, having not used it at all in their (free fermion) analysis; the absence of magnon localization suggests only missing physics in their treatment. The comments about thermal and nonthermal are trivial in the absence of a heat bath. The pump profiles used in the cited previous work are entirely physically reasonable, which is why many authors used them; the double-step quench is an unrealisable combination of long and short timescales. With this it is very hard to agree that the authors have identified any features that are universal to physically realistic pump-probe studies of quantum materials. As above, they are in a completely different parameter space from perovskite iridates.

In summary, the manuscript presents good field-theoretical manipulations, but these are very extensive, come with little or no overview and are very hard to connect with any physics. On presentation, these sections would benefit from better use of appendices to remove the manipulations from the physics, a better section structuring, more figures to illustrate the words, and particularly the use of tables to collect together results that can be compared or contrasted. On physics, if the authors are limited (by the need for constant m) to doing quenches then this work would be much better presented as a discussion of nonequilibrium quench dynamics across all the qualitatively different phases and phase transitions in Fig. 2.1. Is the work limited to Heisenberg models ? And to ladders, or could one also consider gapless chains, which are more intuitively expressed as free fermions ? As above, the manuscript needs to be stripped of all the unrealistic words about real pump-probe studies, especially in materials as different from the 1D Heisenberg model as 2D iridates, but perhaps the authors can find more relevance for their quenching considerations in ultracold atomic systems on optical lattices. To conclude, the manuscript in its current form is entirely unfit for publication.

Requested changes

See report

---

## Round 3 · Referee Report · Anonymous · 2023-10-17

Strengths

1) nice, thourough and detailled analytical treatment of a pump-probe setup in the RIXS context; very clear discussion of the ansatzes, approximations and limitations made.

2) despite its length and large amount of technical details it reads well

3) interesting mapping of the two-leg Heisenberg ladder with plaquette-exchange allows treating of the nonequilibrium dynamics in terms of non-interacting Majorana fermions. This allows for the computation of complicated two-time correlation functions as needed for time-resolved RIXS

4) will be a useful point of reference for future work on related topics

Weaknesses

1) The quench protocol used ('double quench' in the value of the transverse spin coupling between the legs of the ladder) seems a bit artifical - it is justified by the scenario discussed in Ref. [38], but one wonders if one should not also consider time dependence in other coupling parameters.

2) Approximations needed in order to be able to analytically treat the system. For example, in Sec. 5.2.4 contributions $r \sim 0.3$ are shown to be neglected in addition to other ones, and it is hard to see how good or bad these assumptions are for the pumped system.

Report

The manuscript presents an analytical, field-theoretical treatment of a quench scenario, which can be used to compute quantities related to RIXS experiments. The paper is well written and contains a lot of interesting aspects and highlights. My fear is that some of the approximations made in order to analytically solve the problem might be stronger than assumed, but this needs to be seen in future research (e.g., by comparing to numerics). Also it is not clear to me, how well justified their quench scenario is, in which they change only one parameter of the Hamiltonian - should one not expect the value of the plaquette exchange (which also goes over both legs of the ladder) to be also affected by the arguments of Ref. [38], on which they base their modelling? This needs clarification. Otherwise I became aware of only rather minor issues, see below. This is a really interesting work, which should be published.

In the introduction, the authors refer to numerics using exact diagonalization and DMFT techniques to compute two-time correlation functions. However, state-of-the-art matrix product state techniques are also able to compute these quantities, which could be directly relevant for this quasi-1D system. A selection of publications, which should be considered as possible references by the authors, are listed in the detailled list of requested changes (and possibly further ones, here the authors should make a more thorough literature search).

This should be considered and included in the manuscript prior to publication (see also detailled list of requested changes).

Requested changes

1) Discussion for why one should only consider a quench in $J_\perp$ in the introduction, and why the change in other parameters like the plaquette exchange term can be neglected.

2) Consider the following publications as possible references (at the respective points in the intro and conclusions), and possibly further ones:
Phys. Rev. B 100, 195124 (2019)
Phys. Rev. B 102, 235141 (2020)
Phys. Rev. B 101, 180507(R) (2020)
Phys. Rev. Research 4, L012012 (2022)
Phys. Rev. B 105, 245126 (2022)
Commun. Phys. 6, 245 (2023)
SciPost Phys. 15, 166 (2023)

3) Check Eq. (2.1): are the limits correct for the periodic boundary conditions mentioned by the authors?

4) Just above Eq. (2.9): $\mathcal{Z}_2$?

5) Just below Eq. (2.14): [...] ?

6) Figs. 4.3 and 4.4: "original" in the panels = "equilibrium" in the caption? Similar in Fig. 4.5.

7) Fig. 5.4: labels at all figure axes would be helpful.

---

## Editorial Decision

awaiting_resubmission